# Probing Memes in LLMs: A Paradigm for the Entangled Evaluation World

## Abstract

Current evaluations of large language models (LLMs) often treat datasets and models in isolation, obscuring phenomena that only emerge from their collective interaction. Items (Questions) in datasets are reduced to labeled entries, disregarding the multidimensional properties they reveal when examined across model populations. Models, in turn, are summarized by overall scores such as accuracy, neglecting performance patterns that can only be captured through diverse data item interactions. To address this gap, this paper conceptualizes LLMs as composed of invisible memes, understood as cultural genes in the sense of Dawkins that function as replicating units of knowledge and behavior. Building on this perspective, the Probing Memes paradigm reconceptualizes evaluation as an entangled world of models and data. At its core lies the perception matrix, which captures interaction patterns and enables two complementary abstractions: probe properties, extending dataset characterization beyond labels, and phemotypes, revealing fine-grained capability structures of models. Applied to 9 datasets and 4,507 LLMs, Probing Memes reveals hidden capability structures and reveals phenomena invisible under traditional paradigms (e.g., elite models failing on problems that most models answer easily). This paradigm not only supports more informative, extensible, and fair benchmarks but also lays the foundation for population-based evaluation of LLMs.

## 1 Introduction

To advance the development and understanding of large language models (LLMs), researchers have devoted sustained efforts to improving benchmark design (Hendrycks et al., 2020; 2021; Srivastava et al., 2023). On one axis, increasingly challenging or cost-efficient datasets have been introduced (Phan et al., 2025; Polo et al., 2024; Schilling-Wilhelmi et al., 2025); on another, evaluation metrics have been expanded beyond simple accuracy to capture richer dimensions of performance (Ribeiro et al., 2020; Bommasani et al., 2023; Guo et al., 2025a). These efforts aim to enhance the effectiveness of evaluation. Further improvement efforts and limitations are detailed in Appendix A. However, persistent limitations remain: current approaches typically treat models and datasets in isolation, resulting in overly coarse descriptions. As a result, evaluations often lack depth and struggle to reveal phenomena that only emerge when data and models are analyzed in a population context (Figure 1 and 2).

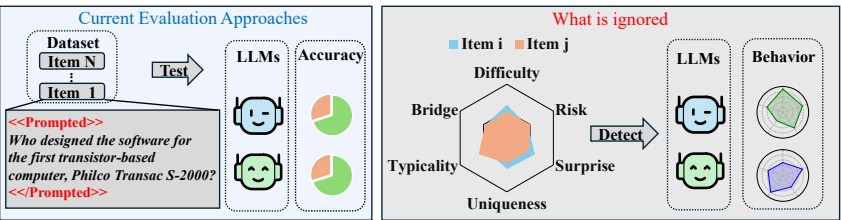

Figure 1: **Limitations of the current evaluation.** Current evaluation reveals only dataset-level accuracy across models and neglects fine-grained data and model attributes, which are observable only through population-level interactions and thus remain hidden under accuracy-based evaluation.

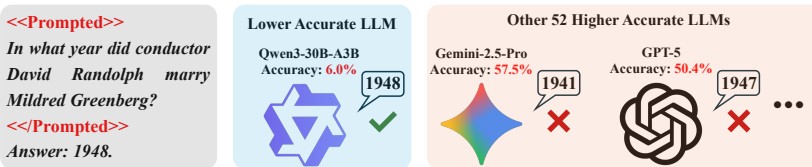

Figure 2: **A surprising case across LLMs.** Qwen3-30B-A3B, despite lower overall accuracy, succeeds on this item, whereas higher-accuracy LLMs (Gemini-2.5-Pro, GPT-5) fail.

On the data side, individual items are usually defined only by pre-assigned labels, without further characterization of their latent properties or their ability to differentiate model capabilities. This limits the explanatory power of datasets. For example, some items exhibit riskiness, where failing them strongly correlates with broader error patterns across the dataset. On the model side, although many new evaluation metrics have been proposed, they largely broaden the range of overall evaluation scores rather than revealing the deeper structure of model capabilities. Fine-grained differences are often obscured within overall scores, yet such differences typically surface only through population-level comparisons. For instance, certain elite models that excel in overall metrics nevertheless display anomalous errors on questions that most other models solve with ease.

These phenomena highlight the inadequacy of existing evaluation paradigms. To address this gap, this paper introduces the Probing Memes paradigm. As shown in Figure 3, the paradigm situates evaluation within an entangled world jointly shaped by interactions between data and models. Here, the notion of meme is borrowed[1] and metaphorically extended to the context of LLM evaluation, denoting latent units of model capability that can be revealed through probing. From this perspective, the abilities of LLMs are conceptualized as composed of latent memes. At the same time, each data item is treated as a Meme Probe (MP) designed to elicit and expose particular aspects of these capabilities. See Appendix A.3 for information about memetics.

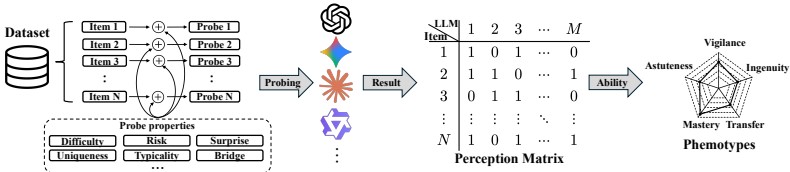

Figure 3: **Phemotype-based LLM probing framework.** Unlike traditional accuracy-focused approaches, this framework uses probes with diverse properties. By analyzing the resulting perception matrix, it better captures subtle LLM behaviors and reveals underlying abilities.

Interactions between probes and models yield a perception matrix. Analyzing this matrix enables two complementary abstractions. On the model side, latent memes can be organized into phenotypic memes (phemotypes), making structural differences in capabilities across models explicit and interpretable. On the data side, the ability of an item to elicit specific memes is captured by its Meme Probe Properties (MPPs). These properties are derived by generalizing across models and data contexts, revealing deeper characteristics of data and enabling more principled dataset optimization.

Crucially, both phemotypes and probe properties are designed to be extensible, allowing researchers to flexibly define new properties or phemotypes to meet diverse evaluation needs. In summary, Probing Memes enriches evaluation along two complementary axes: on the model side, it organizes latent capabilities into interpretable phemotypes; on the data side, it attributes probing power through MPPs. This dual abstraction moves beyond conventional reliance on overall metrics, enabling evaluation that is more flexible, fine-grained, and extensible.

---

[1] In *The Selfish Gene* (Dawkins, 1976), memes are described as "tunes, ideas, catch-phrases, clothes fashions, ways of making pots or of building arches," highlighting cultural units replicated through imitation.

The paradigm is validated through applications to 9 datasets and 4,507 LLMs. First, analyses are conducted on 28 models from 11 institutions across MATH-500, MMLU-Redux, and SimpleQA, focusing separately on probe-level and phemotype-level perspectives. At the probe level, the analysis illustrates how individual items in the entangled evaluation world can reveal fine-grained insights, such as the fact that datasets like MMLU-Redux contain a large number of seemingly simple questions that are nevertheless answered incorrectly by some elite models. At the phemotype level, the analysis reveals differences invisible to conventional evaluations, for example, models with similar accuracy may succeed on very different types of items. Second, by applying the paradigm to the Open LLM Leaderboard (Fourrier et al., 2024), which includes six datasets and 4,479 models, scalability is further demonstrated. This large-scale instantiation shows that Probing Memes sustains interpretability and flexibility at the population level. Taken together, these experiments validate the paradigm and reveal phenomena that remain hidden under conventional evaluations. Through both probe-level and phemotype-level analyses, such phenomena become explicit, underscoring the necessity of moving toward population-based, entangled evaluation.

In conclusion, the contributions of this work are threefold:

* It introduces the **Probing Memes** paradigm, which places evaluation within an entangled world shaped by data and model interactions;
* It formalizes two complementary abstractions, namely **phemotypes** and **meme probe properties**, enabling structured and extensible characterization of models and data;
* It validates the paradigm via large-scale experiments on 9 datasets and 4,507 LLMs, revealing fine-grained phenomena and insights remaining hidden under conventional evaluations.

## 2 THE PROBING MEMES PARADIGM

Building on the motivation outlined in Section 1, this section introduces the Probing Memes paradigm in detail. The exposition proceeds in three steps: first, formalizing the paradigm as an evaluation paradigm within the entangled world shaped by model–data interactions; second, characterizing the meme probe properties that enable the detection of latent memes; and third, defining phemotypes as structured representations of model capabilities.

### 2.1 FORMALIZATION OF THE PARADIGM

The Probing Memes paradigm can be formalized by specifying data, models, and their interaction. Let $\mathcal{D} = \{(x_i, y_i)\}_{i=1}^n$ denote a dataset of paired data items, where each pair consists of an input $x_i$ and a reference output $y_i$. Let $\mathcal{M} = \{M_j\}_{j=1}^m$ be a collection of LLMs, each viewed as a mapping $M_j : \mathcal{X} \to \mathcal{O}$. For any $(x_i, y_i)$, model $M_j$ produces an output $o_{ij} = M_j(x_i) \in \mathcal{O}$. A judging function $g : \mathcal{O} \times \mathcal{Y} \to \{0, 1\}$, applied to the paired outputs $(o_{ij}, y_i)$, returns 1 if $o_{ij}$ is judged correct with respect to $y_i$ and 0 otherwise, yielding a *perception unit*

$$P_{ij} = g\big(M_j(x_i), y_i\big). \tag{1}$$

Collecting all results yields the *perception matrix* $P \in \{0, 1\}^{n \times m}$, where rows correspond to data items and columns to models. Each probe $i$ is associated with a perception span $P_i$ (the row of $P$ corresponding to that probe), which forms the basis for higher-level probe properties and characterizes how the probe interacts with the model population.

In memetic terms, this paper posits an underlying *meme space* $\mathcal{V} = \{\mu_1, \mu_2, \ldots, \mu_R\}$ of elementary memes and associates each model $M_j$ with a subset $\mathcal{V}_j \subseteq \mathcal{V}$ of memes carried by that model. The perception matrix $P$ preserves the full structure of data–model interactions and records how these latent meme subsets respond on the finite set of probes in $\mathcal{D}$. In doing so, it provides the empirical interface between unobserved memes in models and their observable probe-level expressions, and supports probe properties that characterize how individual items reveal latent model capabilities.

### 2.2 MEME PROBE PROPERTIES

The degree to which a probe reveals distinct facets of model capability depends on its intrinsic properties. These properties, termed *Meme Probe Properties* (MPPs), offer a structured lens for

characterizing the probing capacity of individual data items within the joint interaction of model and data populations. Statistically, the perception matrix can be viewed as a sample of the model population. Treating models as random variables drawn from a broader ensemble increases their diversity or number, enhancing the reliability of MPP estimation. In this sense, MPPs are defined at the nexus of data and models: they represent stable characteristics of data items that become increasingly precise as the model population expands.

Formally, let $\mathcal{A} \subset \mathbb{R}^K$ denote a *property space*. Each probe $i$ is mapped to a property vector $\boldsymbol{a}_i = (a_i^{(1)}, a_i^{(2)}, \ldots, a_i^{(K)}) \in \mathcal{A}$, where coordinate $a_i^{(k)}$ represents one probe-side property dimension. In this work, $K$ is instantiated as 6, with dimensions corresponding to six well-designed properties: difficulty, risk, surprise, uniqueness, typicality, and bridge. The following outlines each probe property together with its intended role, definitions, and notation.

**Difficulty.** A probe should dynamically provide a difficulty baseline based on the performance of the model population. Formally, the difficulty of the $i$-th data item can be quantified as

$$d_i = 1 - \frac{1}{m} \sum_{j=1}^{m} P_{ij}, \tag{2}$$

where $P_{ij}$ denotes the perception unit of model $M_j$ on probe $i$ as defined in Equation 1, and $m = |\mathcal{M}|$ is the number of models in the population $\mathcal{M}$. Intuitively, $d_i$ measures the proportion of models that fail on probe $i$, so a higher value indicates greater difficulty relative to the population baseline.

**Risk.** A probe should reveal high-risk failure modes: failure on this probe is associated with elevated co-failure across many other probes. Formally, the risk of probe $i$ is defined as

$$r_i = \frac{1}{n-1} \sum_{k \neq i} \text{WJ}(i, k), \tag{3}$$

where $\text{WJ}(i,k)$ denotes the weighted Jaccard similarity between the perception spans of probes $i$ and $k$, given by $\text{WJ}(i,k) = \sum_{j=1}^{m} I_j \mathbb{1}_{\{(1-P_{ij}) \wedge (1-P_{kj})\}} / \sum_{j=1}^{m} I_j \mathbb{1}_{\{(1-P_{ij}) \vee (1-P_{kj})\}}$, and the weight $I_j$ of model $M_j$ is defined as $I_j = -\ln\left(1 - \frac{1}{n} \sum_{i=1}^{n} P_{ij}\right)$.

Intuitively, $\text{WJ}(i,k)$ measures how often two probes fail together relative to how often either one fails, so high risk corresponds to errors that co-occur broadly across probes. The weight $I_j$ reduces the influence of weak models while emphasizing the contribution of stronger models, ensuring that risk is driven by informative rather than trivial failure patterns. A detailed discussion of the role of $I_j$ and its statistical interpretation is provided in Appendix C.1.1.

**Surprise.** A probe should expose anomalies in which high-ability models fail on relatively easy probes, or conversely, low-ability models succeed on difficult probes. Formally, for the easy-side case, the surprise of probe $i$ is

$$s_i^{\text{easy}} = \left(-\ln d_i\right) \cdot \frac{1}{|W_i|} \sum_{j \in W_i} a_j,$$

where $d_i$ is the difficulty of probe $i$ (Equation 2), $W_i = \{j \mid P_{ij} = 0\}$ is the set of model indices such that $M_j$ fails probe $i$, and $a_j$ denotes the normalized accuracy of model $M_j$ across all probes.

Intuitively, $s_i^{\text{easy}}$ becomes large when a probe is solved by most models but disproportionately failed by stronger ones, while $s_i^{\text{hard}}$ highlights the reverse case. The formal definition of $s_i^{\text{hard}}$ is provided in Appendix C.1.2. Finally, the overall surprise of probe $i$ is given by

$$s_i = \tfrac{1}{2}\left(s_i^{\text{easy}} + s_i^{\text{hard}}\right). \tag{4}$$

**Uniqueness.** If a probe's response pattern does not materially reduce uncertainty about the responses to other probes, it should be flagged as highly unique. For consistency with the information-theoretic formulation, each probe $i$ is not only represented by its perception span $(P_{i1}, \ldots, P_{im})$, but also viewed as a binary random variable $P_i$ over the model population, where $P_i = 1$ indicates a correct response and $P_i = 0$ an incorrect one, and the vector entries serve as empirical samples of this

variable. The uniqueness of probe $i$ is then defined as

$$u_i = \frac{1}{n-1} \sum_{k \neq i}^{n} H(P_k \mid P_i), \tag{5}$$

where $H(P_k \mid P_i)$ is the conditional entropy of random variable $P_k$ given $P_i$, estimated empirically from the samples.

Intuitively, a low $u_i$ means that the model responses to probe $i$ substantially reduce the uncertainty about other probes, indicating stronger representativeness; conversely, a high $u_i$ implies that probe $i$ provides little predictive information about others, indicating stronger uniqueness. The detailed formal definition of $H(P_k \mid P_i)$ is provided in Appendix C.1.3.

To characterize the distinctiveness and commonality among probes' perception spans, this paper constructs a similarity graph from the perception matrix of all probes and applies Leiden community detection (Traag et al., 2019), yielding perception span clusters (i.e., sets of probes with highly similar perception spans).

**Cluster Construction.** Given two probes $i$ and $k$, their similarity $\text{sim}(P_i, P_k)$ is measured by the $\phi$-coefficient (see Appendix C.1.4). Here, each perception span is interpreted as a sample value of a Bernoulli random variable, whose expectation corresponds to the average difficulty of the probe. The $\phi$-coefficient thus measures the correlation between two such random variables. An undirected weighted graph $G = (V, E)$ is then defined, where each node corresponds to a probe, and an edge $(i, k)$ is included if $\text{sim}(P_i, P_k) > \tau$, with the edge weight set as the similarity value; here $\tau$ is a threshold controlling the sparsity of the graph. Applying Leiden community detection on this graph produces a partition $\mathcal{C} = \{C_1, C_2, \ldots, C_K\}$ of probes into clusters of highly similar difficulty patterns. Building on this cluster structure, this paper defines the *typicality* and *bridge* properties.

**Typicality.** A probe should be considered a prototype if its perception span shows high average similarity to other probes in the cluster. Formally, for probe $i \in C_l$, let $\mathcal{N}_i^{\text{intra}} = \{k \in C_l \mid (i, k) \in E\}$ denote the set of neighbors of $i$ within its own cluster. The typicality of probe $i$ is defined as

$$t_i = \frac{\sum_{k \in \mathcal{N}_i^{\text{intra}}} \text{sim}(P_i, P_k)}{|\mathcal{N}_i^{\text{intra}}|}. \tag{6}$$

**Bridge.** A probe should be considered a connector if its perception span shows substantive similarity to probes in multiple distinct clusters. Formally, for probe $i \in C_l$, let $\mathcal{N}_i^{\text{inter}} = \{k \notin C_l \mid (i, k) \in E\}$ denote the set of neighbors of $i$ in other clusters, and define $\kappa_i = |\{C_\ell \mid \exists k \in \mathcal{N}_i^{\text{inter}} \cap C_\ell\}|$ the number of distinct clusters spanned by probe $i$. Then the bridge property of probe $i$ is defined as the product of *participation* and *strength*:

$$b_i = \underbrace{\frac{\kappa_i}{\kappa_i + \text{median}_{r \in V} \kappa_r}}_{\text{Participation}} \times \underbrace{\frac{1}{|\mathcal{N}_i^{\text{inter}}|} \sum_{k \in \mathcal{N}_i^{\text{inter}}} \text{sim}(P_i, P_k)}_{\text{Strength}}. \tag{7}$$

Here, participation quantifies the extent to which a probe connects to multiple clusters, normalized by the population median, while strength captures the average cross-cluster similarity.

## 2.3 PHEMOTYPES OF LLMS

This subsection introduces model phemotypes, summarizing how memes are expressed across probe properties. Let $\mathcal{H}$ denote a phemotype space whose elements are low-dimensional descriptors of model behavior over probes. Given the set $A$ of available probe properties, any non-empty subset $S \subset A$ together with a mapping $f$ defines a phemotype, written $f(S) \in \mathcal{H}$. In this work, $f$ maps the normalized values of the properties in $S$ for each probe to nonnegative probe weights.

Let $\tilde{d}_i, \tilde{r}_i, \tilde{s}_i, \tilde{t}_i, \tilde{b}_i, \tilde{u}_i \in (0, 1)$ denote normalized probe properties, obtained via a generic normalization operator $\text{Norm}(\cdot)$ (See Appendix D.1). For a chosen phemotype, the mapping $f$ specifies how the relevant normalized properties are combined into a weight $w_i \geq 0$ for probe $i$. Given the resulting weights $w$, the phemotype score of model $M_j$ is defined as

$$\text{Phemotype}(M_j; w) = \text{Score}(w; P_{\cdot j}), \tag{8}$$

where $\mathrm{Score}(\cdot)$ is a weight-aggregated model score applied to the $j$th column of the perception matrix (See Appendix D.1.2). The five concrete phemotypes used in this work are summarized in Table 1, which specifies, for each phemotype, its semantic interpretation and the induced probe weights $w_i^{(\cdot)}$; additional details are discussed in Appendix B.

Table 1: **Definitions of LLM phemotypes with semantic interpretation.**

| Phemotype | Interpretation | Definition |
|---|---|---|
| **Vigilance** | Resist high-risk and counter-intuitive traps; maintain correctness where many models co-fail. | $w_i^{\mathrm{Vig}} = \tilde{r}_i \, \tilde{s}_i$ |
| **Mastery** | Proficiency on typical yet difficult cluster-core motifs; $scale_i$ denotes the cluster-scale factor. | $w_i^{\mathrm{Mas}} = \tilde{t}_i \, \tilde{d}_i \, scale_i$ |
| **Transfer** | Generalization across clusters or prompts; success on bridging and difficult probes. | $w_i^{\mathrm{Trf}} = \tilde{b}_i \, \tilde{d}_i$ |
| **Ingenuity** | Flexibility on unique and difficult probes; success on rare or non-canonical cases. | $w_i^{\mathrm{Ing}} = \tilde{u}_i \, \tilde{d}_i$ |
| **Astuteness** | Avoid elite traps; identify key cues on surprising probes where common priors mislead. | $w_i^{\mathrm{Ast}} = \tilde{s}_i$ |

## 3 EXPERIMENTS AND ANALYSIS

This section introduces the derivation of probe properties and the resulting characterization of model phemotypes, as showcased within an Entangled Evaluation World. For completeness, this paper also provide two forms of extended validation in the appendix: stability under different model populations (Appendix F) and robustness to hyperparameter choices (Appendix G).

### 3.1 EXPERIMENTAL SETUP

Under the proposed paradigm, this study evaluates 28 large language models from 11 providers, where models span small to large sizes. The study analyzes three reasoning modes: **default prompting (Base)**, **chain-of-thought prompting (CoT)**, and **internal reasoning (IR)**, definitions and prompts settings can be seen in Appendix J.1. These abbreviations are used consistently throughout the paper, including in figures and tables. Three widely used datasets across distinct tasks (mathematics, general knowledge, and question answering) are selected: MATH-500 (Lightman et al., 2023), MMLU-Redux (Gema et al., 2025), and SimpleQA (Wei et al., 2024). Further details and special cases appear in the Appendix J.

### 3.2 PROBE-LEVEL ANALYSIS

Within this evaluation paradigm, this paper performs probe-level analysis on the perception matrix to derive well-designed probe properties for meme detection. To improve the quality of probe properties, probes whose perception spans are all ones or all zeros in the perception matrix are excluded. For further details, refer to the Appendix J.2.

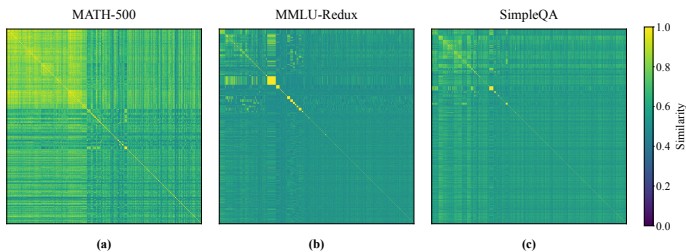

Figure 4: **Probe similarity heatmaps across datasets.**

**Distributions of Probes.** By computing the perception span similarity for each probe pair (Figure 4), the resulting similarity distributions differ markedly across datasets. MATH-500 exhibits higher probe similarity with clear blocks and repeated bands, whereas MMLU-Redux and SimpleQA show lower, more fragmented similarity with small clusters.

**More than Correctness: A Unified Property Space for Probes.** In the proposed paradigm, questions are treated as more than right or wrong. They are thus called probes. Each probe is represented by an expandable attribute vector and embedded in a unified property space. Figure 5 uses difficulty, uniqueness, and surprise as three axes and plots probes from each dataset. The distribution forms a funnel that narrows from easy to hard and shows a long tail of negative surprise at the hard end. The low-difficulty region is more dispersed. This suggests that easy probes can still produce unexpected behavior in both positive and negative directions at the group level. Specifically, MMLU-Redux contains many easy probes with high surprise, indicating that many top models fail on them. See Appendix C.2 for more visualizations of alternative property combinations.

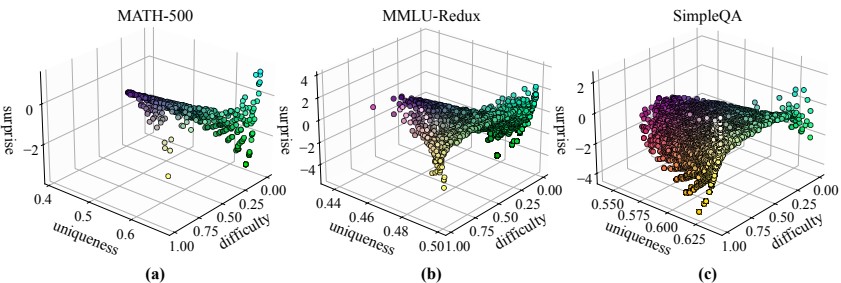

Figure 5: **Probe property distribution.** Axes show difficulty, uniqueness, and vertical surprise.

### 3.3 PHEMOTYPES OF LLMS

Building on the probe analysis, six probe properties are derived to characterize item-level behavioral attributes. Combining the perception matrix with these properties yields five model-level phemotypes, giving rise to **A Memetic Landscape for Scrutinizing LLMs**.

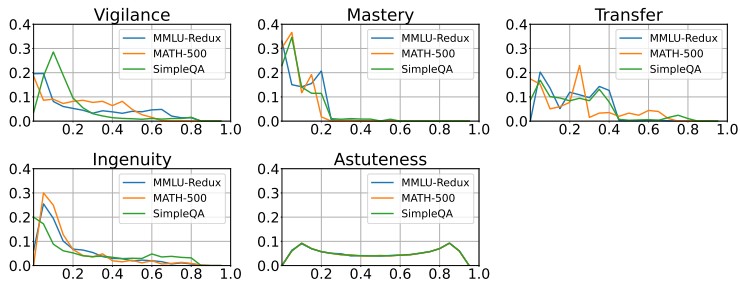

Figure 6: **Distribution of probe-level contributions to each phemotype.** This figure consists of five sub-plots, one per phemotype. In each sub-plot, each line (one per dataset) shows the distribution of probe-level weights, with the x-axis dividing the weight value range $[0, 1]$ into 20 equal-width intervals and the y-axis giving the fraction of probes in each interval.

**From Probe to Phemotypes.** Based on the design introduced in Section 2.3, each probe's properties are combined to detect memes in LLMs, thereby yielding the corresponding phemotypes. For each phemotype, Figure 6 plots the distributions of probe-level contribution weights across datasets, which in turn characterize a model's phemotype from its responses across probes.

**Accuracy versus phemotypes.** Figure 7 presents a comparison of accuracy and the five phemotype dimensions across all models under the three reasoning paradigms. Unlike the smooth trajectory of the accuracy curve, the phemotype curves exhibit nonparallel patterns with abrupt changes, crossings, and occasional reordering. These phenomena show how accuracy can alienate distinct be-

havioral characteristics, while phemotypes recover the latent diversity of memetic traits, providing evidence that models with the same accuracy may in fact display different behavioral patterns. It can be seen that even with comparable accuracy, gpt-4o-2024-11-20 (CoT) exhibits consistently lower phemotypes than qwen3-235b-a22b (IR), suggesting that the high accuracy of gpt-4o-2024-11-20 (CoT) relies more on routine or straightforward items, while its abilities in vigilance to traps and cross-cluster transfer are relatively weaker. The full tabular summaries and the per-dataset results are reported in Appendix D.2.

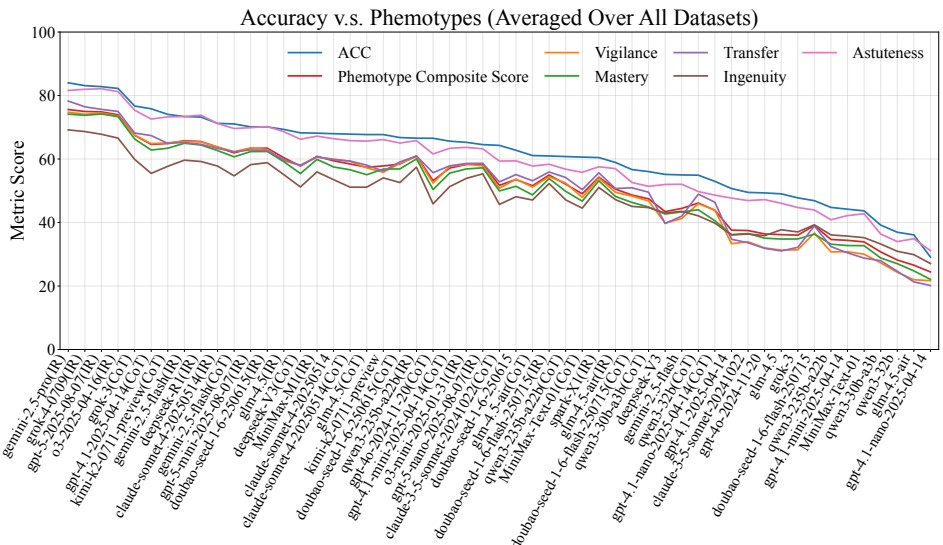

Figure 7: **Accuracy vs. phemotype scores.** Line plots show accuracy and phemotypes for all models under different reasoning modes, sorted by accuracy. The Phemotype Composite Score is the average of the five phemotypes.

# 4 TOWARD A LARGE ENTANGLED EVALUATION WORLD AT SCALE: OPEN LLM LEADERBOARD

This section applies the Probing Memes paradigm to the Open LLM Leaderboard. Valid results from 4,479 models across six datasets are collected to construct a high-dimensional perception matrix. This matrix supports meme-level characterization of models, revealing shared and divergent behaviors. Details on the models and datasets appear in Appendix H.1.

**Landscapes of Probe Properties.** By applying open evaluation results from over 4,479 models across six datasets, the properties of each probe within each dataset are well-characterized, whose distributions are shown in Figure 8. Overall, the distributions vary across different datasets. Among these 4,479 models, the MATH and MUSR datasets contain a relatively high proportion of difficult probes. On the difficulty side of MMLU-Pro, there are many questions with high surprise scores, suggesting that a large number of models with lower performance can correctly answer these difficult probes. Moreover, the probes in IFEval, GPQA-Diamond, and BBH exhibit relatively high uniqueness. The visualization of probe similarity can be seen in Appendix H.2.

**Commonality and Divergence Among Models Revealed by Phemotypes.** Figure 9 shows that models in the phemotype space are not uniformly distributed but instead form several clear clusters. Some datasets (e.g., MMLU-Pro) exhibit tightly packed and well-separated groups, indicating pronounced behavioral commonality and divergence. Using each model's reported base model on Hugging Face, models are organized into families. Colors indicate the top-20 families by size; unlabeled or other models are shown in gray. Notably, models from the same family tend to lie closer together in the visualization. Overall, these results demonstrate that phemotypes can uncover both similarities and differences among models. In other words, this paradigm can serve as a powerful tool to reveal similarities and differences between models, thereby **helping to investigate potential**

**relationships in their training data, base models, and training strategies.** Extended results and analysis are provided in Appendix E.3

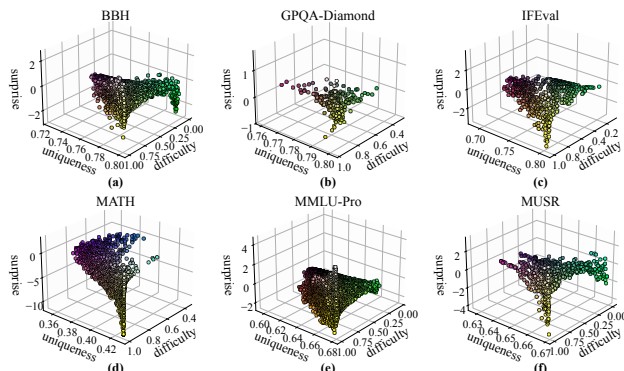

Figure 8: **Probe properties distributions across datasets of the Open LLM Leaderboard.**

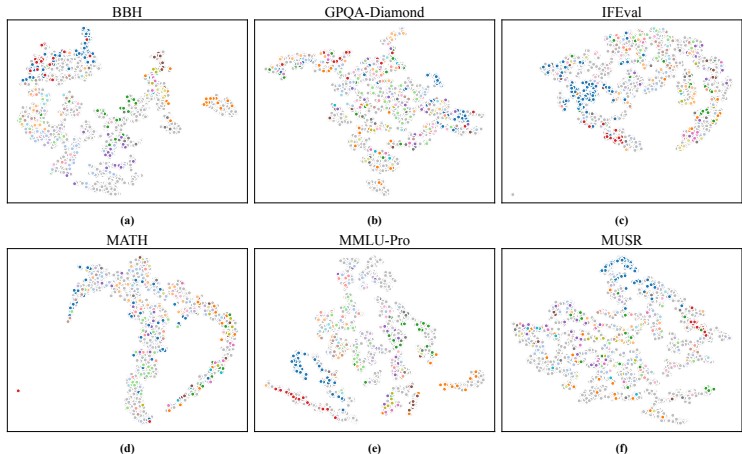

Figure 9: **Commonality and divergence among models revealed by phemotypes.** Each model is embedded with t-SNE from its five-dimensional phemotype representation. Model families are defined by the shared base model, and the 20 families with the largest numbers of models are color-coded (others are shown in gray). Nearby points indicate more similar phemotypic profiles.

## 5 APPLICATIONS AND CASE STUDIES

This section presents three representative application scenarios enabled by the Probing Memes paradigm, **task-aware model selection**, **cost-effective evaluation subset construction**, and **fine-grained behavioral analysis**, illustrating the practical utility of calibrated probe properties and model phemotypes. Full procedures and extended results are provided in Appendix E.

**Task-Aware Model Selection** A key insight of the Probing Memes paradigm is that phemotype scores do not impose a simple better–worse ordering: models with similar accuracy can display complementary behaviors, suggesting utility for task-aware routing. To test this, a feasibility study uses the difficulty dimension, which functions both as a probe property and, when aggregated, as its one-dimensional phemotype. Two model pairs from MATH-500 are chosen whose accuracies differ by only 0.22 points but whose difficulty phemotypes diverge (e.g., +4.83 vs. +3.02). On MATH-FULL, expert-annotated difficulty levels provide the routing signal: items at levels 4–5 are routed to the model with the higher difficulty phemotype, and levels 1–3 to the lower one.

Even this minimal policy yields consistent gains. For Pair 1, the models achieve 72.24% and 76.02%, and the balanced random-routing baseline reaches $73.89\% \pm 0.29$, whereas phemotype-guided routing attains **77.02%**. For Pair 2, the scores are 44.80% and 46.78%, with a $45.73\% \pm 0.32$ baseline, while routing improves performance to **48.47%**. These results show that phemotypes provide actionable signals for task-aware model selection and lightweight multi-model routing. Full settings and additional results appear in Appendix E.1.

**Cost-Effective Evaluation Subset Construction** Beyond analysis, the Probing Memes paradigm also supports tasks typically addressed by IRT, such as constructing compact evaluation subsets. Typicality-guided probe selection is compared with two strong IRT baselines, MetaBench (Kipnis et al., 2025) and TinyBench (Polo et al., 2024), across 4,479 models from the Open LLM Leaderboard. Unlike IRT methods, which fit latent abilities and item parameters, Probing Memes selects probes directly from their properties in the memetic space, without estimating latent parameters.

Despite this, Probing Memes achieves comparable reconstruction performance on most datasets. Under the MetaBench comparison, correlations reach $\rho = 0.977$ on MATH and $\rho = 0.993$ on IFEval (only 0.011 and 0.003 below IRT), and $\rho = 0.793$ on GPQA-Diamond versus IRT's $\rho = 0.884$. Under the TinyBench comparison, Probing Memes even attains lower reconstruction errors on MATH and GPQA-Diamond (MAE 0.026 and 0.023) than TinyBench (0.030 and 0.035). Full details appear in Appendix E.2. These results show that Probing Memes provides competitive subset-construction performance while offering added advantages, including interpretability and explicit probe-level insights unavailable to classical IRT methods.

**Fine-Grained Behavioral Case Study** The Probing Memes paradigm also supports probe-level diagnosis of specific behavioral patterns. Using the *surprise* property, this study identifies anomalous MMLU-Redux questions where weaker models answer correctly but stronger models fail, a pattern invisible to accuracy alone. To distinguish genuine competence from stochastic guessing, the corresponding item–model pairs are evaluated under three conditions: deterministic decoding ($T=0.0$), stochastic sampling ($T=0.6$), and $T=0.0$ with an explicit "do not guess" instruction. The results reveal two modes: some models remain consistently correct, indicating stable competence, whereas others fluctuate or collapse under the anti-guessing instruction, suggesting guessing-like behavior. This case study shows how Probing Memes isolates fine-grained behavioral irregularities for targeted diagnosis and reliability analysis.

# 6 LIMITATIONS

This paper proposes an innovative and effective evaluation paradigm. However, it still has limitations. First, the selected datasets do not comprehensively cover task types such as coding, retrieval-augmented generation (RAG), and agent workflows. Moreover, although the six properties help characterize phemotypes, more revealing property designs may exist that detect a wider range of memes. Finally, due to cost constraints, each question is queried only once per model; even with temperature set to 0 for non-reasoning models, full reproducibility is not guaranteed.

# 7 CONCLUSION

This paper reveals that the evaluation of large models is essentially an entangled world between data and models. To better explore the diverse characteristics of large models, the paper introduces the Probing Memes paradigm. It conceptualizes LLMs, drawing on memetics, as collections of invisible memes. Through interactions between data and models, calibrated probe properties detect these memes and infer each model's phemotype, thereby revealing hidden behavioral traits. Evidence comes first from 28 models tested under three reasoning modes across three datasets, revealing the diversity of models and probes that is obscured by traditional evaluation paradigms. The framework is then applied to the larger entangled world of the Open LLM Leaderboard, demonstrating behavioral similarities and divergences among thousands of models. The Probing Memes paradigm offers a scalable, extensible way to evaluate LLMs: calibrated probes yield interpretable phemotype profiles, enable cross-model comparisons, and expose failure modes that accuracy alone obscures.

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

## A EXTENDED RELATED WORK

### A.1 RESULT-DRIVEN IRT ANALYSIS

Item Response Theory (IRT) (Baker & Kim, 2004) estimates a respondent's latent ability together with item difficulty and discrimination from observed responses. By migrating IRT from psychology and education, Kipnis et al. (2025) and Polo et al. (2024) fit latent model ability alongside item difficulty and discrimination, then select high-information items and enable adaptive testing; as a result, they produce compact subsets that preserve full-benchmark scores while substantially reducing evaluation cost.

Furthermore, Schilling-Wilhelmi et al. (2025) employs a Bayesian two-parameter IRT model that yields calibrated ability estimates with uncertainty, revealing how model rankings shift when viewed through the lens of IRT. However, these attempts do not adequately capture the heterogeneity across items or the behavioral similarities and differences across models.

### A.2 CORRELATED ERRORS IN LLMS

Prior studies have documented that large language models do not fail independently: incorrect model outputs are highly correlated across models, and often structured enough to reveal model families and shared failure modes. Bradley (2024) revealed that model errors are strongly correlated, manifested as high agreement on incorrect options in multiple-choice questions. They introduced a model classification approach based on error correlations, employing z-scores and hierarchical clustering to uncover model families. Goel et al. (2025) found that as the capabilities of LLMs increase, errors across models become more correlated, leading to a higher tendency to select the same incorrect options, and revealed that models with similar overall accuracy can nevertheless make very different mistakes. Kim et al. (2025) conducted a study across hundreds of models and multiple benchmarks. They found that models sharing a developer, base architecture, and comparable size consistently exhibit higher agreement rates. While these studies indicate the presence of inherent similarities among different models, they do not provide a quantitative analysis of these attributes. Therefore, they fail to provide a mechanistic explanation of this similarity. Compared with these works, the Probing Memes paradigm goes a step further: it first evaluates items via probe properties and then induces capability-oriented characterizations of models, moving beyond merely showing that models err on different items to explaining how they diverge along specific capability dimensions.

### A.3 MEMETICS AND ITS APPLICATIONS TO LLMS

Dawkins first proposed the concept of *memes* (Dawkins, 1976), drawing an analogy to genes in cultural transmission. Building on this analogy, memetics introduced memotype and phemotype, paralleling genotype and phenotype (Grant et al., 1990; Blackmore, 2000; Álvarez, 2005; Fomin, 2019). In memetics, the *memotype* refers to the actual information content of a meme, while the *phemotype* denotes its concrete manifestation as produced by the memotype under specific conditions. Within LLM research, memetics has been applied to model ideological propagation (Farlow et al., 2024) and to explain reasoning behaviors (Birchall, 2025). These works fail to show how different memes shape similarities and differences across model populations.

**Adapting Memetic Concepts to LLM Evaluation**    Table 2 summarizes the comparison between classical memetic concepts and their counterparts in the probing memes paradigm.

Table 2: **Conceptual correspondence between classical memetics and the probing memes paradigm.**

|  | **Memetics** | **The Probing Memes Paradigm** |
|---|---|---|
| **Meme** | Basic unit of cultural transmission: a reproducible pattern (e.g., a tune, idea, catchphrase, or way of acting) that spreads between individuals by imitation. | Latent unit of model capability: a reusable behavioural tendency (e.g., output style or recurring error pattern) that is shared across models and inferred indirectly from their responses to probes. |
| **Phemotype** | Observable expression of memes in given conditions: concrete behaviours or artifacts that appear when people act under the influence of particular memes. | Observable expression of a model's memes: behavioural tendencies that can be measured directly from its responses to a specific set of probes and characterised using probe-level properties. |

## A.4    DESIGN AND ANALYSIS OF EVALUATION DATASETS AND BENCHMARKS

Existing research has recognized the importance of characterizing the properties of questions in evaluation datasets. Prabhu et al. (2024) ranked questions by difficulty and sampled them uniformly across the difficulty spectrum to test LLMs. This approach leads to comprehensive coverage of the full range of difficulty levels and improves representativeness. Wu & Lo (2025) stratified questions into easy and hard groups. They observed distinct scaling patterns across the two groups during model scaling. This finding helps explain emergent abilities in LLMs. Ghosh et al. (2025) treat datasets from diverse tasks and evaluation metrics as a single large pool, from which users can query a model's performance on specific tasks (e.g., abstract algebra). The Probing Memes paradigm uses an extensible set of probe properties and does not define model capabilities only with respect to particular tasks (as in Ghosh et al. (2025)); instead, it evaluates intrinsic model characteristics and abilities, such as coping with high-risk and high-surprise phenomena.

## A.5    MULTIDIMENSIONAL CAPABILITIES OF LLMS

The sole reliance on accuracy as a metric provides an incomplete assessment of LLMs. Prior research has explored methods to evaluate the multidimensional capabilities of LLMs. For instance, Alyahya et al. (2025) employs adversarial games where LLMs compete as players. Different games correspond to different capabilities, allowing models to be ranked according to specific competencies. However, their results depend on relative rankings rather than quantitative measurements and focus primarily on performance within a fixed interaction scenario (game-based matchups). In contrast, the Probing Memes paradigm quantifies LLM capabilities on existing benchmarks using multi-dimensional, interpretable metrics, enabling fine-grained analysis of capabilities via their phemotype representations; its probe properties can be extended or tailored to different evaluation goals and task domains.

## B    MEME, PROPERTY, AND PHEMOTYPE SPACES (EXTENDED DISCUSSION)

**Meme space.**    Assume a large meme space

$$\mathcal{V} = \{\mu_1, \mu_2, \ldots, \mu_R\},$$

where each $\mu_r$ denotes a possible elementary meme. For each model $M_j$, there is

$$\mathcal{V}_j \subseteq \mathcal{V}$$

for the set of memes carried by $M_j$. In subsection 2.1, this space is applied to give a memetic meaning of the perception matrix: the observed probe-level behavior arises from latent meme subsets interacting with the finite set of probes in $\mathcal{D}$.

**Property space.** In subsection 2.2, probe properties are defined via a property space $\mathcal{A} \subset \mathbb{R}^K$, and each probe $i$ is mapped to a property vector $\boldsymbol{a}_i = (a_i^{(1)}, a_i^{(2)}, \ldots, a_i^{(K)}) \in \mathcal{A}$, where coordinate $a_i^{(k)}$ represents one probe-side property dimension. This paper also distinguishes the corresponding collection of property dimensions

$$A = \{a^{(1)}, a^{(2)}, \ldots, a^{(K)}\},$$

where $a^{(k)}$ denotes the $k$-th probe-side property dimension. In this work, $K$ is instantiated as 6, with dimensions corresponding to six well-designed properties: difficulty, risk, surprise, uniqueness, typicality, and bridge. These property dimensions can be extended or refined to support richer probing objectives.

**Phemotype space and latent mapping.** Let $\mathcal{H}$ denote a phemotype space whose elements are low-dimensional descriptors of a model's underlying memes. Conceptually, there is a mapping

$$\Phi : 2^{\mathcal{V}} \to \mathcal{H},$$

which assigns to each meme subset $\mathcal{V}' \subseteq \mathcal{V}$ a phemotype $\Phi(\mathcal{V}') \in \mathcal{H}$. In particular, each model $M_j$ has an associated meme subset $\mathcal{V}_j$ and a corresponding latent phemotype $\Phi(\mathcal{V}_j)$. The Probing Memes paradigm does not attempt to recover $\mathcal{V}_j$ or $\Phi$ explicitly; instead, it operationalizes phemotypes as elements of $\mathcal{H}$ computed from observable probe properties.

**Measurable Phemotypes from Property Subsets.** Given the collection of property dimensions $A$ and a non-empty subset $S \subseteq A$, a combination operator $f$ maps the properties in $S$ to a phemotype

$$f(S) \in \mathcal{H},$$

which can be viewed as a design choice for aggregating probe-side information into a model-level descriptor. Let $d = |S|$ be the number of property dimensions in the subset. Fixing a particular combination operator $f$, the set of all phemotypes built from $d$-property subsets is

$$\mathcal{H}_d = \{ f(S) \mid S \subseteq A, |S| = d \}, \qquad d = 1, 2, \ldots, K - 1.$$

The overall phemotype space induced by this construction is

$$\mathcal{H} = \bigcup_{d=1}^{K-1} \mathcal{H}_d, \qquad |\mathcal{H}_d| = \binom{K}{d}.$$

Different choices of $S$ and $f$ correspond to different phemotypes. The five concrete phemotypes used in subsection 2.3 (Vigilance, Mastery, Transfer, Ingenuity, and Astuteness) arise from particular subsets of the six probe properties together with specific multiplicative forms of $f$ over the normalized properties; Table 1 summarizes these choices. This construction highlights that the set of phemotypes is inherently extensible: new evaluation needs or meme-focused analyses can be accommodated by introducing alternative subsets $S$ (e.g., properties emphasizing a targeted meme family) and by designing new combination rules $f$ that aggregate the corresponding probe properties into interpretable model-level scores.

## C  Probe Properties

### C.1  Additional Discussion on MPPs

#### C.1.1  Risk

The weighting factor $I_j$ can be viewed as an information weight derived from the overall error rate of model $M_j$. Its form resembles the notion of self-information $-\ln p$, assigning higher values when errors are rarer and lower values when errors are common. In this weighting scheme, models with extremely high error rates yield values close to zero, thereby diminishing their impact on the risk estimate. This prevents low-quality models, which fail almost universally, from artificially inflating co-failure statistics. Conversely, models that are generally accurate but occasionally fail on specific probes receive larger weights, highlighting their role in identifying probes that induce genuinely high-risk failure modes. Thus, weighting by $I_j$ not only incorporates an information-theoretic perspective on model behavior but also mitigates distortions caused by extreme outlier models.

### C.1.2 SURPRISE

**Normalization**  To ensure comparability across models with different overall ability levels, model accuracy $a_j$ is normalized by $z$-score:

$$a_j^z = \frac{a_j - \mu}{\sigma},$$

where $\mu$ and $\sigma$ are the mean and standard deviation of $\{a_j\}_{j=1}^m$. This normalization guarantees that the contribution of each model is measured relative to the population, and it is consistently applied in all computations of Surprise.

**Calculation of the Hard-side Surprise**  For the hard-side case, let

$$R_i = \{j \mid P_{ij} = 1\},$$

that is, $R_i$ is the set of model indices such that $M_j$ succeeds on probe $i$. The hard-side surprise of probe $i$ is then defined as

$$s_i^{\text{hard}} = \left(-\ln(1 - d_i)\right) \cdot \frac{1}{|R_i|} \sum_{j \in R_i} (1 - a_j^z),$$

where $d_i$ is the difficulty of probe $i$. This formulation mirrors the easy-side case: while $s_i^{\text{easy}}$ emphasizes probes that are generally easy yet unexpectedly failed by stronger models, $s_i^{\text{hard}}$ emphasizes probes that are generally difficult yet unexpectedly solved by weaker models.

### C.1.3 UNIQUENESS

For each probe $i$, the perception span $(P_{i1}, \ldots, P_{im})$ is a binary row vector recording the responses of $m$ models. For the purpose of information-theoretic analysis, probe $i$ is also viewed as a binary random variable $P_i$ over the model population, where $P_i = 1$ indicates a correct response and $P_i = 0$ an incorrect one, and the vector entries $(P_{i1}, \ldots, P_{im})$ are regarded as empirical samples of this variable.

The uniqueness of probe $i$ is defined as

$$u_i = \frac{1}{n-1} \sum_{k \neq i}^{n} H(P_k \mid P_i),$$

where $P_k$ denotes the random variable associated with probe $k$.

The conditional entropy term is expanded as

$$H(P_k \mid P_i) = \Pr(P_i = 1)\, H(P_k \mid P_i = 1) + \Pr(P_i = 0)\, H(P_k \mid P_i = 0),$$

with

$$H(P_k \mid P_i = x) = -p_x \log_2 p_x - (1 - p_x) \log_2(1 - p_x), \quad p_x = \Pr(P_k = 1 \mid P_i = x),$$

where probabilities are estimated empirically from the model population.

Thus, $u_i$ measures the average conditional entropy of other probes given probe $i$, capturing how much information the responses to probe $i$ contribute about the rest of the probes.

### C.1.4  $\phi$-COEFFICIENT

For each probe $i$, $P_i = (P_{i1}, P_{i2}, \ldots, P_{im})$, where $P_{ij} \in \{0, 1\}$ indicates whether probe $i$ is answered incorrectly by model $M_j$. Each $P_{ij}$ can be interpreted as the observed samples of a Bernoulli random variable, whose expectation corresponds to the empirical difficulty $d_i$ of probe $i$.

The similarity between probes $i$ and $k$ is computed using the $\phi$-coefficient, defined as

$$\phi(i, k) = \frac{n_{11} n_{00} - n_{10} n_{01}}{\sqrt{(n_{1\cdot} n_{0\cdot} n_{\cdot 1} n_{\cdot 0})}}, \tag{9}$$

where $n_{\alpha\beta}$ denotes the number of models for which $P_{im} = \alpha$ and $P_{km} = \beta$ with $\alpha, \beta \in \{0, 1\}$. Although the $\phi$-coefficient is formally a correlation measure between two Bernoulli random variables, in this work, it is employed as a similarity score quantifying the extent to which two probes exhibit consistent difficulty patterns across the model population.

## C.2 EXPANDED VISUALIZATIONS

This section presents the visualization results of various property combinations across different datasets.

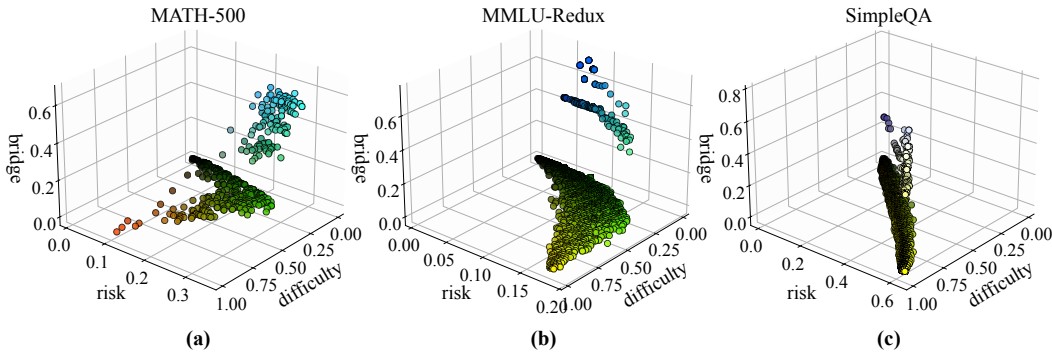

Figure 10: **Probe properties distributions across datasets.** Axes depict difficulty, risk, and bridge.

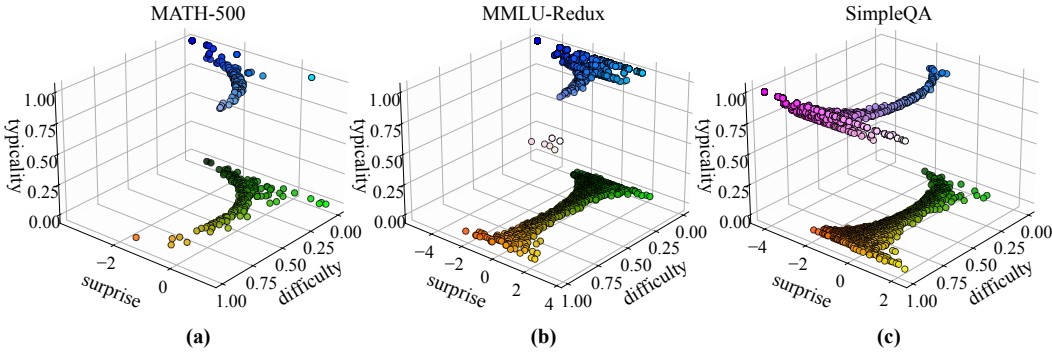

Figure 11: **Probe properties distributions across datasets.** Axes depict difficulty, surprise, and typicality. A typicality value of 0 indicates that the probe does not belong to any cluster during the clustering process.

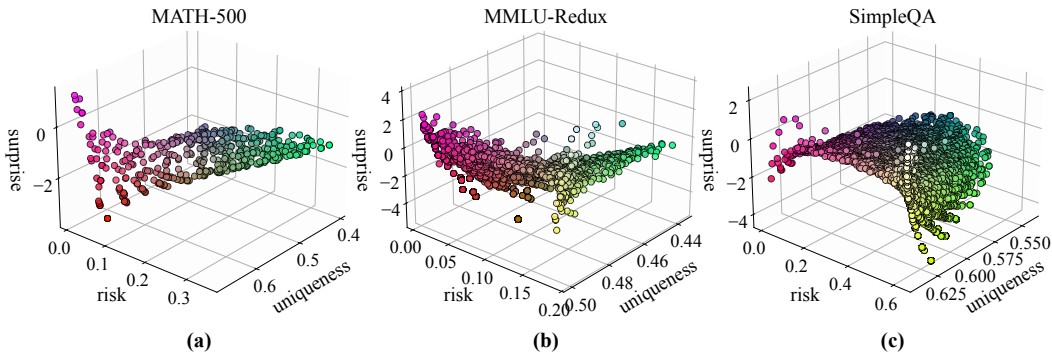

Figure 12: **Probe properties distributions across datasets.** Axes depict uniqueness, risk, and surprise.

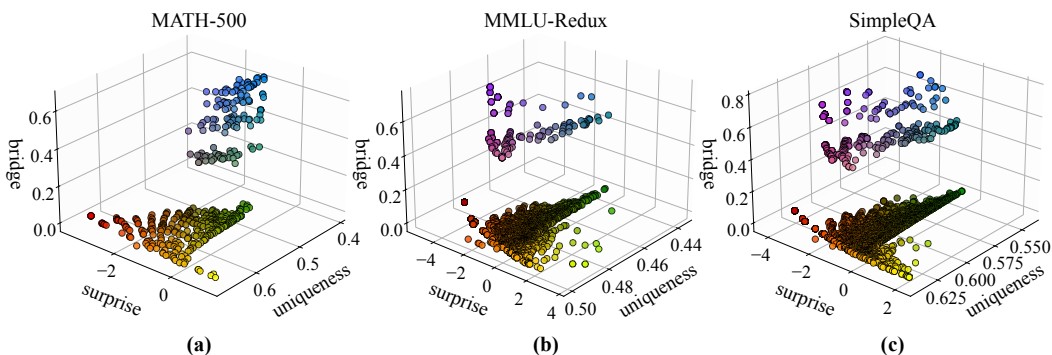

**Figure 13: Probe properties distributions across datasets.** Axes depict uniqueness, surprise, and bridge.

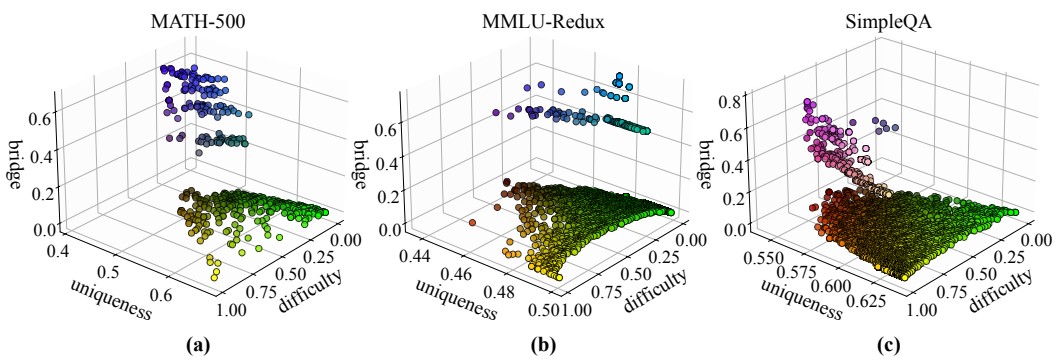

**Figure 14: Probe properties distributions across datasets.** Axes depict difficulty, uniqueness, and bridge.

## D PHEMOTYPES

### D.1 ADDITIONAL DISCUSSION ON PHEMOTYPES

**Notation.** $P \in \{0,1\}^{n \times m}$ is the perception matrix, with entries $P_{ij}$ (probe $i$, model $j$), where $n$ is the number of probes and $m$ is the number of models. Raw Properties of Probes are denoted by $\rho_i \in \mathbb{R}$ (e.g., risk, surprise, difficulty, typicality, bridge, uniqueness). Weights are $w = (w_1, \ldots, w_n)^\top \in \mathbb{R}_{\geq 0}^n$.

#### D.1.1 NORMALIZATION

Given scores $\{\rho_i\}_{i=1}^n$, define the average-tie rank

$$\text{rk}_i = 1 + \sum_{k=1}^n \mathbf{1}\{\rho_k < \rho_i\} + \frac{1}{2}\left(\sum_{k=1}^n \mathbf{1}\{\rho_k = \rho_i\} - 1\right). \tag{10}$$

Normalize ranks to the unit interval

$$\text{frac}_i = \frac{\text{rk}_i - \frac{1}{2}}{n} \in \left(\frac{1}{2n}, 1 - \frac{1}{2n}\right). \tag{11}$$

Here, $n$ denotes the number of probes.

With temperature $\tau > 0$ and output range $[\ell, h]$, define

$$\tilde{\rho}_i \;=\; \mathrm{Norm}(\rho_i) \;=\; \sigma\!\left(\frac{\mathrm{frac}_i - \frac{1}{2}}{\tau}\right), \qquad \hat{\rho}_i \;=\; \ell + (h - \ell)\,\tilde{\rho}_i, \tag{12}$$

where $\sigma(x) = \frac{1}{1+e^{-x}}$.

Shorthand:

$$\begin{aligned}
\tilde{r}_i &= \mathrm{Norm}(\rho_i^{\mathrm{risk}}), & \tilde{s}_i &= \mathrm{Norm}(\rho_i^{\mathrm{surprise}}), \\
\tilde{d}_i &= \mathrm{Norm}(\rho_i^{\mathrm{difficulty}}), & \tilde{t}_i &= \mathrm{Norm}(\rho_i^{\mathrm{typicality}}), \\
\tilde{b}_i &= \mathrm{Norm}(\rho_i^{\mathrm{bridge}}), & \tilde{u}_i &= \mathrm{Norm}(\rho_i^{\mathrm{uniqueness}}).
\end{aligned} \tag{13}$$

### D.1.2 SCORE

For weights $w \geq 0$:

$$\mathrm{Score}(w;\, P_{\cdot j}) \;=\; \frac{\displaystyle\sum_{i=1}^{n} w_i\, P_{ij}}{\displaystyle\sum_{i=1}^{n} w_i}. \tag{14}$$

Range: $\mathrm{Score} \in [0, 1]$.

### D.1.3 CLUSTER SCALE

Let clusters $\{C_c\}_{c=1}^{K}$ partition probes; $c_i$ is probe $i$'s cluster, size $|C_{c_i}|$.

$$scale_i \;=\; \left(|C_{c_i}|\right)^{-\beta}, \qquad \beta \in [0, 1], \tag{15}$$

optionally clipped:

$$scale_i \;\leftarrow\; \min\{\max\{scale_i, \ell\}, h\}, \quad \ell \leq h. \tag{16}$$

This cluster-level scaling term prevents large clusters with many similar or duplicate probes from disproportionately dominating the resulting phenotype scores. Defaults: $\beta = 0.5$, $[\ell, h] = [0, 1]$.

### D.1.4 PHENOTYPE WEIGHTS AND SCORES

$$w_i^{\mathrm{Vig}} = \tilde{r}_i\, \tilde{s}_i, \quad w_i^{\mathrm{Mas}} = \tilde{t}_i\, \tilde{d}_i\, scale_i, \quad w_i^{\mathrm{Trf}} = \tilde{b}_i\, \tilde{d}_i, \quad w_i^{\mathrm{Ing}} = \tilde{u}_i\, \tilde{d}_i, \quad w_i^{\mathrm{Ast}} = \tilde{s}_i. \tag{17}$$

Phenotype score for model $M_j$: $\mathrm{Score}(w^{\cdot};\, P_{\cdot j})$.

## D.2 MORE RESULTS

Figure 15 shows the results of phemotypes compared with the accuracy for each dataset separately. The models are sorted from high to low according to their accuracy. It can be seen that the scores of the five phemotypes do not change synchronously with the accuracy.

Table 3 presents the phemotype benchmarks of all models across three datasets (MMLU-Redux, MATH-500, and SimpleQA), with models sorted in descending order of accuracy. All scores are averaged over the three datasets. PCS refers to the Phenotype Composite Score, which represents the average of the five phemotypes, with scores scaled to 0–100 with one decimal. Column abbreviations are as follows: Acc for Accuracy, PCS for the composite, Vig for Vigilance, Mas for Mastery, Tra for Transfer, Ing for Ingenuity, and Ast for Astuteness. The table enables side-by-side inspection of aggregate accuracy and phemotype dimensions, making it possible to identify cases where accuracy-similar models exhibit divergent phemotype profiles. Table 4 further visualizes the ranks of each phemotype and highlights how these ranks shift relative to accuracy.

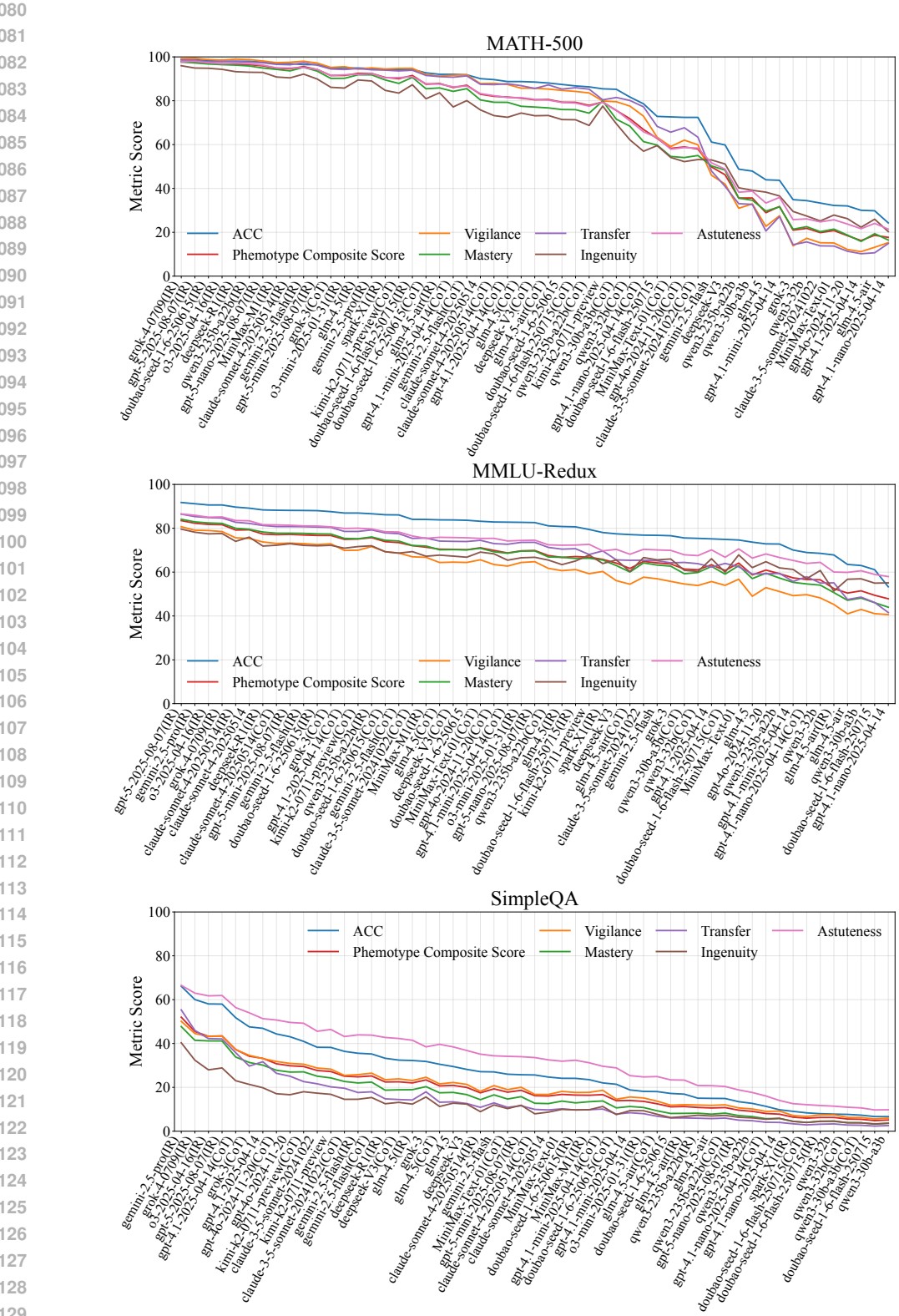

Figure 15: **The scores between accuracy and phemotypes.** This figure shows a line plot of the phemotypes and accuracy rates for all models under different reasoning modes (sorted in descending order of accuracy for each model).

Table 3: **Benchmark scores on phemotypes (scores).** Columns report accuracy (Acc), the Phemotype Composite Score (PCS; the average of the five phemotypes), and the five phemotypes: Vigilance (Vig), Mastery (Mas), Transfer (Tra), Ingenuity (Ing), and Astuteness (Ast).

| ID | Model | Acc | PCS | Vig | Mas | Tra | Ing | Ast |
|---|---|---|---|---|---|---|---|---|
| 1 | gemini-2.5-pro(IR) | 84.0 | 75.6 | 74.7 | 74.1 | 78.3 | 69.2 | 81.6 |
| 2 | grok-4-0709(IR) | 83.1 | 75.0 | 74.1 | 73.8 | 76.4 | 68.6 | 82.0 |
| 3 | gpt-5-2025-08-07(IR) | 82.8 | 74.8 | 74.5 | 74.2 | 75.7 | 67.8 | 82.1 |
| 4 | o3-2025-04-16(IR) | 82.2 | 73.9 | 73.6 | 73.4 | 75.0 | 66.6 | 81.2 |
| 5 | grok-3(CoT) | 76.7 | 67.5 | 67.6 | 66.3 | 68.2 | 59.9 | 75.4 |
| 6 | gpt-4.1-2025-04-14(CoT) | 75.8 | 64.6 | 64.8 | 62.8 | 67.4 | 55.5 | 72.6 |
| 7 | kimi-k2-0711-preview(CoT) | 74.1 | 64.9 | 65.1 | 63.4 | 64.9 | 57.7 | 73.3 |
| 8 | gemini-2.5-flash(IR) | 73.4 | 65.8 | 65.6 | 65.0 | 65.2 | 59.6 | 73.4 |
| 9 | deepseek-R1(IR) | 73.2 | 65.5 | 65.4 | 64.4 | 64.6 | 59.2 | 73.8 |
| 10 | claude-sonnet-4-20250514(IR) | 71.2 | 63.8 | 63.8 | 62.7 | 63.3 | 57.8 | 71.2 |
| 11 | gemini-2.5-flash(CoT) | 71.0 | 61.9 | 62.3 | 60.7 | 62.3 | 54.7 | 69.6 |
| 12 | gpt-5-mini-2025-08-07(IR) | 70.1 | 63.4 | 63.5 | 62.3 | 62.9 | 58.2 | 69.9 |
| 13 | doubao-seed-1-6-250615(IR) | 70.1 | 63.4 | 63.0 | 62.4 | 62.7 | 58.8 | 70.2 |
| 14 | glm-4.5(IR) | 69.3 | 60.4 | 59.4 | 59.2 | 59.8 | 55.1 | 68.6 |
| 15 | deepseek-V3(CoT) | 68.2 | 57.8 | 58.0 | 55.4 | 58.0 | 51.2 | 66.2 |
| 16 | MiniMax-M1(IR) | 68.1 | 60.9 | 60.6 | 59.9 | 60.6 | 55.9 | 67.2 |
| 17 | claude-sonnet-4-20250514 | 68.0 | 59.4 | 60.0 | 57.5 | 59.8 | 53.5 | 66.4 |
| 18 | claude-sonnet-4-20250514(CoT) | 67.8 | 58.4 | 59.3 | 56.6 | 59.4 | 51.1 | 65.8 |
| 19 | glm-4.5(CoT) | 67.7 | 57.4 | 57.2 | 55.1 | 58.1 | 51.1 | 65.6 |
| 20 | kimi-k2-0711-preview | 67.7 | 57.8 | 55.8 | 56.8 | 56.2 | 54.1 | 66.1 |
| 21 | doubao-seed-1-6-250615(CoT) | 66.8 | 58.4 | 58.6 | 56.9 | 59.1 | 52.6 | 65.0 |
| 22 | qwen3-235b-a22b(IR) | 66.6 | 61.0 | 60.8 | 60.0 | 60.9 | 57.4 | 65.8 |
| 23 | gpt-4o-2024-11-20(CoT) | 66.5 | 53.2 | 52.4 | 50.4 | 55.7 | 45.9 | 61.6 |
| 24 | gpt-4.1-mini-2025-04-14(CoT) | 65.6 | 57.1 | 57.7 | 55.6 | 57.9 | 51.3 | 63.4 |
| 25 | o3-mini-2025-01-31(IR) | 65.3 | 58.3 | 58.4 | 56.9 | 58.6 | 53.9 | 63.7 |
| 26 | gpt-5-nano-2025-08-07(IR) | 64.5 | 58.5 | 57.8 | 57.2 | 58.7 | 55.3 | 63.2 |
| 27 | claude-3-5-sonnet-20241022(CoT) | 64.3 | 51.7 | 50.8 | 49.9 | 52.8 | 45.7 | 59.4 |
| 28 | doubao-seed-1-6-250615 | 62.8 | 53.5 | 53.7 | 51.4 | 55.1 | 48.1 | 59.4 |
| 29 | glm-4.5-air(CoT) | 61.1 | 51.6 | 51.2 | 48.7 | 53.1 | 47.1 | 57.8 |
| 30 | doubao-seed-1-6-flash-250715(IR) | 61.0 | 55.0 | 54.4 | 53.8 | 55.9 | 52.3 | 58.3 |
| 31 | qwen3-235b-a22b(CoT) | 60.8 | 52.1 | 52.4 | 49.8 | 54.2 | 47.1 | 56.8 |
| 32 | MiniMax-Text-01(CoT) | 60.6 | 49.1 | 47.9 | 46.7 | 50.3 | 44.5 | 55.8 |
| 33 | spark-X1(IR) | 60.5 | 54.3 | 54.0 | 53.2 | 55.7 | 51.0 | 57.6 |
| 34 | glm-4.5-air(IR) | 58.9 | 50.5 | 49.5 | 48.2 | 50.7 | 47.2 | 57.0 |
| 35 | doubao-seed-1-6-flash-250715(CoT) | 56.7 | 48.7 | 48.3 | 46.3 | 50.9 | 45.1 | 52.6 |
| 36 | qwen3-30b-a3b(CoT) | 56.0 | 47.5 | 46.8 | 44.9 | 49.5 | 44.7 | 51.4 |
| 37 | deepseek-V3 | 55.2 | 43.4 | 39.8 | 42.7 | 39.7 | 43.0 | 52.0 |
| 38 | gemini-2.5-flash | 55.0 | 44.4 | 41.3 | 43.3 | 42.0 | 43.5 | 52.1 |
| 39 | qwen3-32b(CoT) | 54.9 | 46.1 | 45.9 | 44.0 | 48.9 | 42.1 | 49.8 |
| 40 | gpt-4.1-nano-2025-04-14(CoT) | 53.0 | 43.9 | 43.9 | 40.6 | 46.4 | 39.8 | 48.6 |
| 41 | gpt-4.1-2025-04-14 | 50.7 | 37.6 | 33.3 | 36.2 | 34.7 | 36.1 | 47.7 |
| 42 | claude-3-5-sonnet-20241022 | 49.5 | 37.5 | 33.9 | 36.5 | 33.6 | 36.4 | 46.9 |
| 43 | gpt-4o-2024-11-20 | 49.3 | 36.4 | 32.0 | 35.1 | 31.9 | 35.9 | 47.2 |
| 44 | glm-4.5 | 49.0 | 36.2 | 31.3 | 34.8 | 31.0 | 37.7 | 46.1 |
| 45 | grok-3 | 47.8 | 36.0 | 31.4 | 34.8 | 32.2 | 37.0 | 44.7 |
| 46 | doubao-seed-1-6-flash-250715 | 46.9 | 39.0 | 36.7 | 36.3 | 38.9 | 39.3 | 43.9 |
| 47 | qwen3-235b-a22b | 44.8 | 34.7 | 30.7 | 33.2 | 32.4 | 36.1 | 40.9 |
| 48 | gpt-4.1-mini-2025-04-14 | 44.2 | 34.4 | 30.8 | 32.7 | 30.4 | 35.7 | 42.2 |
| 49 | MiniMax-Text-01 | 43.7 | 33.9 | 30.0 | 32.7 | 28.8 | 35.2 | 42.7 |
| 50 | qwen3-30b-a3b | 39.2 | 30.7 | 27.3 | 28.8 | 28.0 | 33.2 | 36.4 |
| 51 | qwen3-32b | 36.9 | 28.2 | 24.3 | 27.0 | 24.7 | 30.9 | 34.0 |
| 52 | glm-4.5-air | 36.1 | 26.6 | 22.0 | 24.8 | 21.3 | 29.9 | 34.9 |
| 53 | gpt-4.1-nano-2025-04-14 | 29.1 | 24.4 | 21.7 | 22.1 | 20.1 | 27.1 | 31.1 |

Table 4: **Benchmark scores on phemotypes (ranks).** Blue indicates rank improvement, and red indicates rank degradation. Columns: the Phemotype Composite Score (PCS; the average of the five phemotypes), and the five phemotypes: Vigilance (Vig), Mastery (Mas), Transfer (Tra), Ingenuity (Ing), and Astuteness (Ast).

| ID | Model | PCS | Vig | Mas | Tra | Ing | Ast |
|----|-------|-----|-----|-----|-----|-----|-----|
| 1 | gemini-2.5-pro(IR) | 1 | 1 | 2 (-1) | 1 | 1 | 3 (-2) |
| 2 | grok-4-0709(IR) | 2 | 3 (-1) | 3 (-1) | 2 | 2 | 2 |
| 3 | gpt-5-2025-08-07(IR) | 3 | 2 (+1) | 1 (+2) | 3 | 3 | 1 (+2) |
| 4 | o3-2025-04-16(IR) | 4 | 4 | 4 | 4 | 4 | 4 |
| 5 | grok-3(CoT) | 5 | 5 | 5 | 5 | 5 | 5 |
| 6 | gpt-4.1-2025-04-14(CoT) | 9 (-3) | 9 (-3) | 9 (-3) | 6 | 14 (-8) | 9 (-3) |
| 7 | kimi-k2-0711-preview(CoT) | 8 (-1) | 8 (-1) | 8 (-1) | 8 (-1) | 11 (-4) | 8 (-1) |
| 8 | gemini-2.5-flash(IR) | 6 (+2) | 6 (+2) | 6 (+2) | 7 (+1) | 6 (+2) | 7 (+1) |
| 9 | deepseek-R1(IR) | 7 (+2) | 7 (+2) | 7 (+2) | 9 | 7 (+2) | 6 (+3) |
| 10 | claude-sonnet-4-20250514(IR) | 10 | 10 | 10 | 10 | 10 | 10 |
| 11 | gemini-2.5-flash(CoT) | 13 (-2) | 13 (-2) | 13 (-2) | 13 (-2) | 17 (-6) | 13 (-2) |
| 12 | gpt-5-mini-2025-08-07(IR) | 11 (+1) | 11 (+1) | 12 | 11 (+1) | 9 (+3) | 12 |
| 13 | doubao-seed-1-6-250615(IR) | 11 (+1) | 12 | 11 (+1) | 12 | 8 (+4) | 11 (+1) |
| 14 | glm-4.5(IR) | 16 (-2) | 17 (-3) | 16 (-2) | 16 (-2) | 16 (-2) | 14 |
| 15 | deepseek-V3(CoT) | 22 (-7) | 21 (-6) | 24 (-9) | 23 (-8) | 24 (-9) | 17 (-2) |
| 16 | MiniMax-M1(IR) | 15 (+1) | 15 (+1) | 15 (+1) | 15 (+1) | 13 (+3) | 15 (+1) |
| 17 | claude-sonnet-4-20250514 | 17 | 16 (+1) | 17 | 16 (+1) | 20 (-3) | 16 (+1) |
| 18 | claude-sonnet-4-20250514(CoT) | 19 (-1) | 18 | 22 (-4) | 18 | 25 (-7) | 19 (-1) |
| 19 | glm-4.5(CoT) | 24 (-5) | 24 (-5) | 25 (-6) | 22 (-3) | 25 (-6) | 21 (-2) |
| 20 | kimi-k2-0711-preview | 22 (-3) | 25 (-6) | 21 (-2) | 25 (-6) | 18 (+1) | 18 (+1) |
| 21 | doubao-seed-1-6-250615(CoT) | 19 (+2) | 19 (+2) | 19 (+2) | 19 (+2) | 21 | 22 (-1) |
| 22 | qwen3-235b-a22b(IR) | 14 (+8) | 14 (+8) | 14 (+8) | 14 (+8) | 12 (+10) | 19 (+3) |
| 23 | gpt-4o-2024-11-20(CoT) | 29 (-6) | 29 (-6) | 29 (-6) | 27 (-4) | 32 (-9) | 26 (-3) |
| 24 | gpt-4.1-mini-2025-04-14(CoT) | 25 (-1) | 23 (+1) | 23 (+1) | 24 | 23 (+1) | 24 |
| 25 | o3-mini-2025-01-31(IR) | 21 (+4) | 20 (+5) | 19 (+6) | 21 (+4) | 19 (+6) | 23 (+2) |
| 26 | gpt-5-nano-2025-08-07(IR) | 18 (+8) | 22 (+4) | 18 (+8) | 20 (+6) | 15 (+11) | 25 (+1) |
| 27 | claude-3-5-sonnet-20241022(CoT) | 31 (-4) | 32 (-5) | 30 (-3) | 32 (-5) | 33 (-6) | 27 |
| 28 | doubao-seed-1-6-250615 | 28 | 28 | 28 | 29 (-1) | 28 | 27 (+1) |
| 29 | glm-4.5-air(CoT) | 32 (-3) | 31 (-2) | 32 (-3) | 31 (-2) | 30 (-1) | 30 (-1) |
| 30 | doubao-seed-1-6-flash-250715(IR) | 26 (+4) | 26 (+4) | 26 (+4) | 26 (+4) | 22 (+8) | 29 (+1) |
| 31 | qwen3-235b-a22b(CoT) | 30 (+1) | 29 (+2) | 31 | 30 (+1) | 30 (+1) | 33 (-2) |
| 32 | MiniMax-Text-01(CoT) | 34 (-2) | 35 (-3) | 34 (-2) | 35 (-3) | 36 (-4) | 34 (-2) |
| 33 | spark-X1(IR) | 27 (+6) | 27 (+6) | 27 (+6) | 27 (+6) | 27 (+6) | 31 (+2) |
| 34 | glm-4.5-air(IR) | 33 (+1) | 33 (+1) | 33 (+1) | 34 | 29 (+5) | 32 (+2) |
| 35 | doubao-seed-1-6-flash-250715(CoT) | 35 | 34 (+1) | 35 | 33 (+2) | 34 (+1) | 35 |
| 36 | qwen3-30b-a3b(CoT) | 36 | 36 | 36 | 36 | 35 (+1) | 38 (-2) |
| 37 | deepseek-V3 | 40 (-3) | 40 (-3) | 39 (-2) | 40 (-3) | 38 (-1) | 37 |
| 38 | gemini-2.5-flash | 38 | 39 (-1) | 38 | 39 (-1) | 37 (+1) | 36 (+2) |
| 39 | qwen3-32b(CoT) | 37 (+2) | 37 (+2) | 37 (+2) | 37 (+2) | 39 | 39 |
| 40 | gpt-4.1-nano-2025-04-14(CoT) | 39 (+1) | 38 (+2) | 40 | 38 (+2) | 40 | 40 |
| 41 | gpt-4.1-2025-04-14 | 42 (-1) | 43 (-2) | 43 (-2) | 42 (-1) | 45 (-4) | 41 |
| 42 | claude-3-5-sonnet-20241022 | 43 (-1) | 42 | 41 (+1) | 43 (-1) | 44 (-2) | 43 (-1) |
| 43 | gpt-4o-2024-11-20 | 44 (-1) | 44 (-1) | 44 (-1) | 46 (-3) | 47 (-4) | 42 (+1) |
| 44 | glm-4.5 | 45 (-1) | 46 (-2) | 45 (-1) | 47 (-3) | 42 (+2) | 44 |
| 45 | grok-3 | 46 (-1) | 45 | 45 | 45 | 43 (+2) | 45 |
| 46 | doubao-seed-1-6-flash-250715 | 41 (+5) | 41 (+5) | 42 (+4) | 41 (+5) | 41 (+5) | 46 |
| 47 | qwen3-235b-a22b | 47 | 48 (-1) | 47 | 44 (+3) | 45 (+2) | 49 (-2) |
| 48 | gpt-4.1-mini-2025-04-14 | 48 | 47 (+1) | 48 | 48 | 48 | 48 |
| 49 | MiniMax-Text-01 | 49 | 49 | 48 (+1) | 49 | 49 | 47 (+2) |
| 50 | qwen3-30b-a3b | 50 | 50 | 50 | 50 | 50 | 50 |
| 51 | qwen3-32b | 51 | 51 | 51 | 51 | 51 | 52 (-1) |
| 52 | glm-4.5-air | 52 | 52 | 52 | 52 | 52 | 51 (+1) |
| 53 | gpt-4.1-nano-2025-04-14 | 53 | 53 | 53 | 53 | 53 | 53 |

# E  APPLICATIONS AND CASE STUDIES

## E.1  MODEL SELECTION

This subsection examines whether phemotypes can guide task-aware model selection on the MATH benchmark (Hendrycks et al., 2021) by pairing models with similar overall accuracy on MATH-500 but different difficulty phemotype scores (i.e., phemotypes based solely on the difficulty property), and then routing MATH-FULL items between them based on difficulty levels. Results indicate that, for a fixed total number of items, routing items of different difficulty to phemotype-specific models can improve overall accuracy.

**Setup.** Following the method in Section 2.3, the difficulty property is converted into a one-dimensional phemotype. Two model pairs are first selected on MATH-500. Within each pair, the two models exhibit nearly identical overall accuracy on MATH-500 but differ in their difficulty phemotype scores. Table 5 reports the accuracies and difficulty scores on MATH-500. In Pair 1, `doubao-seed-1-6-flash-250715` has slightly higher accuracy (+0.22) and a clearly higher difficulty score (+4.83) than `MiniMax-Text-01(CoT)`. In Pair 2, `glm-4.5` has slightly higher accuracy (+0.22), while `gpt-4.1-mini-2025-04-14` has the higher difficulty score (+3.02).

Table 5: **Calibration on MATH-500 for difficulty-aware model selection.** For each pair we select two models with similar overall accuracy but different difficulty phemotype. "Acc" is the average accuracy on MATH-500, "Diff." is the difficulty phemotype score.

| Model | Acc | Diff. |
|---|---|---|
| *Pair 1 (strong models, hi-diff. vs. low-diff.)* | | |
| doubao-seed-1-6-flash-250715 | 72.85 | 62.44 |
| MiniMax-Text-01(CoT) | 72.63 | 57.61 |
| *Pair 2 (weaker models, hi-diff. vs. low-diff.)* | | |
| glm-4.5 | 43.93 | 26.51 |
| gpt-4.1-mini-2025-04-14 | 43.71 | 29.53 |

**Phemotype-based routing and baseline.** The same pairs are then evaluated on the full test set of the MATH benchmark (MATH-FULL, excluding MATH-500). For each pair, the analysis considers the 4,500 items answered by both models, with 2,276 items labeled as Level 4 or 5 and 2,224 items labeled as Level 1, 2, or 3. The model with the higher difficulty phemotype is treated as $M^{\text{hi}}$ (high-difficulty model), and the other as $M^{\text{lo}}$ (low-difficulty model). A simple *phemotype-based routing* policy is then applied: items with level 4 or 5 are routed to $M^{\text{hi}}$ (hard subset), and items with level 1, 2, or 3 are routed to $M^{\text{lo}}$ (easy subset). To control for the effect of simply combining two models, a *random per-level balanced partition* is used as a baseline: for each difficulty level, items are randomly split into two equal halves, one assigned to $M^{\text{hi}}$ and the other to $M^{\text{lo}}$, and this procedure is repeated 10 times with different random seeds (from 0 to 9).

**Results.** Table 6 reports the accuracies on MATH-FULL. For Pair 1 (`doubao` vs. `MiniMax`), phemotype-based routing reaches a union accuracy of 77.02%, compared with $73.89\% \pm 0.29$ for the balanced partition baseline, an improvement of about 3.1 percentage points, and it also exceeds both the hi-diff model's overall accuracy on MATH-FULL (72.24%) and the low-diff model's overall accuracy (76.02%). For Pair 2 (`gpt-4.1-mini` vs. `glm-4.5`), phemotype-based routing attains 48.47% union accuracy, whereas the baseline achieves $45.73\% \pm 0.32$, an improvement of about 2.7 points, and again outperforms using either model alone (46.78% for the hi-diff model and 44.80% for the low-diff model). In both pairs, the hi-diff model performs better on hard items and the low-diff model performs better on easy items, indicating that the difficulty phemotype captures useful specialization patterns that can be exploited by a simple routing policy.

Overall, this application suggests that difficulty phemotypes calibrated on a relatively small set such as MATH-500 can be used to design simple and interpretable routing policies that generalize to a much larger pool of items in the same task, and that such policies provide consistent gains over a random balanced partition baseline without any modification of the underlying models. Moreover, phemotype-based routing can help multi-agent tasks achieve higher accuracy while keeping the total number of queried items the same.

Table 6: **Task-aware model selection on MATH-FULL (difficulty-based routing).** For each pair, we select a high-difficulty (hi-diff) and a low-difficulty (low-diff) model based on the difficulty phemotype calibrated on MATH-500. Under phemotype routing, Level 4–5 problems are routed to the hi-diff model ("Hard") and Level 1–3 problems to the low-diff model ("Easy"). The random baseline assigns, for each level, half of the questions to each model (per-level 50/50 split), repeated 10 times (seeds 0–9); we report mean ± std.

| *Pair 1: doubao-seed-1-6-flash-250715 (hi-diff) vs MiniMax-Text-01(CoT) (low-diff)* | | | |
|---|---|---|---|
| Metric | Phemotype routing | Balanced Partition | Without routing |
| hi-diff model: accuracy on Hard (L4–5) | 66.21 | 71.96 (±0.46) | – |
| low-diff model: accuracy on Easy (L1–3) | 88.08 | 75.81 (±0.60) | – |
| Union accuracy over all questions | **77.02** | 73.89 (±0.29) | – |
| hi-diff model: accuracy on MATH-FULL | – | – | 72.24 |
| low-diff model: accuracy on MATH-FULL | – | – | 76.02 |

| *Pair 2: gpt-4.1-mini-2025-04-14 (hi-diff) vs glm-4.5 (low-diff)* | | | |
|---|---|---|---|
| Metric | Phemotype routing | Balanced Partition | Without routing |
| hi-diff model: accuracy on Hard (L4–5) | 37.08 | 46.72 (±0.55) | – |
| low-diff model: accuracy on Easy (L1–3) | 60.12 | 44.74 (±0.34) | – |
| Union accuracy over all questions | **48.47** | 45.73 (±0.32) | – |
| hi-diff model: accuracy on MATH-FULL | – | – | 46.78 |
| low-diff model: accuracy on MATH-FULL | – | – | 44.80 |

## E.2 SUBSET CONSTRUCTION

The Probing Memes paradigm can also support subset construction by selecting items according to probe properties. This section explores dataset reduction under the Probing Memes paradigm with the goal of curating smaller, cost-effective evaluation subsets. The study assesses whether this method can deliver results comparable to those of two established IRT-based baselines, **MetaBench** (Kipnis et al., 2025) and **TinyBench** (Polo et al., 2024).

### E.2.1 EXPERIMENTAL SETUP

**MetaBench Baseline.** This baseline follows the MetaBench framework and fits an IRT model separately for each dataset. For IFEval, BBH, GPQA-Diamond, and MUSR, it uses a three-parameter logistic (3PL) method, and for MATH and MMLU-Pro, it uses a two-parameter logistic (2PL) method. Item parameters are estimated with marginal maximum likelihood estimation. The subset objective includes a regularization factor $\lambda$ that multiplies the subset size $k$, so a larger $\lambda$ prefers smaller subsets. Regularization factors are set to 0.001 for IFEval, BBH, and MUSR, 0.01 for MATH, and 0.005 for MMLU-Pro and GPQA-Diamond.

**TinyBench Baseline.** This baseline follows the TinyBench setting and uses a two-parameter logistic (2PL) IRT model for all datasets. Item and ability parameters are estimated with Bayesian methods based on variational inference. Priors are placed on the model ability $\theta$, the item discrimination $\alpha$, and the item difficulty $\beta$, and each prior has its own hyperparameters, which in turn follow hyperprior distributions. This hierarchical design allows the Bayesian estimator to share statistical strength across items and models. The test model set is fixed to the same 396 models as in MetaBench, and all remaining models are used for training without a separate validation split.

**Probing Memes Method.** The Probing Memes method selects evaluation items via Algorithm 1, which ranks probes by their typicality property measuring how representative each probe is within its cluster. After the subset is selected, probe properties are computed using the models in the training set, and the models in the test set are then evaluated on the selected subset. Subset accuracy on the selected items is used to derive model rankings, which are compared with the full dataset rankings via Spearman rank correlation.

**Configuration A: MetaBench vs. Probing Memes.** Configuration A compares MetaBench and the Probing Memes method. For all datasets, the training set contains 4,083 models and the test set contains 396 models. This split mirrors the original MetaBench protocol and is shared by both methods. For each dataset, both methods construct subsets of the same size. The subset sizes are 137, 196, 106, 152, 100, and 45 for MATH, IFEval, MMLU-Pro, BBH, MUSR, and GPQA-Diamond, respectively. As the comparison metric, Spearman rank correlation is computed between model accuracies on the subset and model accuracies on the full dataset, evaluated over the test models.

**Configuration B: TinyBench vs. Probing Memes.** Configuration B compares TinyBench and the Probing Memes method. Training and test model sets are the same as in Configuration A: 4,083 models for training and 396 models for testing. This split mirrors the MetaBench protocol and is used by TinyBench and by the Probing Memes method. For each dataset, both methods select subsets of size 100. The comparison metric is the average estimation error, which is identical to that of Mean Absolute Error (MAE), between the subset accuracy and the full-dataset accuracy of each test model, averaged over the test set models.

---

**Algorithm 1** Typicality Guided Probe Selection

---

**Require:**
    Probe set $D$;
    Each probe $i \in D$ has typicality score $t_i$, perception vector $P_i$, and cluster label $g(i)$;
    Target subset size $k$;
    Initialize selected subset $S$, used signature set $U$, and remaining budget $r$.

1: $S \leftarrow \emptyset$
2: $U \leftarrow \emptyset$
3: $r \leftarrow k$
4: Let $\mathcal{C}$ be the set of clusters.
5: **for** each cluster $c \in \mathcal{C}$ **do**
6:     $C_c \leftarrow \{\, i \in D : g(i) = c \,\}$
7:     $|C_c| \leftarrow$ size of $C_c$
8:     $\sum t(c) \leftarrow \sum_{i \in C_c} t_i$
9: **end for**
10: Sort clusters $\mathcal{C}$ by $(|C_c|$ desc, $\sum t(c)$ desc$)$.
11: **for** each cluster $c \in \mathcal{C}$ **do**
12:     $L_c \leftarrow$ items in $C_c$ sorted by $t_i$ in descending order
13:     $q(c) \leftarrow \min\big(|C_c|,\ \max(1, \lceil \sqrt{|C_c|} \rceil)\big)$
14: **end for**
15: **for** each cluster $c$ in sorted $\mathcal{C}$ **do**
16:     **if** $r = 0$ **then**
17:         **break**
18:     **end if**
19:     count $\leftarrow 0$
20:     **for** each probe $i \in L_c$ **do**
21:         **if** $r = 0$ or count $= q(c)$ **then**
22:             **break**
23:         **end if**
24:         **if** $P_i \notin U$ **then**
25:             $S \leftarrow S \cup \{i\}$
26:             $U \leftarrow U \cup \{P_i\}$
27:             $r \leftarrow r - 1$
28:             count $\leftarrow$ count $+ 1$
29:         **end if**
30:     **end for**
31: **end for**
32: **return** $S$

---

### E.2.2 RESULTS AND ANALYSIS

Table 7 reports the results of comparing the Probing Memes method with two IRT-based baselines. In the comparison against MetaBench, the Probing Memes method shows slightly lower Spearman correlations across datasets. The results on MATH (0.977 vs. 0.988) and IFEval (0.993 vs. 0.996) are close, while larger gaps appear on datasets such as BBH and GPQA-Diamond. In the comparison against TinyBench, the Probing Memes method performs competitively on most datasets. For MMLU-Pro and BBH, the mean absolute errors are higher. For the other datasets, the errors remain close to TinyBench or even lower, such as on MATH (0.026 vs. 0.030) and GPQA-Diamond (0.023 vs. 0.035).

Table 7: **Comparison between Probing Memes and IRT-based baselines on subset construction.** Left block reports Spearman rank correlation ($\rho$) between subset-based and full-dataset accuracies. Right block reports mean absolute error (MAE) between subset accuracy and full-dataset accuracy.

| | | MetaBench vs. Probing Memes ($\rho$) | | | TinyBench vs. Probing Memes (MAE) | |
|---|---|---|---|---|---|---|
| Dataset | $k$ | MetaBench | Probing Memes | $k$ | TinyBench | Probing Memes |
| MATH | 137 | **0.988** | 0.977 (-0.011) | 100 | 0.030 | **0.026** (-0.004) |
| IFEval | 196 | **0.996** | 0.993 (-0.003) | 100 | **0.036** | 0.039 (+0.003) |
| MMLU-Pro | 106 | **0.990** | 0.962 (-0.028) | 100 | **0.031** | 0.343 (+0.312) |
| BBH | 152 | **0.990** | 0.902 (-0.088) | 100 | **0.035** | 0.265 (+0.230) |
| MUSR | 100 | **0.948** | 0.755 (-0.193) | 100 | **0.031** | 0.054 (+0.023) |
| GPQA-Diamond | 45 | **0.884** | 0.793 (-0.091) | 100 | 0.035 | **0.023** (-0.012) |

The two baselines are designed with the explicit objective of maximizing the reconstruction of full-dataset accuracies from a reduced subset. Under this objective, the Probing Memes method achieves performance close to these strong industry baselines in the subset reduction task. While the aim of Probing Memes is to support property-specific subset selection strategies, the experiments in this section use dataset reduction as an example and illustrate the potential of the Probing Memes paradigm for constructing subsets tailored to specific probe properties.

### E.3 CHARACTERIZING MODEL SIMILARITIES AND DIFFERENCES THROUGH PHEMOTYPES

The t-SNE visualization in Section 4 shows that, in experiments of OpenLLM Leaderboard, models sharing the same base model tend to appear close to one another in the embedded space, forming coherent local groups. These clusters were also consistent across datasets, suggesting that phemotypes capture behavioral similarity. To further verify this structure, Figure 16 presents a UMAP projection based on the same five-dimensional phemotype vectors.

To further examine the potential relationships associated with training strategies, this work visualizes the Qwen-0.5B family by coloring models according to their respective strategies. Figure 17(t-SNE) and Figure 18(UMAP) shows that models using the same strategy also appear close to one another in the phemotype space. Building on this characterization of models via the Probing Memes paradigm, researchers can focus on clusters of nearby models in this space to study how shared

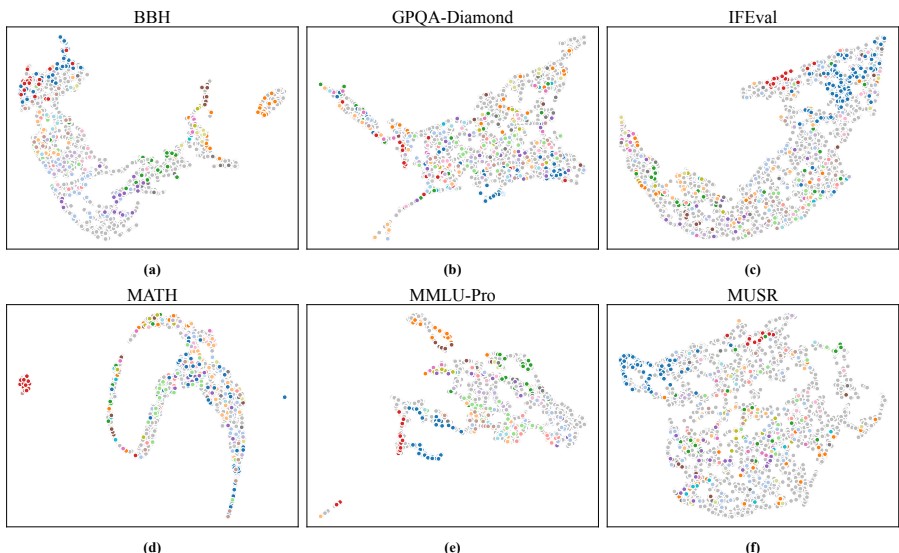

Figure 16: **Commonality and divergence among models revealed by phemotypes (UMAP visualization).**

underlying design choices (e.g., base architectures, training strategies, or data sources) contribute to both common behaviors and systematic differences.

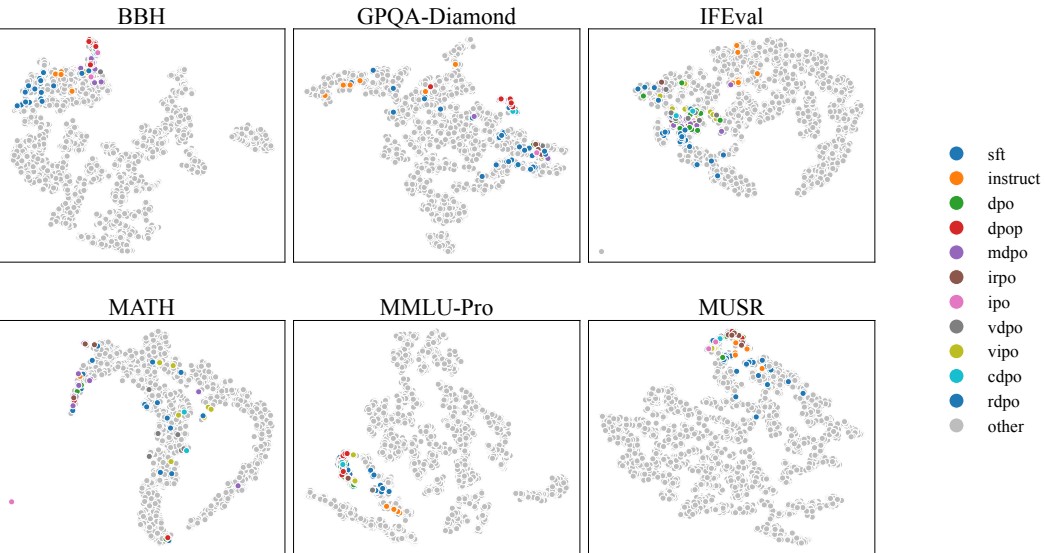

Figure 17: **Qwen-0.5B models colored by training strategies (t-SNE visualization).**

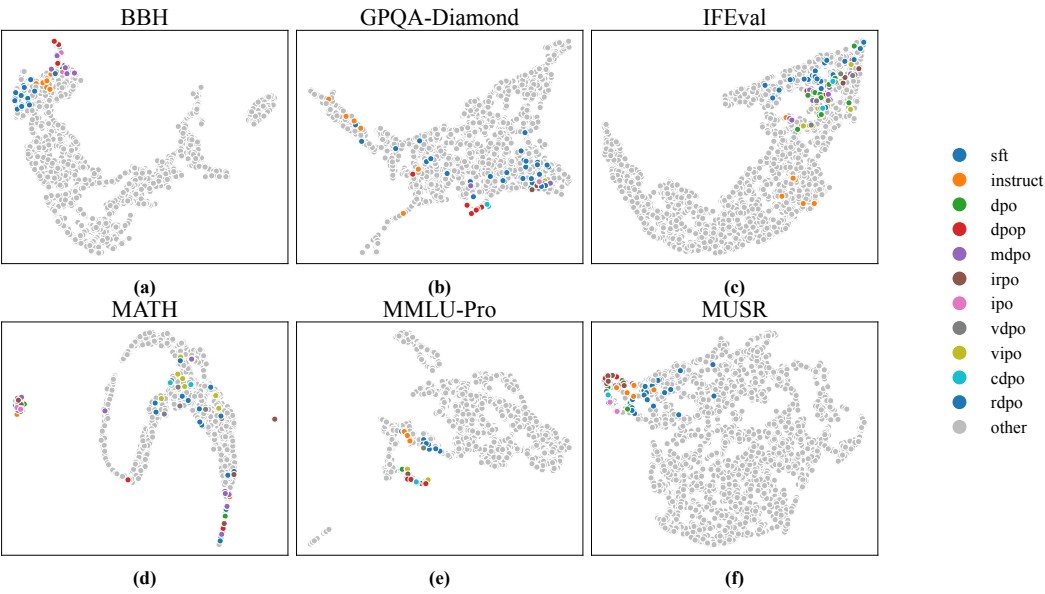

Figure 18: **Qwen-0.5B models colored by training strategies (UMAP visualization).**

### E.4 CASE STUDY: DIAGNOSING HIGH-SURPRISE MCQS BEYOND RANDOM GUESSING

As shown in Figure 5 and Figure 8, there are many high-surprise items in multiple-choice question (MCQ) datasets such as MMLU-Redux and MMLU-Pro. To further explore the underlying causes of such high surprise, in particular whether it may arise from random guessing that allows comparatively weaker models (by overall accuracy) to answer difficult items correctly when stronger models fail, this subsection conducts a small case analysis on six MCQs in MMLU-Redux with the highest surprise values where at least one weaker model answers correctly.

In this analysis, an item-model pair is said to show a *random guessing pattern* when, due to the nature of MCQs, the model fails to maintain consistently high accuracy across temperature and hint conditions and instead stays near chance level or collapses under the explicit "do not guess" hint. For each selected item and each highlighted model, the evaluation issues 20 independent queries under three decoding regimes:

1. **Deterministic decoding** ($T=0.0$)**.** Temperature is set to zero.

2. **Higher-temperature sampling** ($T=0.6$)**.** Temperature is set to 0.6, which encourages more diverse model outputs.

3. **Deterministic decoding with hint** ($T=0.0$ **+ hint**)**.** Temperature is set to zero and the prompt includes an explicit instruction: *"If you do not know the answer or are not confident about which option is correct, do not guess any option. Instead, answer exactly 'I don't know' or 'I'm not sure'."*

For each regime, the protocol records the average correctness over the 20 runs. Table 8 reports the per-probe and per-model accuracies.

Table 8: **Per-probe accuracy (%) under three decoding regimes for the top-6 high-surprise MCQs.** For each item–model pair we run 20 independent repetitions and report the average correctness under: (1) deterministic decoding ($T=0.0$), (2) higher-temperature sampling ($T=0.6$), and (3) deterministic decoding with an explicit hint.

| Item | Model | $T=0.0$ | $T=0.6$ | $T=0.0$ + hint |
|------|-------|---------|---------|----------------|
| Item 1 | doubao-seed-1-6-flash-250715 | 75 | 60 | 25 |
| Item 2 | doubao-seed-1-6-flash-250715 | 100 | 95 | 100 |
| Item 3 | gpt-4.1-nano-2025-04-14 (CoT) | 65 | 65 | 60 |
| | gpt-4.1-nano-2025-04-14 | 10 | 40 | 0 |
| Item 4 | glm-4.5-air | 100 | 55 | 100 |
| Item 5 | glm-4.5-air | 15 | 15 | 0 |
| | gpt-4.1-nano-2025-04-14 (CoT) | 100 | 75 | 95 |
| Item 6 | gpt-4.1-nano-2025-04-14 | 100 | 100 | 100 |
| | glm-4.5-air | 100 | 100 | 100 |

**Findings.** Table 8 reveals three empirical patterns:

1. Across item-model pairs, both deterministic correctness and random guessing patterns can be observed. Some pairs show stable, and deterministic high accuracy rather than random guessing, such as `doubao-seed-1-6-flash-250715` on Item 2 and `glm-4.5-air` on Items 6, which stay close to $100\%$ across all three regimes. Others exhibit large fluctuations that are more compatible with random hits or unstable decisions, for example `glm-4.5-air` on Item 5, which remains at $15\%$ under both $T=0.0$ and $T=0.6$.

2. The same model can exhibit both deterministic correctness and random guessing patterns on different probes. For instance, `gpt-4.1-nano-2025-04-14` on Item 6 achieves $100\%$ accuracy under all three conditions, but on Item 3 the accuracy is only $10\%$ at $T=0.0$, rises to $40\%$ at $T=0.6$, and then falls to $0\%$ with the hint. A higher decoding temperature ($T=0.6$ instead of $T=0.0$) thus increases output uncertainty, sometimes pushing low-accuracy pairs up, as in this Item 3 example.

3. The explicit "do not guess" hint affects model accuracy. In the cases examined here, it reduces accuracy, as with `doubao-seed-1-6-flash-250715` on Item 1 and `glm-4.5-air` on Item 5, where performance drops to $25\%$ and $0\%$, respectively, under the hint.

Taken together, these results indicate that, for the high-surprise MCQs studied here, some correct answers reflect a genuine, deterministic pattern, while others are plausibly driven by random hits.

The same model can exhibit both deterministic correctness and random guessing patterns on different probes, and explicit anti-guessing hints can shift the balance between them. The Probing Memes paradigm and the associated *surprise* property **make such patterns visible at the probe level and support targeted case studies like this one.**

# F   POPULATION-RELATIVE ANALYSES OF PROBE PROPERTIES AND PHEMOTYPES

This section examines two forms of population dependence. First, Appendix F.1 studies the stability of probe properties and model phemotype scores when the number of available models varies. Second, Appendix F.2 analyzes insights that arise when the three reasoning modes are viewed as distinct model populations.

## F.1   ROBUSTNESS OF PROBE PROPERTIES AND LLM PHEMOTYPES TO POPULATION SIZE

This experiment evaluates how stable and sensitive both probe properties and phemotypes of models are to the size and composition of the model population used to estimate them.

**Setup.** The subsampling analyses use the same model population described in Section 3.1. Let $\mathcal{M}$ denote the set of all model–reasoning-mode configurations evaluated in this work. Accounting for model availability yields a total of $|\mathcal{M}| = 53$ models, each corresponding to a specific base model and reasoning mode.

To study robustness to population size, subsample sizes $K \in \{5, 10, 20, 30, 40, 50\}$ are considered. For each $K$, the subsampling procedure is repeated 10 times: in each repeat, a subset of $K$ models is drawn from $\mathcal{M}$ (without replacement within that subset). For each subsample, the complete pipeline is rerun using the $K$ selected models, producing a full set of probe properties and phemotype scores for all models in $\mathcal{M}$ derived from that subsample. Stability is then quantified by comparing all pairs of subsamples at the same $K$: at the probe level, robustness is measured by the Jensen–Shannon (JS) divergence between the resulting distributions of probe properties, and at the phemotype level, robustness is measured by the Spearman rank correlation between the corresponding model scores.

Table 9: **Probe-level stability under random subsampling.** $K$ (from 5 to 50) denotes the number of models used to estimate probe properties in each subsample. Entries report the Jensen–Shannon (JS) divergence among subsample-based probe properties, averaged over datasets; lower is better.

| Property | Jensen–Shannon (JS) divergence | | | | | |
|---|---|---|---|---|---|---|
| | $K = 5$ | $K = 10$ | $K = 20$ | $K = 30$ | $K = 40$ | $K = 50$ |
| Difficulty | 0.069 | 0.042 | 0.027 | 0.024 | 0.015 | 0.010 |
| Risk | 0.761 | 0.422 | 0.167 | 0.107 | 0.064 | 0.028 |
| Surprise | 0.656 | 0.333 | 0.136 | 0.095 | 0.059 | 0.028 |
| Uniqueness | 0.828 | 0.489 | 0.279 | 0.250 | 0.167 | 0.078 |
| Typicality | 0.001 | 0.043 | 0.025 | 0.013 | 0.005 | 0.001 |
| Bridge | – | 0.097 | 0.072 | 0.088 | 0.071 | 0.056 |

**Probe-level stability.** Table 9 reports the subsampling results. Difficulty and typicality already attain small JS divergence at $K = 5$, and become very stable once $K \geq 20$. Risk, surprise, and bridge decrease more slowly with $K$, but all fall below 0.10 when $K = 40$. Uniqueness is the only property that still shows a larger divergence at $K = 40$. This behavior is consistent with its definition: uniqueness asks whether a probe's response pattern actually helps to predict how models behave on other probes, which depends on fine-grained joint response patterns (both correct and incorrect). As a result, different subsamples can see very different patterns, so uniqueness naturally requires a larger and more stable model population to be estimated reliably, and this slower convergence is in line with the goal of the measure.

**Phemotype-level stability.** Table 10 summarizes the results for phemotype scores. For all five phemotypes, the mean Spearman rank correlation between subsamples already exceeds 0.97 at $K = 5$, rises above 0.98 at $K = 10$, and reaches 0.99 or higher once $K \geq 20$. These results show that

the ranking of models in the phemotype space is highly consistent across random subsamples, even when only a few models are used to estimate probe properties. Uniqueness asks whether a probe's response pattern actually helps to predict how models behave on other probes, which depends on fine-grained co-error structure and small, rare subgroups of models; as a result, different subsamples can see very different patterns, so uniqueness naturally requires a larger and more stable model population to be estimated reliably.

Table 10: **Phemotype-level stability under random subsampling.** $K$ denotes the number of models used to estimate probe properties in each subsample. Entries report the mean Spearman rank correlation among subsample-based model scores across repeats; higher is better.

| Phemotype | Number of models $K$ | | | | | |
|---|---|---|---|---|---|---|
| | 5 | 10 | 20 | 30 | 40 | 50 |
| Vigilance | 0.979 | 0.990 | 0.995 | 0.997 | 0.999 | 1.000 |
| Mastery | 0.972 | 0.980 | 0.991 | 0.995 | 0.998 | 0.999 |
| Transfer | 0.977 | 0.988 | 0.995 | 0.996 | 0.998 | 0.999 |
| Ingenuity | 0.972 | 0.987 | 0.992 | 0.993 | 0.996 | 0.998 |
| Astuteness | 0.980 | 0.993 | 0.997 | 0.998 | 0.999 | 1.000 |

**Overall stability.** These experiments compare independent subsamples of size $K$, and stability directly reflects whether the Probing Memes computations remain reliable under different model populations. The results show that the distributions of probe properties become highly stable across subsamples once the population reaches a moderate size: for most properties, the JS divergence at $K = 30$ is already close to or below $0.1$, with uniqueness being the only metric that converges more slowly due to its reliance on richer relational structure. Overall, the consistency observed across different subsample populations indicates that the Probing Memes paradigm produces consistent and population-insensitive characterizations of model behavior.

### F.2 POPULATION-LEVEL DISTRIBUTIONS OF PROBE PROPERTIES ACROSS REASONING MODES

Figure 19 presents the distribution of probe properties under the three reasoning modes across the three datasets. For all three reasoning modes, SimpleQA has the highest median difficulty and surprise, MATH-500 stays in the middle, and MMLU-Redux has the highest risk, bridge, and uniqueness. Several shifts match intuitive expectations about reasoning complexity. From Base to CoT to IR, both difficulty and risk decrease across all datasets, as stronger reasoning reduces the number of items on which the model fails and lowers the item difficulty. A few properties change more strongly and in this reasoning-specific way. Uniqueness increases under IR, especially on SimpleQA, showing that some items keep diverse error patterns even when accuracy improves. Bridge values also rise in IR, indicating more distinct community-level connections among items.

### F.3 EXPERIMENTAL INSIGHTS ON PROBE PROPERTIES UNDER REASONING POPULATIONS

As shown in Appendix F.2, items tend to become easier and less risky as the reasoning mode becomes more complex. However, the Sankey Figures 20 and 21 reveal several noteworthy phenomena that run counter to naive intuitions.

Figure 20 visualizes how probes move between difficulty tiers when the population changes from the Base to CoT population and from CoT to IR population. The main flow shows that more probes fall into the low and mid difficulty tiers for "stronger" populations with more complex reasoning abilities. At the same time, the flows also reveal **"degradation"** cases: some probes that lie in the low or mid tier under Base appear in the high tier under CoT, and some probes that lie in the mid tier under CoT appear in the high tier under IR. The observed **"degradation"** phenomenon suggests that excessive reasoning can sometimes hurt accuracy. In datasets like SimpleQA, where each item is a knowledge or fact-based question, a model may over-interpret the question and produce incorrect answers. Figure 21 shows an analogous phenomena for the risk property, where the overall risk decreases but similar degradation cases remain.

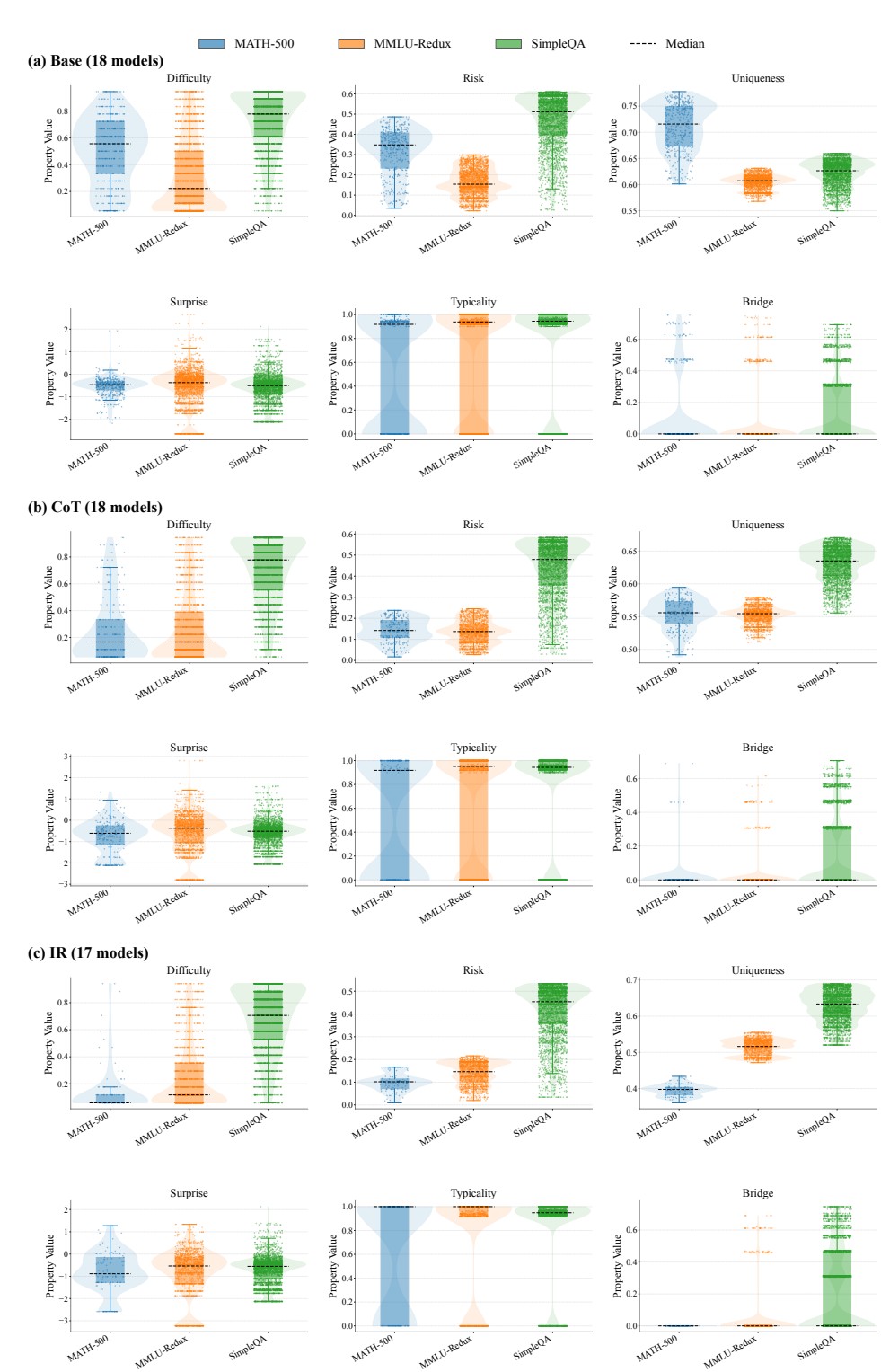

Figure 19: **Population-relative probe properties across datasets.** (a), (b), and (c) correspond to the Base, CoT, and IR reasoning modes, respectively.

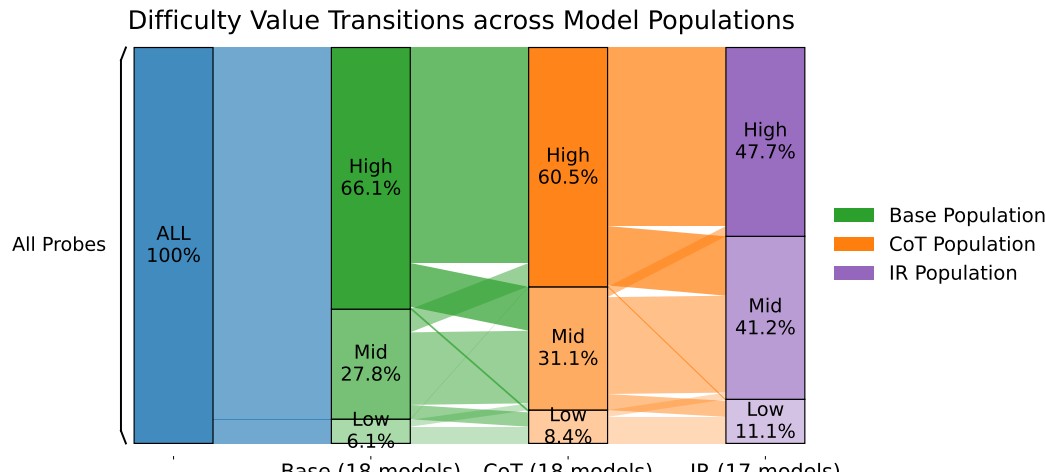

Figure 20: **Sankey diagram of probe difficulty transitions across model populations on SimpleQA.** Each column corresponds to a population of models under a specific reasoning mode. Within each population, probes are grouped by their difficulty values into three levels (*Low*, *Mid*, *High*). The bar heights show the relative fraction of probes in each tertile. Colored flows trace the same probes across populations, revealing how items move between difficulty levels as the population changes from Base to CoT to IR.

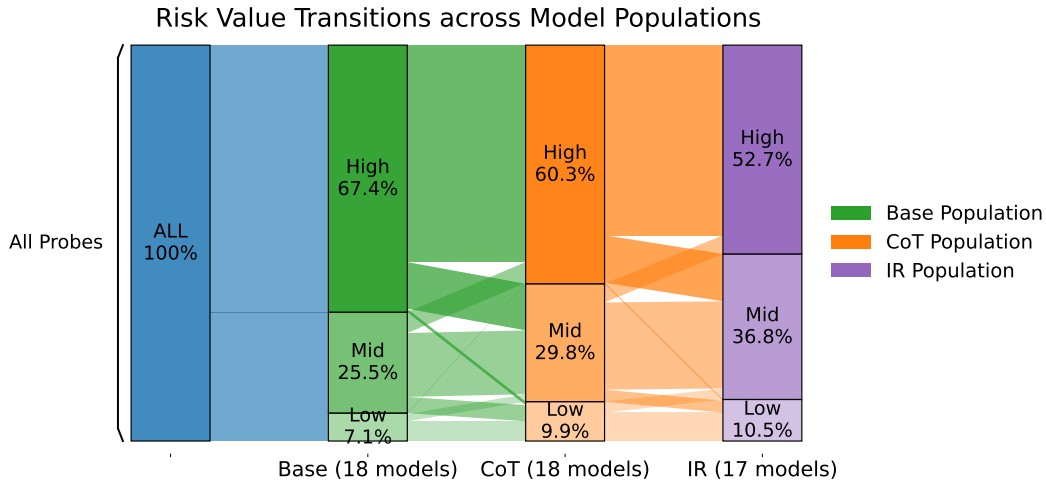

Figure 21: **Sankey diagram of probe risk transitions across model populations on SimpleQA.** Probes are grouped into three risk levels (*Low*, *Mid*, *High*) based on their risk values, and the bar heights give the fraction of probes in each group. These flows show how individual probes shift between low, mid, and high risk as the underlying population changes. Colored flows trace the same probes across populations, revealing how items move between risk levels as the population changes from Base to CoT to IR.

# G EFFECTS OF PARAMETER CHOICES ON THE PROBING MEMES PARADIGM

This section studies how variations in experiments influence the resulting probe properties and model phemotypes. These variations do not serve as correctness tests; instead, they reveal how different parameter regimes emphasize different aspects of the underlying structure.

## G.1 IMPACT OF CLUSTERING PARAMETERS

To evaluate their impact, this study varies the Leiden resolution parameter (Section 2.2) from its baseline value of 1.0 to 0.5 and 1.5, and lowers the similarity threshold from its baseline value of 0.9 to 0.8 and 0.7.

Table 11 demonstrates the comparison results with three statistics. Spearman correlation ($\rho$) measures the agreement in probe rankings. A higher $\rho$ means the two settings are closer. Jensen–Shannon divergence (JS) measures the difference between two score distributions. A lower JS means the two settings are closer. The Kolmogorov–Smirnov statistic (KS) measures the maximum gap between two empirical distributions. A lower KS also means the two settings are closer.

The first experiment changes the Leiden resolution in clustering, while a lower resolution gives fewer and larger clusters. A higher resolution gives more and smaller clusters. Moving from 0.5 to 1.5 has only a small effect on both bridge and typicality. Spearman correlation stays high and JS and KS stay close to zero. The study, therefore, adopts resolution 1.0 as a simple and balanced choice.

The second experiment changes the similarity threshold used to add edges between probes in the similarity graph. Different thresholds lead to large changes in the distributions of bridge and typicality. Spearman correlation decreases, and JS and KS increase when the threshold moves away from the chosen value. This shows that the threshold has a strong impact. The study sets the similarity threshold to 0.9 so that only probes with very similar error patterns are connected.

Table 11: **Comparison on clustering resolution and similarity threshold for *bridge* and *typicality*.** Each entry compares a variant run against the baseline metrics; higher Spearman $\rho$ and lower JS / KS indicate closer distributions.

| Dataset | Property | Param | $\rho$ (Spearman) | JS | KS |
|---|---|---|---|---|---|
| **Leiden resolution (vs. baseline)** | | | | | |
| MATH-500 | bridge | 0.5 | 0.758 | 0.067 | 0.119 |
| | | 1.5 | 0.919 | 0.152 | 0.088 |
| | typicality | 0.5 | 0.992 | 0.001 | 0.035 |
| | | 1.5 | 0.989 | 0.000 | 0.020 |
| MMLU-Redux | bridge | 0.5 | 0.759 | 0.002 | 0.017 |
| | | 1.5 | 0.781 | 0.004 | 0.019 |
| | typicality | 0.5 | 1.000 | 0.000 | 0.005 |
| | | 1.5 | 1.000 | 0.000 | 0.007 |
| SimpleQA | bridge | 0.5 | 0.817 | 0.004 | 0.020 |
| | | 1.5 | 0.850 | 0.003 | 0.013 |
| | typicality | 0.5 | 0.999 | 0.001 | 0.012 |
| | | 1.5 | 0.999 | 0.001 | 0.007 |
| **Similarity threshold (vs. baseline)** | | | | | |
| MATH-500 | bridge | 0.7 | 0.760 | 0.574 | 0.656 |
| | | 0.8 | 0.782 | 0.390 | 0.408 |
| | typicality | 0.7 | 0.753 | 0.652 | 0.543 |
| | | 0.8 | 0.801 | 0.529 | 0.532 |
| MMLU-Redux | bridge | 0.7 | 0.265 | 0.548 | 0.855 |
| | | 0.8 | 0.241 | 0.263 | 0.501 |
| | typicality | 0.7 | 0.838 | 0.636 | 0.541 |
| | | 0.8 | 0.856 | 0.312 | 0.321 |
| SimpleQA | bridge | 0.7 | 0.297 | 0.587 | 0.861 |
| | | 0.8 | 0.313 | 0.301 | 0.539 |
| | typicality | 0.7 | 0.758 | 0.678 | 0.627 |
| | | 0.8 | 0.749 | 0.421 | 0.481 |

## G.2 IMPACT OF PHEMOTYPE CONTRIBUTION PARAMETERS

The parameter $\tau$ (temperature parameter used when normalizing probe properties, as shown in Equation 12) controls how strongly probe-level contributions are concentrated on the most informative probes. Figure 22–24 report the model-level phemotype scores under different $\tau$ values.

When $\tau$ is small (0.1), the contribution becomes strongly polarized, and the resulting phemotype scores exhibit pronounced separation between models. When $\tau$ increases, the scores become more smoothed and less separated.

If $\tau$ is set too small, the contributions of different probe properties to the phemotype scores become highly polarized: although this yields strong model separation, the scores become dominated by a small number of probes, which is undesirable.

The setting $\tau = 0.2$ used in the main experiments achieves a balanced behavior: it reflects overall model strengths while retaining meaningful fine-grained differences.

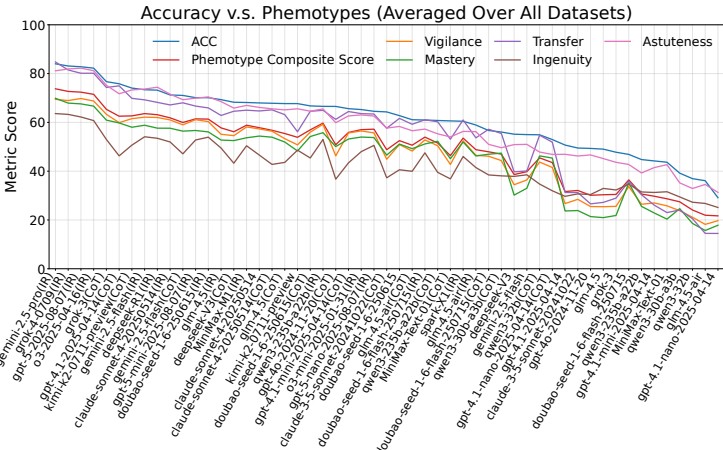

Figure 22: **Phemotype scores under $\tau = 0.1$.** Small $\tau$ yields highly separated model scores.

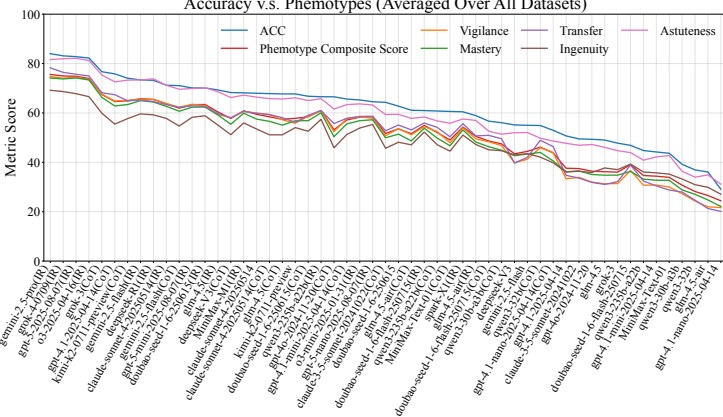

Figure 23: **Phemotype scores under $\tau = 0.2$ (main setting).** This setting balances separation and global consistency.

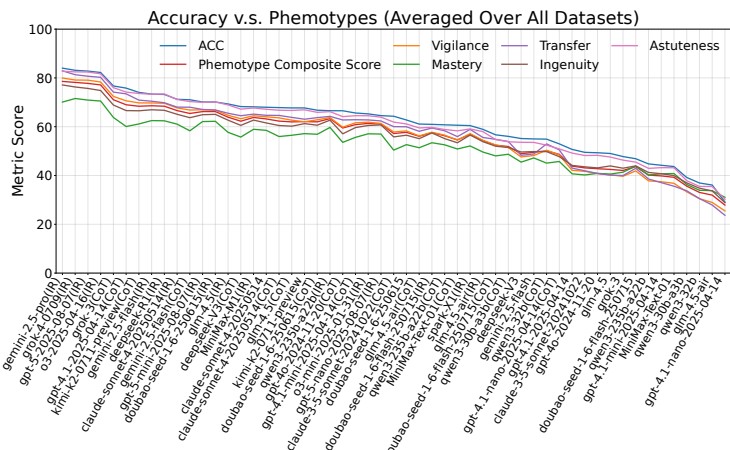

Figure 24: **Phemotype scores under** $\tau = 0.5$**.** Larger $\tau$ produces smoother, less differentiated scores.

## H MORE DETAILS OF OPEN LLM LEADERBOARD RESULTS

### H.1 DETAILS ABOUT MODELS AND DATASETS

Results from the Open LLM Leaderboard v2 (Fourrier et al., 2024) on Hugging Face were used to assess the validity of the probing-memes paradigm. This work collects the publicly available response data of models from the Open LLM Leaderboard on Hugging Face, and removes models with missing records as well as items with incomplete information, ensuring that all models have complete results on the same set of items. Six datasets were contained in the leaderboard: IFEval (Zhou et al., 2023), MATH (Hendrycks et al., 2021), MUSR (Sprague et al., 2023), MMLU-Pro (Wang et al., 2024), BBH (Suzgun et al., 2023) and GPQA (Rein et al., 2023). The three primary GPQA subsets (Diamond, Main and Extended) were available for all models. Therefore, GPQA-Diamond was used as a substitute.

### H.2 MORE RESULTS FROM OPEN LLM LEADERBOARD

Table 12 presents the phemotype benchmarks of models the from Open LLM Leaderboard. All scores are averaged over the six datasets. PCS refers to the Phemotype Composite Score, which represents the average of the five phemotypes, with scores scaled to 0–100 with one decimal. Column abbreviations are as follows: Acc for Accuracy, PCS for the composite, Vig for Vigilance, Mas for Mastery, Tra for Transfer, Ing for Ingenuity, and Ast for Astuteness.

### H.3 EXTENDED VISUALIZATIONS

Figure 25 shows the heatmaps of the perception span similarity of probes in the Open LLM Leaderboard dataset.

## I MODELS AND DATASETS

### I.1 MODELS

The following 28 models are included in this work. OpenAI (9 models): gpt-4.1-2025-04-14, gpt-4.1-mini-2025-04-14, gpt-4.1-nano-2025-04-14, gpt-4o-2024-11-20, o3-2025-04-16, o3-mini-2025-01-31, gpt-5-2025-08-07, gpt-5-mini-2025-08-07, gpt-5-nano-2025-08-07 (Achiam et al., 2023; OpenAI, 2025b;a); Anthropic (2 models): claude-3-5-sonnet-20241022, claude-sonnet-4-20250514 (Anthropic, 2024; 2025); Google (2 models): gemini-2.5-flash, gemini-2.5-pro (Comanici et al., 2025); DeepSeek (2 models): deepseek-V3, deepseek-R1 (Liu et al., 2024; Guo et al., 2025b);

Table 12: **Benchmark scores on phemotype**

| Model | Acc | PCS | Vig | Mas | Tra | Ing | Ast |
|---|---|---|---|---|---|---|---|
| calme-3.2-instruct-78b | 60.3 | 46.8 | 44.8 | 46.9 | 47.1 | 39.2 | 56.2 |
| CalmeRys-78B-Orpo-v0.1 | 60.0 | 46.5 | 44.5 | 46.6 | 46.8 | 38.8 | 55.8 |
| calme-3.1-instruct-78b | 59.6 | 45.9 | 43.7 | 46.0 | 46.1 | 38.4 | 55.4 |
| calme-2.4-rys-78b | 59.5 | 45.7 | 43.5 | 45.8 | 45.9 | 37.8 | 55.3 |
| FluentlyLM-Prinum | 58.2 | 41.8 | 39.2 | 42.3 | 42.4 | 32.3 | 53.0 |
| Homer-v1.0-Qwen2.5-72B | 57.9 | 41.7 | 38.4 | 42.1 | 42.2 | 33.8 | 52.2 |
| ultiima-72B | 57.1 | 40.7 | 37.2 | 41.1 | 41.2 | 32.7 | 51.1 |
| Gilgamesh-72B | 57.0 | 42.3 | 39.3 | 42.8 | 42.9 | 34.4 | 51.9 |
| shuttle-3 | 56.6 | 39.8 | 37.0 | 40.1 | 40.2 | 30.2 | 51.5 |
| T3Q-qwen2.5-14b-v1.0-e3 | 56.2 | 41.3 | 38.0 | 42.5 | 42.6 | 33.9 | 49.7 |
| T3Q-Qwen2.5-14B-Instruct-1M-e3 | 56.2 | 41.3 | 38.0 | 42.5 | 42.6 | 33.9 | 49.7 |
| test-2.5-72B | 56.1 | 40.8 | 37.9 | 41.6 | 41.7 | 32.7 | 50.3 |
| Qwen2.5-72B-Instruct-abliterated | 55.4 | 39.4 | 36.2 | 39.6 | 39.7 | 31.7 | 49.8 |
| sky-t1-coder-32b-flash | 55.3 | 40.3 | 38.4 | 40.1 | 40.2 | 31.4 | 51.6 |
| RYS-XLarge | 55.3 | 39.1 | 36.5 | 39.5 | 39.6 | 30.2 | 49.9 |
| calme-2.1-rys-78b | 55.2 | 39.7 | 37.3 | 39.8 | 39.9 | 31.2 | 50.3 |
| tempmotacilla-cinerea-0308 | 55.2 | 37.7 | 34.3 | 38.8 | 39.0 | 28.6 | 48.0 |
| ultiima-72B-v1.5 | 54.9 | 38.5 | 34.7 | 38.7 | 38.8 | 31.3 | 49.2 |
| Rombos-LLM-V2.5-Qwen-72b | 54.9 | 39.2 | 36.0 | 39.4 | 39.5 | 31.6 | 49.2 |
| li-14b-v0.4 | 54.7 | 37.8 | 35.4 | 38.1 | 38.3 | 27.5 | 49.7 |

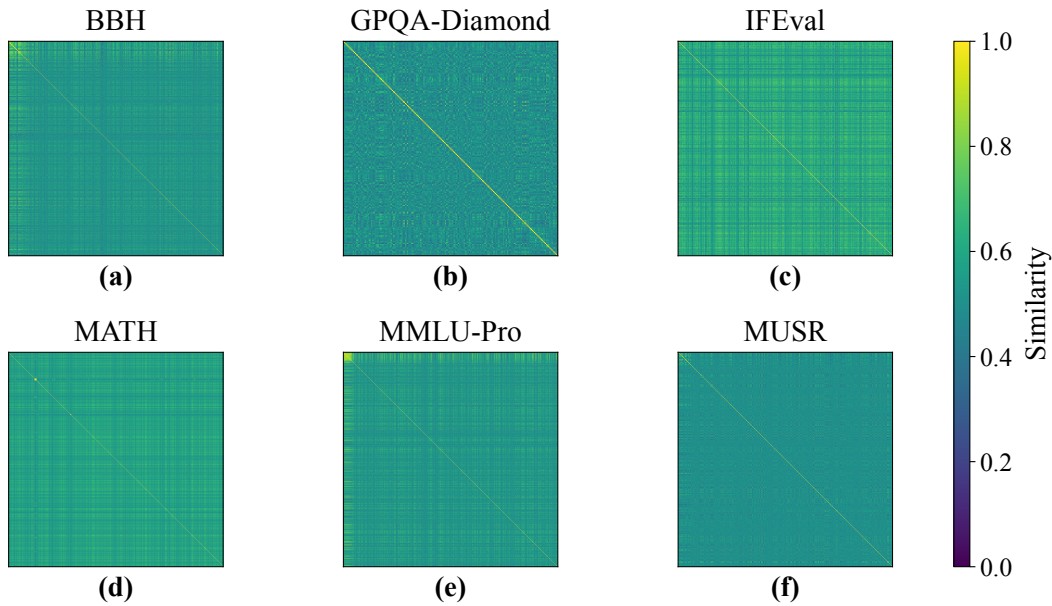

Figure 25: **Probe similarity heatmaps across datasets.** Probes are reordered by cluster to reveal blocks; similarities are computed from each probe's perception span.

Alibaba (3 models): qwen3-235b-a22b, qwen3-30b-a3b, qwen3-32b (Yang et al., 2025); xAI (2 models): grok-3, grok-4-0709(xAI, 2025a;b); MiniMax (2 models): MiniMax-Text-01, MiniMax-M1 (Li et al., 2025; Chen et al., 2025); ByteDance (2 models): doubao-seed-1-6-250615, doubao-seed-1-6-flash-250715(ByteDance, 2025a;b); Zhipu AI (2 models): glm-4.5, glm-4.5-air (Zeng et al., 2025); Moonshot AI (1 model): kimi-k2-0711-preview (Bai et al., 2025); iFlytek (1 model): spark-X1 (iFlytek, 2025).

## I.2 DATASETS

There are three datasets involved in the experiment part of this work, including MATH-500 (Lightman et al., 2023), MMLU-Redux (Gema et al., 2025), and SimpleQA (Wei et al., 2024).

**MATH-500** is a 500-problem subset of MATH (Hendrycks et al., 2021) curated by the OpenAI team. Its items come from high-school mathematics competitions, and many are challenging for humans. The high difficulty increases the discriminative power, which is crucial for revealing the distinct phemotypes of different models in the research. Another point is that all the problems in MATH-500 can be solved with step-by-step reasoning. Therefore, it is an excellent dataset for evaluating a model's stepwise reasoning ability.

**MMLU-Redux** is a revised version of MMLU (Hendrycks et al., 2020) dataset. It comprises 5,399 choice questions spanning 57 subject areas, including fields such as mathematics, physics, chemistry, political science, economics, law, and philosophy. Its broad subject coverage enables a comprehensive assessment of general knowledge across disciplines. In addition, MMLU-Redux is composed entirely of multiple-choice questions. This format allows for straightforward and highly accurate evaluation of model answers.

**SimpleQA** is a challenging dataset comprising 4,326 commonsense question-answer pairs. A distinctive feature of this dataset is its ternary answer evaluation scheme ("Correct", "Incorrect", or "Not Attempted"); for consistency, this study treated "Not Attempted" answers as "Incorrect". Due to its high difficulty, few models performed well in our experiments. The dataset's broad topical coverage enables a comprehensive evaluation of model capabilities across diverse domains. Moreover, all questions are phrased precisely and unambiguously. Their high reliability is ensured through a rigorous validation process involving multiple annotators who independently provided and cross-verified answers. This procedure guarantees a unique gold answer for each question.

# J DETAILED EXPERIMENTAL SETTINGS

## J.1 REASONING MODES

Two prompting templates are used in this paper: a default prompt and a chain of thought prompt. Models with internal reasoning (so-called reasoning models, like deepseek-R1 (Guo et al., 2025b)) use the default template, with the internal reasoning executed by the model; several of these models allow internal reasoning to be disabled, which permits use of both templates. Models without internal reasoning use both templates.

## J.2 PROBE FILTERING

As shown in Table 13, most dataset contains a number of probes that are either answered correctly or incorrectly by all models.

## J.3 HYPERPARAMETERS

For models without internal reasoning, temperature is set to 0, top-p to 1, and max tokens to 8192. Internal reasoning models often output a large amount of reasoning content. For models with internal reasoning, max tokens is set to 28672, and the remaining parameters follow the providers' defaults because these models often do not support very low temperature, settings and defaults are recommended. Internal reasoning models from the Qwen family max tokens does not include the number of reasoning tokens, so set max tokens to 8192 and thinking budget to 20480 (for limiting max reasoning output).

Table 13: **Summary of unanimous probes and counts after filtering.**

| Dataset | Total Probes | Unanimous (Correct) | Unanimous (Incorrect) | Remaining Probes |
|---|---|---|---|---|
| MMLU-Redux | 5,399 | 2,698 | 35 | 2,666 |
| Math-500 | 500 | 47 | 0 | 453 |
| SimpleQA | 4,326 | 10 | 569 | 3,747 |
| BBH | 5,759 | 0 | 0 | 5,759 |
| GPQA-Diamond | 198 | 0 | 0 | 198 |
| IFEval | 536 | 0 | 5 | 536 |
| MATH | 1,297 | 0 | 27 | 1,297 |
| MUSR | 756 | 0 | 0 | 756 |
| MMLU-Pro | 12,032 | 0 | 0 | 12,032 |

### J.4 PROMPT TEMPLATES

The boxes below present the prompts used for each dataset and reasoning mode, where "<question text>" denotes the text of the question. These prompts follow certain conventions. For instance, all prompts specify the required answer format, which varies across datasets. Furthermore, when testing under the Chain-of-Thought (CoT) setting, the phrase "Please reason step by step" is included to enable CoT reasoning, and models are instructed to output their reasoning process separately.

In terms of formatting, models are required to provide their final answers in the form "Answer:" followed by the answer, with no further explanation permitted. In addition, the MATH-500 prompt requires answers to be enclosed in \boxed{} to facilitate the extraction of mathematical expressions; the MMLU-Redux prompt requires the answer to be a single letter corresponding to the selected option; and the SimpleQA prompt imposes no additional requirements beyond the "Answer:" format.

**MATH-500** *CoT Prompting (CoT).*

```
Answer the following question.
Question:  ``<question text>''
Please reason step by step.
Your response must strictly follow the format below:
Reasoning Process:  {Explain your reasoning step by step}
Answer:  \boxed{Your final result without any explanation}
```

**MATH-500** *Default Prompting (Base) and Internal Reasoning (IR).*

```
Answer the following question.
Question:  ``<question text>''
Your response must strictly follow the format below:
Answer:  \boxed{Your final result without any explanation}
```

**MMLU-Redux** *CoT Prompting (CoT).*

```
Answer the following question.
Question:  ``<question text>''
Please reason step by step.
Your response must strictly follow the format below:
Reasoning Process:  {Explain your reasoning step by step}
Answer:  {Your final choice letter without any explanation}
```

**MMLU-Redux** *Default Prompting (Base) and Internal Reasoning (IR).*

```
Answer the following question.
Question:  ``<question text>''
Your response must strictly follow the format below:
Answer:  {Your final choice letter without any explanation}
```

**SimpleQA** *CoT Prompting (CoT).*

```
Answer the following question.
Question:  ``<question text>''
Please reason step by step.
Your response must strictly follow the format below:
Reasoning Process:  {Explain your reasoning step by step}
Answer:  {Your final answer without any explanation}
```

**SimpleQA** *Default Prompting (Base) and Internal Reasoning (IR).*

```
Answer the following question.
Question:  ``<question text>''
Your response must strictly follow the format below:
Answer:  {Your final answer without any explanation}
```

### J.5 DETAILS AND METHODS OF ANSWER VERIFICATION

Several issues arose during the extraction and evaluation of model responses. These included non-compliant output formats (despite explicit instructions), responses exceeding token limits, and model refusals to answer sensitive questions. Tables 14, 15, and 16 present the statistics for these respective issues. The token limits are specified in Appendix J.3.

The experiment applied two rounds of verification. The first round enforced the prompt's formatting requirements strictly: any response that failed to comply with the required format was treated as incorrect. The second round attempted to match and extract answers using a variety of possible formats, which did not conform to the prompt. Therefore, a purely formatting error was always regarded as incorrect in the first round verification, but could be viewed as correct in the second round verification if the model's output contained the correct answer. Responses that exceeded the token number limits and responses in which the model refused to answer were treated as incorrect in both verification rounds. The data presented in the experiments and analyses were obtained from the second round of verification.

Each round of answer verification comprises two steps: answer extraction and answer evaluation. The answer extractor extracts the model's answer (without any explanation) from the model's response, while the answer evaluator compares the extracted answer with the golden answer. Different datasets used different methods to extract and evaluate answers, and the methods are presented below.

**MATH-500.** MATH-500 dataset uses Math-Verify (Kydlíček, 2025) library to extract and evaluate answers. The extractor first attempts to extract the content enclosed by \boxed{} from the model response using regular expressions. If the attempt fails, the response will be sent directly to Math-Verify. Math-Verify is capable of extracting answers in LaTeX format as well as numeric/expression formats from the model response. It uses the following formats to extract answers in descending priority:

- Explicit final answer (e.g., "Final answer is 3. I hope");
- General final answer (e.g., "final answer is 3") and boxed expressions (e.g., \boxed{3}) at the same priority;
- Answer with a colon (e.g., "answer: 3");
- Answer without a colon (e.g., "answer is 3");
- Unanchored matches (e.g., "3").

Unanchored matches carry some risk of extracting numbers/expressions that appear in the response but are not the model's perceived answer; however, manual per-item inspection found no such errors. After extraction, Math-Verify normalizes the answer format and then parses it with SymPy. The golden answer is likewise converted to SymPy, and Math-Verify judges correctness by comparing the two SymPy expressions.

Table 14: **Numbers of refusals to answer**

| Model | MMLU-Redux | SimpleQA |
|---|---|---|
| qwen3-235b-a22b(IR) | 6 | 34 |
| spark-x1(IR) | 12 | 7 |
| MiniMax-M1(IR) | 10 | 8 |
| qwen3-30b-a3b(CoT) | 5 | 8 |
| qwen3-235b-a22b(CoT) | 5 | 7 |
| glm-4.5(CoT) | 7 | 5 |
| qwen3-30b-a3b | 5 | 5 |
| qwen3-235b-a22b | 4 | 6 |
| qwen3-32b(CoT) | 5 | 5 |
| qwen3-32b | 2 | 5 |
| glm-4.5-air(CoT) | 3 | 3 |
| glm-4.5(IR) | 5 | 0 |
| glm-4.5-air(IR) | 3 | 1 |
| glm-4.5 | 1 | 0 |
| glm-4.5-air | 1 | 0 |
| others | 0 | 0 |

Table 15: **Numbers of responses exceeding token limit**

| Model | MATH-500 | MMLU-Redux | SimpleQA |
|---|---|---|---|
| glm-4.5-air(IR) | 31 | 506 | 452 |
| glm-4.5(IR) | 15 | 209 | 447 |
| gemini-2.5-flash(CoT) | 23 | 13 | 12 |
| doubao-seed-1-6-flash-250715(CoT) | 5 | 2 | 31 |
| glm-4.5-air(CoT) | 10 | 15 | 12 |
| glm-4.5(CoT) | 8 | 4 | 10 |
| MiniMax-M1(IR) | 0 | 2 | 18 |
| gpt-4.1-mini-2025-04-14(CoT) | 0 | 0 | 14 |
| gpt-4.1-2025-04-14(CoT) | 6 | 2 | 2 |
| kimi-k2-0711-preview(CoT) | 7 | 1 | 2 |
| kimi-k2-0711-preview | 6 | 0 | 0 |
| grok-4-0709(IR) | 2 | 1 | 3 |
| doubao-seed-1-6-250615(CoT) | 2 | 2 | 2 |
| doubao-seed-1-6-flash-250715(IR) | 2 | 0 | 3 |
| gpt-4.1-nano-2025-04-14(CoT) | 0 | 0 | 5 |
| gemini-2.5-flash | 2 | 0 | 2 |
| doubao-seed-1-6-flash-250715 | 1 | 0 | 2 |
| qwen3-30b-a3b(CoT) | 0 | 0 | 3 |
| deepseek-reasoner(IR) | 2 | 0 | 0 |
| gpt-4.1-nano-2025-04-14 | 0 | 0 | 2 |
| deepseek-chat(CoT) | 1 | 0 | 0 |
| o3-2025-04-16(IR) | 1 | 0 | 0 |
| gpt-5-2025-08-07(IR) | 1 | 0 | 0 |
| claude-sonnet-4-20250514(CoT) | 1 | 0 | 0 |
| doubao-seed-1-6-250615 | 0 | 1 | 0 |
| qwen3-30b-a3b | 0 | 0 | 1 |
| doubao-seed-1-6-250615(IR) | 0 | 0 | 1 |
| others | 0 | 0 | 0 |

Table 16: **Numbers of malformed answers**

| Model | MATH-500 | MMLU-Redux | SimpleQA |
|---|---|---|---|
| doubao-seed-1-6-flash-250715(IR) | 100 | 956 | 838 |
| doubao-seed-1-6-flash-250715 | 19 | 707 | 912 |
| MiniMax-M1(IR) | 497 | 1119 | 21 |
| doubao-seed-1-6-250615 | 5 | 21 | 693 |
| spark-x1(IR) | 88 | 399 | 16 |
| gemini-2.5-flash(CoT) | 219 | 220 | 0 |
| gemini-2.5-flash(IR) | 249 | 122 | 4 |
| doubao-seed-1-6-flash-250715(CoT) | 39 | 26 | 292 |
| grok-3(CoT) | 282 | 24 | 9 |
| deepseek-reasoner(IR) | 299 | 4 | 0 |
| glm-4.5-air(IR) | 168 | 84 | 22 |
| grok-4-0709(IR) | 234 | 1 | 0 |
| qwen3-235b-a22b(IR) | 191 | 6 | 2 |
| glm-4.5(IR) | 163 | 3 | 3 |
| gemini-2.5-flash | 152 | 3 | 0 |
| qwen3-235b-a22b | 18 | 97 | 0 |
| doubao-seed-1-6-250615(CoT) | 3 | 12 | 100 |
| doubao-seed-1-6-250615(IR) | 2 | 12 | 101 |
| gpt-4o-2024-11-20(CoT) | 12 | 77 | 2 |
| gemini-2.5-pro(IR) | 61 | 9 | 0 |
| qwen3-235b-a22b(CoT) | 14 | 42 | 0 |
| kimi-k2-0711-preview(CoT) | 46 | 4 | 4 |
| gpt-4.1-2025-04-14(CoT) | 0 | 42 | 0 |
| MiniMax-Text-01(CoT) | 4 | 24 | 8 |
| gpt-4.1-nano-2025-04-14(CoT) | 5 | 31 | 0 |
| qwen3-32b(CoT) | 15 | 20 | 0 |
| o3-mini-2025-01-31(IR) | 1 | 11 | 21 |
| qwen3-30b-a3b(CoT) | 9 | 22 | 0 |
| grok-3 | 2 | 24 | 2 |
| qwen3-30b-a3b | 19 | 2 | 0 |
| deepseek-chat | 12 | 7 | 0 |
| kimi-k2-0711-preview | 18 | 1 | 0 |
| glm-4.5(CoT) | 10 | 8 | 0 |
| MiniMax-Text-01 | 0 | 14 | 4 |
| gpt-5-mini-2025-08-07(IR) | 0 | 11 | 5 |
| gpt-4.1-nano-2025-04-14 | 9 | 5 | 0 |
| qwen3-32b | 10 | 2 | 0 |
| gpt-4.1-mini-2025-04-14(CoT) | 4 | 7 | 0 |
| deepseek-chat(CoT) | 4 | 6 | 0 |
| claude-3-5-sonnet-20241022(CoT) | 1 | 8 | 0 |
| glm-4.5-air(CoT) | 1 | 7 | 0 |
| gpt-5-2025-08-07(IR) | 0 | 5 | 1 |
| gpt-4.1-2025-04-14 | 0 | 6 | 0 |
| claude-sonnet-4-20250514(IR) | 0 | 1 | 5 |
| o3-2025-04-16(IR) | 0 | 5 | 0 |
| claude-sonnet-4-20250514(CoT) | 0 | 4 | 0 |
| gpt-4.1-mini-2025-04-14 | 1 | 3 | 0 |
| gpt-5-nano-2025-08-07(IR) | 0 | 2 | 1 |
| glm-4.5 | 1 | 1 | 0 |
| glm-4.5-air | 0 | 2 | 0 |
| gpt-4o-2024-11-20 | 0 | 0 | 1 |
| others | 0 | 0 | 0 |

**MMLU-Redux.** Regular expressions are used to extract the one-letter answer in MMLU-Redux. There are three modes in the answer extraction as follows:

- Searching answer using "Answer"/"answer" anchor. If multiple matches occur, then take the last match.

- Searching answer with other anchors, like "{}" and "**". These anchors do not mean the letter beside them is definitely the answer. Therefore, the extractor accepts a match as an answer only if exactly one match is found.

- Full-string match. Sometimes models give one-letter responses, with no anchors existing in these responses. However, it is risky to extract non-anchor answers in responses. To address this issue, the extractor applies full-string matches, matching responses like "A", "A." and so forth.

After the extraction, the evaluation step only requires a simple string comparison between the extracted answer and the golden answer.

**SimpleQA.** In SimpleQA, the extractor only extracts the content following "Answer:" as the answer, without any other format requirements. GPT-4.1 is employed as an LLM evaluator to evaluate the answers of the models under test by comparing their answers with the golden answer. The prompt for the LLM evaluator is the same as that in SimpleQA's official publication paper (Wei et al., 2024). Under this prompt, the LLM evaluator classifies the answer into three categories: Correct, Incorrect, and Not Attempted. A response will be classified into "Not Attempted" if the model recognizes its inability to solve the problem and refrains from providing an answer. As long as it gives an answer, it will be classified into "Correct" or "Incorrect". In this work, only answers classified into the "Correct" category were regarded as correct answers, and other answers were all deemed incorrect.

# K   USE OF LARGE LANGUAGE MODELS

This article was written with the moderate use of LLMs as polishing tools.

