# OpenReview forum: "Probing Memes in LLMs: A Paradigm for the Entangled Evaluation World"
_ICLR.cc/2026/Conference — Submitted to ICLR 2026_

### Official Review · Reviewer_iqfE · 2025-10-21

**Soundness:** 3
**Presentation:** 1
**Contribution:** 2
**Rating:** 2
**Confidence:** 3

**Summary:**

This paper proposes Probing Memes, an evaluation paradigm that jointly considers individual samples in the dataset and models evaluated on them, in contrast to the previous convention that only looks at a single model's performance aggregated over samples. The authors design evaluation metrics to model each sample as "probes" to study the capability properties of the dataset vs. the model. Experimental results reveal distinct dataset-model properties across several benchmarks, such as a high probing value of "surprise" on MMLU-pro.

**Strengths:**

1. Thorough joint analysis along the dimensions of sample and model is an interesting yet underexplored direction.
2. The proposed Probing Memes paradigm is well-motivated. The overall idea is clear.
3. Experiments are extensive and well-deliver the proposed idea.

**Weaknesses:**

1. The novelty of this paper lies in the proposed two-dimensional (sample-model) evaluation. However, the novelty may be undermined by the fact that previous work has been using such analysis for specific purposes. For example, [a] defines and calculates the difficulty of samples in benchmark datasets, along with 52 LLMs of different sizes, to investigate emergent abilities. This raises concern about the insufficient discussion of and distinction from related work.

2. This proposed paradigm relies on the assumption that there have been sufficiently many LLMs evaluated on the target dataset. If the number of LLMs is "not enough," the values of probes will become unreliable. On the other hand, with more LLMs evaluated on the target dataset, the values of probes and phenotypes will change since they depend on the tested LLMs.

3. I am concerned about this paper's framing. Terms like "meme", "probe," and "phenotype" might be a bit distracting and imprecise.

4. Discussions on experiments and findings seem not deep. For example, it is no surprise that there are samples with high surprise value in MMLU-pro. Can your paradigm help answer why this happens? Arguments for the practical value of this paradigm are insufficient and unclear. Another example: Given you find that "probes in IFEval, GPQA-Diamond, and BBH exhibit relatively high uniqueness" (line 431), what can we do next to make it not only a "good to know" thing? What is this finding's practical value?

a. [U-shaped and Inverted-U Scaling behind Emergent Abilities of Large Language Models](https://openreview.net/forum?id=jjfve2gIXe)

**Questions:**

1. Can you make a comparison of the definitions of memes in this paper and in The Selfish Gene? It's unclear to me how this term can be borrowed from the book.

2. My current understanding regarding 1. is that a meme is a facet of LLM's ability (by line 79, " From this perspective, the abilities of LLMs are conceptualized as composed of latent memes"). In this case, are the proposed 7 kinds of probes only a small subset of an unknowingly big set of all memes?

3. In Section 2.2, you define **uniqueness** through the averaged conditional entropy of two probes and define **ϕ-coefficient** to calculate the similarity of two probes. It seems to me that entropy (e.g., JS entropy) is also a natural metric to determine whether two probes should have an "edge". Is my understanding correct?

4. In Figure 6, can you explain why Astuteness has nearly identical weight distribution over the three datasets?

5. In line 429, you mention MMLU-pro has many samples with high surprise. Is this possibly due to the nature of MCQs? Assume a challenging question where a frontier LLM gets wrong. If a low-ability LLM simply randomly guesses all samples and thus picks the correct choice by chance, the surprise value of this question will be high.

6. The statement in lines 461-462 is a bit abrupt and without any supporting argument. You can move this statement to the earlier section and support it with concrete examples—text only is ok; quantitative experiments are a huge improvement.

7. In line 471, why is full reproducibility not guaranteed even with temperature=0? Isn't the probe calculation process deterministic?

My current score for this paper is 2. Given that I have many questions, I may raise my score to 4 or 6 if this paper's significance and novelty becomes clear and good to me during the discussion stage.

---

> ### Author Response · Authors · 2025-11-25
> **Response to Reviewer iqfE (Part 1/4)**
>
> **Dear Reviewer iqfE:**
>
> We are grateful for your positive assessment of our Probing Memes paradigm and the corresponding experiments, and for the time you have devoted to reviewing this work. **Your question #2 has substantially contributed to improving the clarity and rigor of our paper**, and we are very grateful for that. Our responses to your comments are as follows:
>
> **1. [*For Weakness #3, Question #1*] Motivation for Introducing the Concept of Memes**
>
> Our motivation for adopting the "meme" framing is to model the fine-grained behavioral regularities revealed through data–model interaction at a resolution that traditional "skills" or "capabilities" terms cannot express. This perspective is consistent with prior work. For example, Birchall et al. (reference [1] in our comments) describe reasoning patterns in LLMs as meme-like units. Our contribution is to make this idea formal and operational, enabling a structured and extensible evaluation paradigm based on memes, probe properties, and phemotypes.
>
> Concretely, we define a meme space $\mathcal{V}$ = { $\mu_1, \mu_2,\dots,\mu_R$ }., where each elementary meme represents a reusable, compositional behavioral tendency that remains latent unless elicited by specific probes. Probes serve as controlled stimuli that expose subsets of $\mathcal{V}$, and phemotypes formalize the resulting observable patterns of these subsets into quantifiable descriptors.
>
> Without this "meme" framing, several key constructs in our evaluation, such as probe properties like uniqueness and bridge, and the compositional phemotypes, would be ill-defined under a coarse capability view.
>
> We appreciate the reviewer's suggestion and have improved clarity by explicitly formalizing the meme, probe property, and phemotype spaces in `Appendix B`. This addition clarifies the underlying structure but does not affect the method or results.
>
> A comparison between the concepts of memes and phemotypes in this work and the corresponding ideas in memetics can be found in `Table 2` from `Appendix A.3` (compared with Dawkins and other foundational work in memetics).
>
>
> **2. [*For Question #2*] Discussion about Memes, Phemotypes and Their Spaces**
>
> We sincerely thank you for raising this insightful question. It highlighted that our original formal exposition of the underlying spaces in the paradigm was insufficient. We state in the paper that “both phemotypes and probe properties are designed to be extensible,” and "more revealing property designs may exist that detect a wider range of memes.", but we do not explicitly formalize the design of the corresponding spaces. In the revised version, we now provide a more detailed formalization of the spaces of memes, probe properties, and phemotypes in `Section 2` and `Appendix B`. Your question has substantially contributed to improving the clarity and rigor of our paper, and we are very grateful for that.
>
> Your understanding in ***Question #2*** is correct: in our formulation, a meme corresponds to a latent unit of an LLM’s ability, drawn from a very large underlying space $\mathcal{V}$ = { $\mu_1, \mu_2,\dots,\mu_R$ }. In the same spirit, probe properties $A$ = { $a^{(1)}, a^{(2)}, \dots, a^{(K)}$ } and phemotypes $\mathcal{H}$ are also instances drawn from their own conceptual spaces. Conceptually, there is a mapping $\Phi: 2^{\mathcal{V}} \to \mathcal{H}$, which assigns to each meme subset $\mathcal{V}' \subseteq \mathcal{V}$ a phemotype $\Phi(\mathcal{V}') \in \mathcal{H}$. The specific properties and phemotypes used in this paper are a well designed, interpretable, and practically useful choice within these spaces. Similarly, the models and datasets used in evaluation can be viewed as different populations, and our framework studies these populations by examining which memes they express. This is the sense in which we use the phrase *Entangled Evaluation World* in our title.
>
> **reference**
>
> [1] Birchall et al. Parrots Are All You Need: A Memetic Framework for Apparent Reasoning in LLMs (PhilArchive 2025)

---

> ### Author Response · Authors · 2025-11-25
> **Response to Reviewer iqfE (Part 2/4)**
>
> **3. [*For Weakness #1*] Novelty and Significance Beyond Related Work**
>
> The paradigm itself is our primary contribution. The Probing Memes paradigm makes it possible to jointly evaluate both datasets and models, viewing them as part of an entangled evaluation world. As detailed in `Appendix B`, depending on the research objective, different probe properties can be combined to target specific memes and further aggregated into phemotypes, yielding interpretable descriptors of model behaviour. The six properties introduced in this paper are only a starting point; they are chosen because they are both easy to interpret and effective at quantifying behavioural patterns (for example, surprise is a novel property first proposed in this work, and the case in `Figure 2` is derived from the highest-surprise item in SimpleQA). More importantly, the paradigm itself is extensible: new properties and new phemotype definitions can be incorporated as needed, while preserving the same structured way of analysing model–dataset interactions.
>
> Most existing work either focuses on evaluating models or on evaluating datasets. For example, in the paper (Wu and Lo, reference [2] in this comment) you mentioned, the resulting item-level "difficulty" metric is not used to define any corresponding model-level metric; its evaluation of models still relies solely on accuracy.
>
> Goel et al. (reference [1] in this comment) also have shown that models with similar overall accuracy may fail on very different subsets of items, indicating that a single accuracy metric can obscure important differences between models. Our Probing Memes paradigm goes a step further beyond this phenomenon: instead of only comparing the sets of correctly and incorrectly answered items, it explicitly analyses the items themselves, assigning each item a set of interpretable probe properties and using these properties to characterise model behaviour.
>
> Additional discussion of related work and our paradigm has been added to `Appendix A`.
>
> **4. [*For Weakness #2*] The Impact of Population Size on the Experimental Results and Further Discussion**
>
> `Appendix F` examines how changes in the model population affect probe properties and phemotypes. `Appendix F.1` studies robustness to population size: for each K in {5, 10, 20, 30, 40, 50}, we draw 10 independent subsamples from the 53 models, recompute probe properties and phemotypes from each subsample, and measure stability by comparing all pairs of subsamples at the same K (Jensen–Shannon divergence for properties and Spearman rank correlation for phemotypes). The results show that most properties become stable (JS divergence below 0.1 once K reaches 30, except uniqueness, which converges more slowly but is reasonable and interpretable) and that all phemotypes already achieve Spearman correlations of 0.99 at K = 20.
>
> While it is true that changing the model population can lead to shifts in probe properties and phemotypes, this is in fact a core motivation of our paradigm: it is designed to study different populations and to reveal how datasets and models behave under each of them.  `Appendix F.2` then uses the paradigm to study different populations directly by partitioning the 53 models into three groups based on their reasoning modes. The results show that, as models move from light reasoning to more complex reasoning modes, overall item difficulty and risk tend to decrease (`Figure 20` and `Figure 21`). However, we also observe a counter-intuitive phenomenon: some items that are easy under light reasoning but become harder under more complex reasoning, indicating that increased reasoning can sometimes lead to lower accuracy.
>
> **reference**
>
> [1] Goel et al. Great Models Think Alike and this Undermines AI Oversight, (ICML 2025)
>
> [2] Wu & Lo. U-shaped and Inverted-U Scaling behind Emergent Abilities of Large Language Models (ICLR 2025)

---

> ### Author Response · Authors · 2025-11-25
> **Response to Reviewer iqfE (Part 3/4)**
>
> **5. [*For Weakness #4, Question #5*] Explaining the Mechanisms Behind the Results and Their Practical Utility**
>
> In `Appendix E.4`, we add a case study of the high-surprise phenomenon in MMLU-Redux. We re-run the selected high-surprise probes 20 times under three hyperparameter settings (temperature 0.0, temperature 0.6, and temperature 0.0 with a "do not guess" hint) and observe both deterministic and random-guessing behaviors.
>
> Some item–model pairs show consistently high accuracy across all settings (for example, *doubao-seed-1-6-flash-250715* and *glm-4.5-air* reach close to 100\% on certain items), while others stay near chance or fluctuate strongly in a way that matches random guessing (for example, *glm-4.5-air* staying around 15\% accuracy on one item).
> While some underlying causes of these insights (e.g., why random guessing occurs) lie in lower-level mechanisms of model training and internal representations that our paradigm alone cannot fully explain, this analysis illustrates how the Probing Memes paradigm, and in particular the surprise property, makes such patterns visible at the probe level and supports targeted case studies like this one.
>
> We also outline several potential uses and application directions, and illustrate them with experiments in `Appendix E`:
>
> - **Phemotype-guided model selection (E.1)** The Probing Memes paradigm can naturally be applied to realistic multi-agent routing tasks, where items are dispatched to different agents based on their phemotype profiles. We use phemotype scores computed on MATH-500 to identify, among models with similar overall accuracy, those that are better suited for hard items and those that are better suited for easy items. On MATH-FULL, for one representative pair of similarly accurate models, difficulty-based routing attains a union accuracy of 77.02\%, compared to 72.24\% and 76.02\% for the two individual models and 73.89\%±0.29 for a balanced partition baseline. These experiments show how phemotype scores can be used to choose models that better match downstream requirements than considering accuracy alone.
>
> - **Property-guided dataset construction (E.2)**
> Our paradigm can be used for property-guided subset construction. We study a concrete dataset reduction task inspired by IRT-based work. In this setting, we use typicality to select a compact subset of items. The resulting subset achieves performance close to strong domain-specific IRT baselines (for example, on MATH and IFEval the Spearman rank correlations differ from the MetaBench baseline by only 0.011 and 0.003, respectively). On some datasets, it even surpasses these baselines (e.g., it achieves lower MAE than TinyBench on MATH, 0.026 vs 0.030, and on GPQA-Diamond, 0.023 vs 0.035). This concrete case illustrates how probe properties can be used to select compact yet informative subsets of items and preserve evaluation fidelity. On some datasets, the typicality-based subsets still show a substantial gap relative to the IRT baselines (e.g., on MMLU-Pro and BBH), reflecting a different design goal: the IRT methods are explicitly trained to optimize dataset reduction on each benchmark, whereas our method directly uses probe properties for selection. More broadly, depending on the specific research goal, different properties can be used together to guide subset construction. For example, one can construct reduced datasets that focus on high-risk and high-surprise items.
>
>
> - **Methods for helping researchers conduct in-depth analysis of relationships among models (E.3)**
> We use phemotypes to characterize over 4,000 models in the OpenLLM Leaderboard population and visualize their distribution in the memetic space. Models that share the same base model tend to lie close to each other in this space. A similar conclusion holds for training strategies: nearby models also tend to adopt similar training strategies.
> This demonstrates that the design of phemotypes provides a practical way to uncover similarities and differences among LLMs.

---

> ### Author Response · Authors · 2025-11-25
> **Response to Reviewer iqfE (Part 4/4)**
>
> **6. [*For Question #3*] Edge Construction Between Probes**
>
> We sincerely appreciate your interest in our metric design. We have considered such issues during the initial design of the metrics. In our view, entropy-based methods may be not ideal for defining probe-probe edges in our similarity matrix.
>
> First, in the uniqueness metric we use conditional entropy, which is inherently
> asymmetric: $H(P_k \mid P_i) \neq H(P_i \mid P_k)$ in general, so it does not
> serve naturally as a symmetric edge weight between probes $i$ and $k$.
>
> Second, while Jensen-Shannon (JS) divergence is symmetric, it raises a
> different issue in our setting:
>
> Using the standard definition:
>
>  $\mathrm{JSD}(P\|Q) = \tfrac{1}{2}\mathrm{KL}(P\|M) + \tfrac{1}{2}\mathrm{KL}(Q\|M),~~ M = \tfrac{1}{2}(P+Q), $
>
> the result depends on how the probability distributions $P$ and $Q$ are constructed from the underlying $0/1$ vectors. If we construct $P$ and $Q$ using only the "correct" side (where 1 in the perception matrix denotes a correct answer), then only cases like $(1,0)$ and $(1,1)$ contribute to the similarity computation, while $(0,0)$ pairs contribute nothing, which is unreasonable when comparing the similarity between two vectors.
>
> By contrast, the $\phi$-coefficient explicitly incorporates all four outcome combinations $(0,0)$, $(1,0)$, $(0,1)$, and $(1,1)$, making it a more suitable similarity measure for constructing probe-probe edges in our framework. Nevertheless, the property definitions can be designed according to specific requirements, as long as they are reasonable.
>
>
> **7. [*For Question #4*] Details about Visualization**
>
> Astuteness has nearly identical weight distributions across the three datasets, which is expected and follows directly from the design of `Equation (12)`. To make phemotypes comparable across datasets, we first normalize each property based on its rank and then apply a sigmoid transformation to standardize values across datasets. As a result, for any single property, the curve becomes similar across datasets, especially for properties with continuous values (such as surprise). When multiple properties are combined into a phemotype (e.g., Vigilance), the situation changes: probes exhibit different value patterns across properties, and datasets differ in which probes are simultaneously high or low on certain properties. This leads to more diverse combined shapes, which is exactly what the differing curves across datasets in the figure reflect.
>
>
> **8. [*For Question 6*] Commonality and Divergence Among Models Revealed by Phemotypes**
>
> The statement "helping to investigate potential relationships in their training data, base models, and training strategies" in line 461-462 (line 431-432 in revision) means that, since phemotypes can reveal differences in model behaviors and capabilities, using multiple phemotypes as feature representations for models should make certain patterns visible in the space: models that share similar design principles are expected to exhibit commonality (i.e., cluster together), whereas models with different design principles are expected to exhibit divergence (i.e., spread apart).
>
> In `Appendix E.3`, we provide new visualizations that further highlight model training strategies. Within each model family, models using the same strategy cluster together (`Figure 17-18`), reinforcing our statement on lines 461-462 (431-432 in revision). This shows that researchers can use our paradigm and phemotypes to characterize model behavior and identify similarities and differences across model populations, for instance, probing whether closed-source models share an underlying base model. Our notion of memes likewise offers a richer way to describe model properties in this entangled world of data and models.
>
> **9. [*For Question #7*] The Effects of Deterministic hyperparameter (Temperature = 0)**
> The probe computation process is deterministic and all probe properties are computed using fixed formulas with no randomness involved. The reproducibility issue comes from the model outputs, not the probes.
>
> The temperature parameter influences the randomness of model outputs by affecting the softmax computation.
> In fact, even with temperature = 0 and top-p = 1 (this configuration is used in most benchmarks and also in our experiments), modern LLMs do not guarantee fully deterministic outputs for reasons:
>
> 1. In practice, as model architectures evolve, certain design elements introduce additional sources of randomness (for example, multi-expert routing in MoE models).
> 2. Moreover, float point numerical fluctuations can also cause output variability.
>
> Therefore, although the probe calculations are deterministic, the model responses used to construct the perception matrix may vary slightly across runs. This is why full reproducibility cannot be guaranteed. This is the common issue faced by most benchmarks.

---

### Official Review · Reviewer_esKs · 2025-10-31

**Soundness:** 3
**Presentation:** 3
**Contribution:** 3
**Rating:** 6
**Confidence:** 3

**Summary:**

This paper solved the prob of evaluating LLMs only with coarse, accuracy-centric scores that ignore how models and datasets interact at scale. This paper proposed the Probing Memes paradigm: build a perception matrix over many models × many items, derive Meme Probe Properties, and aggregate them into model-level phemotypes to reveal fine-grained capability structure across populations.

**Strengths:**

1/ In this work, the authors reconceptualize evaluation as an entangled world of models and data, formalizing a perception matrix that supports probe-level properties and interpretable phemotypes; this exposes phenomena hidden by traditional benchmarks (e.g., elite models failing items most models solve) and scales to thousands of models.

2/ The authors validate the framework on a huge amount of LLMs, showing clear probe/property distributions, family-level structure in phemotype space, and practical insights (accuracy-equal models with different behavioral profiles), demonstrating both scalability and interpretability.

**Weaknesses:**

1/ I suggest maybe the authors can consider broadening tasks (coding, RAG, agents) and adding head-to-head baselines (e.g., IRT-based compact sets, adversarial stress tests) to verify that phemotypes add incremental value beyond existing item- and ability-modeling approaches.

2/ An ablation on property definitions, thresholds, and clustering (e.g., Leiden parameters) would clarify robustness and generality.

3/ For the evaluation results, the authors can add multi-judge adjudication, uncertainty estimates, or multi-runs to enhance the robustness of the results.

**Questions:**

See weaknesses

---

> ### Author Response · Authors · 2025-11-25
> **Response to Reviewer esKs**
>
> **Dear Reviewer esKs:**
>
> Thank you very much for your positive recognition of our Probing Memes paradigm and the insights it reveals. Regarding your concerns about the experiments, our responses are as follows:
>
> **1. [*For Weakness #1*] Generalization to Other Tasks and Comparison with Baselines**
>
> Our current paradigm has been applied to tasks covering mathematical reasoning (MATH), general knowledge (MMLU), question answering (SimpleQA), and instruction following (IFEval, in the OpenLLM Leaderboard). Coding and RAG are indeed important domains that merit further investigation. The Probing Memes paradigm is expected to naturally accommodate these tasks, since their evaluations can be converted into a perception matrix, on which probe properties can be computed and, in turn, model phemotypes can be characterized. We plan to extend our evaluation to these settings in future work.
>
> In `Appendix E.2` we also study a dataset-reduction task inspired by this IRT-based line of work. In this setting, we use typicality to select a compact subset of items that achieves performance close to strong domain-specific IRT baselines (for example, on MATH and IFEval the Spearman rank correlations differ from the MetaBench baseline by only 0.011 and 0.003, respectively), and on some datasets even surpasses them (e.g., it achieves lower MAE than TinyBench on MATH, 0.026 vs 0.030, and on GPQA-Diamond, 0.023 vs 0.035). On other datasets (e.g., MMLU-Pro and BBH), the typicality-based subsets still show a substantial gap relative to the IRT baselines, reflecting a different design goal: the IRT methods are explicitly trained to optimize dataset reduction on each benchmark, whereas our method directly reuses probe properties for selection. This case study illustrates how probe properties can be used to select compact yet informative subsets of items while largely preserving evaluation fidelity, and, more broadly, how different properties (e.g., high-risk and high-surprise items) can be used together to guide property-driven subset construction.
>
> The remaining parts of `Appendix E`, in turn, illustrate applications of our paradigm to model selection, population-level analysis of models, and the case study of surprise phenomena.
>
> **2. [*For Weakness #2*] Parameter Choices and Robustness Analysis**
>
> We discuss the impact of parameter choices in the clustering and phemotype computation algorithms in `Appendix G.1`. For the Leiden clustering method, varying the resolution parameter around 1.0 (both higher and lower) has only a small effect on the results, which indicates that our experimental setting of 1.0 is a reasonable choice. `Appendix G.2` then analyzes other clustering-related parameters, in particular the similarity threshold. The experiments show that cluster-related metrics such as typicality and bridge are more sensitive to this threshold, because adjusting it changes the topology of the communities formed by the probes. We ultimately set the cluster similarity threshold to 0.9, since, as shown in `Figure 4`, some datasets exhibit generally high similarity; a stricter threshold is therefore required to clearly separate probe clusters.
>
> In addition, `Appendix F` studies robustness with respect to subsampling the model population used to compute probe properties. These experiments demonstrate the reliability of the results and highlight characteristic behaviors of several metrics, while also revealing insights about differences between model populations (`Appendix F.3`).
>
> **3. [*For Weakness #3*] Discussion of the Robustness of the Results**
>
> Thank you for highlighting the importance of reliability experiments; we fully agree that they are essential. However, large-scale LLM benchmarks face a serious cost issue, especially when dozens of models are involved, which makes multi-judge adjudication difficult to apply in practice. Nevertheless, in `Appendix F.1` we assess reliability by repeatedly subsampling the model population used in our experiments, and show that both the probe properties and the resulting phemotypes remain stable under such resampling.

---

### Official Review · Reviewer_2D5g · 2025-10-31

**Soundness:** 3
**Presentation:** 3
**Contribution:** 3
**Rating:** 6
**Confidence:** 3

**Summary:**

This paper introduces a new evaluation framwork to characterise data samples from test sets and large language models, focussing strongly on model-data interactions under the umbrella of memetics. The authors test 4507 LLMS with many dataset characterisations and model capability probes on these datasets.

**Strengths:**

1. Capability probing is an important aspect of foundation model evaluation. It helps us make evaluations more granular and extract more information from datapoints. This work makes a positive contribution in that direction.

2. The different probes and phemotypes are well-defined, in theory.

3. Testing 4507 models from OpenLLM Leaderboard is a substantial empirical contribution.

**Weaknesses:**

1. The meme framework seems unnecessary and, without a more grounded theoretical framework and justification in the context of LLM evaluations, could be removed. It leads to unnecessary terminology such as perception matrix, meme probes and phemotypes. The core contributions would not be affected if the metaphor were removed. It also leads to some sections becoming quite confusing to read("latent units of model capability that can be revealed through probing").

2. Several relevant papers have not been cited or referenced in this work. Flexibly defining new properties based on evaluation needs and the observation that datasets “contain a large number of seemingly simple questions that are nevertheless answered incorrectly by some elite models” are both insights established in [1]. Then, “models with similar accuracy may succeed on very different types of items” is adopted from [2]. Several of the meme probe properties are established frameworks already: difficulty has been defined and used in [3] as as the IRT model in [4](cited elsewhere in the paper), the  . Other forms of capabilities are tested by works such as [5].

3. This work only considers binary (0,1) evaluations. Other works [1,4] take into consideration heterogeneous metrics such as accuracy (0/1), BLEU score ([0,1]) which is a more realistic setup, ensuring that the framework of evaluation takes into consideration all datasets.

4. There is no insight into whether the phemotypes are correlated to each other and to what extent. This paper would benefit from quantitative studies to determine this.

Minor
1. Figure 5 and 6 are not easily interpretable, more details and insights would be beneficial.
2. T-sne is known to result in spurious clusters. UMAP, on the other hand, preserves both local and global structure in the data and is a better algorithm for data visualisation.

[1] Ghosh et al. ONEBENCH to Test Them All: Sample-Level Benchmarking Over Open-Ended Capabilities, ACL 2025

[2] Goel et al. Great Models Think Alike and this Undermines AI Oversight, ICML 2025

[3] Prabhu et al. Efficient Lifelong Model Evaluation in an Era of Rapid Progress, NeurIPS 2024

[4] Polo et al. tinyBenchmarks: evaluating LLMs with fewer examples, ICML 2024

[5] Alyahya et al. ZEROSUMEVAL: An Extensible Framework For Scaling LLM Evaluation with Inter-Model Competition, ACL 2025

**Questions:**

1. “Certain elite models that excel in overall metrics nevertheless display anomalous errors on questions that most other models solve with ease”. Could this be the case due to train-test contamination, the presence of test samples during pretraining? What would be a way to test this?
2. Is there a unique insight into evaluation provided in the memetic definition (not talking about the probe properties here)? The interaction of data and model is basically how evaluation is done. Other works such as [1] use more applicable definitions from other fields, such as social choice theory and also conduct experiments to compare their framework with other methods in that field.
3. To measure I_j, which is it logarithmic? It seems invariant to “reduces the influence of weak models while emphasizing the contribution of stronger models”.

[1] Ghosh et al. ONEBENCH to Test Them All: Sample-Level Benchmarking Over Open-Ended Capabilities, ACL 2025

---

> ### Author Response · Authors · 2025-11-25
> **Response to Reviewer 2D5g (Part 1/3)**
>
> **Dear Reviewer 2D5g:**
>
> We sincerely appreciate your recognition of the importance of our research problem and your positive comments on our metrics. We would like to address these concerns, which mainly relate to the motivation for the meme framing and the comparison with other works, from the following aspects:
>
> **1. [*For weakness #1, Question #2*] Motivation for Introducing the Concept of Memes**
>
> We apologize for the insufficient explanation provided when introducing the concepts. Our motivation for adopting the "meme" framing is to model the fine-grained behavioral regularities revealed through data–model interaction at a resolution that traditional "skills" or "capabilities" terms cannot express. This perspective is consistent with prior work. For example, Birchall et al. (reference [6] in our comments) describe reasoning patterns in LLMs as meme-like units. Our contribution is to make this idea formal and operational, enabling a structured and extensible evaluation paradigm based on memes, probe properties, and phemotypes.
>
> Concretely, we define a meme space $\mathcal{V}$ = {$\mu_1,\mu_2,\dots,\mu_R$}, where each elementary meme represents a reusable, compositional behavioral tendency that remains latent unless elicited by specific probes. Probes serve as controlled stimuli that expose subsets of $\mathcal{V}$, and phemotypes formalize the resulting observable patterns of these subsets into quantifiable descriptors.
>
> Without this "meme" framing, several key constructs in our evaluation, such as probe properties like uniqueness and bridge, and the compositional phemotypes, would be ill-defined under a coarse capability view.
>
> We appreciate the reviewer's suggestion and have improved clarity by explicitly formalizing the meme, probe property, and phemotype spaces in `Appendix B`. This addition clarifies the underlying structure but does not affect the method or results.
>
> **2. [*For Weakness #2*] Discussions about Related Works**
>
> Thank you very much for pointing us to these papers; we have added citations to all of them in the revised paper. Specifically, [1] and [3] are discussed in `Appendix A.4` in connection with dataset-level evaluation, [5] is covered in `Appendix A.5` as a framework for multidimensional LLM capacities, and [2] is discussed in `Appendix A.2`. [4] is already included in `Appendix A.1`.
>
> They also differ in clear and significant ways:
>
> [1] treat datasets from diverse tasks and evaluation metrics as a single large pool, from which users can query a model's performance on specific tasks (e.g., abstract algebra). In contrast, our approach does not define model capabilities with respect to particular tasks, but instead evaluates intrinsic model characteristics and abilities, such as coping with high-risk and high-surprise phenomena.
>
> [2] introduce the CAPA metric to capture error agreement between two models, showing that models with similar overall accuracy can still make very different mistakes. The Probing Memes paradigm goes a step further: it first evaluates items via probe properties and then derives capability-oriented characterizations of models, moving from merely showing that models err on different items to explaining how they differ along specific capability dimensions. For example, the phemotype Vigilance measures a model’s ability to cope with high-risk, high-surprise phenomena, allowing us to distinguish models with similar accuracy but systematically different behavioral patterns.
>
> [3] and [4] aim to construct informative subsets in order to reduce the cost of large-scale evaluation. While we also make use of a difficulty measure, it is only one dimension among our full set of metrics, and it is computed in a different way (IRT fits difficulty as a latent parameter). In our setting, difficulty primarily serves the design of each probe (item) in the dataset and the characterization of complex model behaviours.
>
> [5] measure model capabilities via competitive games between models, focusing on relative rankings within a fixed interaction scenario. In contrast, the Probing Memes paradigm applies more broadly: it can be instantiated directly on existing benchmarks for fine-grained capability analysis, and its probe properties can be extended or tailored to different evaluation goals and tasks.
>
> **Reference**
> [1] Ghosh et al. ONEBENCH to Test Them All: Sample-Level Benchmarking Over Open-Ended Capabilities, ACL 2025
> [2] Goel et al. Great Models Think Alike and this Undermines AI Oversight, ICML 2025
> [3] Prabhu et al. Efficient Lifelong Model Evaluation in an Era of Rapid Progress, NeurIPS 2024
> [4] Polo et al. tinyBenchmarks: evaluating LLMs with fewer examples, ICML 2024
> [5] Alyahya et al. ZEROSUMEVAL: An Extensible Framework For Scaling LLM Evaluation with Inter-Model Competition, ACL 2025
> [6] Birchall et al. Parrots Are All You Need: A Memetic Framework for Apparent Reasoning in LLMs, PhilArchive 2025

---

> ### Author Response · Authors · 2025-11-25
> **Response to Reviewer 2D5g (Part 2/3)**
>
> **3. [*For Weakness #3*] Discussion about Metrics Design**
>
> Your concern about extending the metrics beyond binary 0/1 indicators is very insightful. In fact, our metrics can also be applied in non-binary settings. For example, one can introduce a threshold so that non-binary values are mapped to 0/1, or directly keep the matrix as non-binary and replace the $\phi$-coefficient with a standard correlation coefficient (since $\phi$-coefficient is essentially the 0/1 discrete special case of a correlation measure).
>
> Some existing metrics, such as BLEU, are primarily designed to compare the similarity between output texts and are typically used in translation scenarios. By contrast, our proposed properties are intended to characterize model behaviour, which provides a deeper and broader ceiling for evaluation.
>
> **4. [*For weakness #4*] The Correlation of Phemotypes**
>
> Some phemotypes are indeed correlated to some extent. As shown in `Table 1` in `Section 3`, the computation of Mastery, Transfer, and Ingenuity all involves difficulty. As described in `Appendix B`, our phemotypes are instantiated from a broad design space, and their design depends on which aspects of model ability the user wishes to probe. The five phemotypes used in this paper are five carefully chosen instances that we believe can reveal practically useful characteristics of models. We include difficulty in three of them because, while accounting for item-specific effects (for example, in Ingenuity), we consider it more valuable when a model is better at solving difficult items. However, even though difficulty enters multiple definitions, the phemotypes do not change in a perfectly synchronized way. For instance, in `Figure 7` and `Table 3`, for *Gemini-2.5-Flash* and *Qwen3-32B (CoT)*, Transfer increases for the latter (from 42.0 to 48.9) while Ingenuity decreases (from 43.5 to 42.1).
>
> **5. [*For Question #1*] The underlying reasons behind the insights**
>
> Thank you very much for your interest in the insights we report. For the phenomenon that "certain elite models that excel in overall metrics nevertheless display anomalous errors on questions that most other models solve with ease," we hypothesize that, in addition to possible data contamination, catastrophic forgetting may also play a role (as larger models acquire more new knowledge, previously learned knowledge may effectively "disappear").
>
> Our paradigm provides a way to systematically surface such issues, since these items can be easily identified using our probe properties. We further investigate some of these insights. For example, for the high-surprise phenomenon on the MMLU dataset, we conduct experiments (`Appendix E.4`) and find that, for some items that most models struggle to answer correctly but weaker models get right, a portion of these cases can be attributed to random guessing in multiple-choice questions (MCQs), which to some extent leads to high surprise scores. However, a substantial number of responses from weaker models exhibit strong certainty, indicating that these "weak" models are indeed able to solve some genuinely difficult problems.
>
> However, some reasons behind the insights touch on lower-level mechanisms of model training and internal representation that our paradigm alone cannot resolve. We view this as an important direction for future work and plan to conduct dedicated studies based on these findings.

---

> ### Author Response · Authors · 2025-11-25
> **Response to Reviewer 2D5g (Part 3/3)**
>
> **6. [*For Question #3, Weakness-minor #2*] Details of Algorithm and Visualization**
>
> The weighting term I\_j is based on the notion of self-information (line 191 in revision), which measures how informative a model's error is.
> A weak model fails on many probes, so its errors are high-probability events and therefore carry very little information.
> Conversely, a strong model rarely makes mistakes, so its errors carry much higher information.
> For example, when a model's accuracy is 0.2, 0.5, and 0.8, the corresponding I\_j values are 0.22, 0.69, and 1.61.
> This means that an error from a strong model contributes over 7 times more information than an error from a weak one.
> Thus, I_j adjusts the risk score by down-weighting low-information co-failures from weak models while amplifying the high-information failures of stronger models.
> This prevents trivial and ubiquitous errors from dominating the risk metric and ensures that high-risk probes are identified through the rare but informative failures of strong models.
>
> For visualizing the commonality and divergence among models revealed by phemotypes, `Figure 9` presents a t-SNE embedding. We additionally include a UMAP visualization in `Appendix E.3` (`Figure 16`), which leads to the same qualitative conclusions.
>
> **7. [*For weakness-minor #1*] Clarity of Figure 5 and Figure 6**
>
> We apologize that the introduction of the properties and phemotypes contains imperfections. As described in `Appendix B`, our properties are carefully designed from a broad property space. Therefore, we aim to use a 3D visualization to show how all probes in a dataset are positioned when multiple properties are considered together. `Figure 5` shows that when looking at a single property (a single axis), MMLU-Redux has the lowest probe difficulty while SimpleQA has the highest. When considering three properties jointly, SimpleQA contains many probes with high difficulty, high uniqueness, and high surprise, which are largely absent in the other datasets. These probes indicate that many weaker models answer difficult items correctly (difficulty and surprise), and that the behavioral patterns revealed by these items differ from most other probes (uniqueness).
>
> Since four of the five phemotypes (all except Astuteness) are computed from combinations of two properties, we aim to make the contribution of each property more transparent. `Figure 6` illustrates the weight composition of the five phemotypes. Although `Figure 6` is plotted as a line chart, it is conceptually closer to a histogram: the horizontal axis represents the contribution weight of each probe to the corresponding phemotype, and the vertical axis represents the proportion of probes with that weight (which can be interpreted as proportion). However, because histograms are not suitable for comparing multiple distributions simultaneously, we present the results using line plots.

---

### Official Review · Reviewer_WKs2 · 2025-10-31

**Soundness:** 2
**Presentation:** 1
**Contribution:** 2
**Rating:** 2
**Confidence:** 3

**Summary:**

This paper introduces the "Probing Memes" paradigm for LLM evaluation, drawing conceptually on Dawkins' theory of memes as replicating cultural units. The authors propose treating LLM capabilities as composed of latent "memes" that can be revealed through carefully designed probes. They construct a “perception matrix” from model-data interactions.

**Strengths:**

- The problem of better analyzing model evaluations is important
- Some of the proposed metrics are interesting

**Weaknesses:**

1. **Unclear motivation for the meme framing**: The conceptual link to Dawkins and memetics feels forced and adds unnecessary complexity without clear benefit. The core contributions (probe properties and phemotypes) could stand without this metaphor. The paper states that memes are "latent units of model capability that can be revealed through probing", but this is more of a renaming than a substantive theoretical contribution. Why is this memetics lens necessary or illuminating?

2. **Limited differentiation in results**: The key weakness is visible in Figure 7, which shows phemotype scores tracking remarkably closely with accuracy across models. While the paper claims to reveal "fine-grained phenomena invisible under conventional evaluations," the phemotypes appear highly correlated with overall performance. This contradicts the motivation that current approaches "obscure fine-grained differences" - if the proposed phemotypes largely parallel accuracy, what additional insight do they provide?

3. **Inconsistent with cited literature**: The paper cites Schilling-Wilhelmi et al. on IRT where different analysis methods reveal significant ranking changes. However, Figure 7 shows surprisingly consistent orderings across metrics, undermining this motivation.

4. **Unclear practical utility**: What should practitioners do differently with phemotypes versus accuracy? The paper doesn't provide clear guidance on how these metrics inform model selection, dataset design, or capability assessment in practice.

**Questions:**

1. Can you provide examples where phemotypes lead to different model rankings or selection decisions compared to accuracy?

2. In Figure 7, why do phemotypes correlate so strongly with accuracy if they're capturing distinct capability dimensions? What percentage of variance in phemotypes is explained by accuracy?

3. What is the empirical evidence that the meme metaphor provides insight beyond standard psychometric approaches (IRT, factor analysis, etc.)?

4. How should researchers choose which phemotype to optimize for? Are some phemotypes more important for certain applications?

---

> ### Author Response · Authors · 2025-11-25
> **Response to Reviewer WKs2 (Part 1/2)**
>
> **Dear Reviewer WKs2:**
>
> We would like to express our gratitude to your thorough and invaluable review. We appreciate your high regard for the concept ``Memes'' and presentation of experimental results of our work. We respond to your suggestions as follows:
>
> **1. [*For Weakness #1*] Motivation for Introducing the Concept of Memes**
>
> Thank you for raising this point. Our motivation for adopting the "meme" framing is to model the fine-grained behavioral regularities revealed through data–model interaction at a resolution that traditional "skills" or "capabilities" terms cannot express. This perspective is consistent with prior work. For example, Birchall et al. (reference [1] in our comments) describe reasoning patterns in LLMs as meme-like units. Our contribution is to make this idea formal and operational, enabling a structured and extensible evaluation paradigm based on memes, probe properties, and phemotypes.
>
> Concretely, we define a meme space $\mathcal{V}$ = {$\mu_1,\mu_2,\dots,\mu_R$}, where each elementary meme represents a reusable, compositional behavioral tendency that remains latent unless elicited by specific probes. Probes serve as controlled stimuli that expose subsets of $\mathcal{V}$, and phemotypes formalize the resulting observable patterns of these subsets into quantifiable descriptors.
>
> Without this "meme" framing, several key constructs in our evaluation, such as probe properties like uniqueness and bridge, and the compositional phemotypes, would be ill-defined under a coarse capability view.
>
> We appreciate the reviewer's suggestion and have improved clarity by explicitly formalizing the meme, probe property, and phemotype spaces in Appendix B. This addition clarifies the underlying structure but does not affect the method or results.
>
>
> **2. [*For Weakness #2 & #3, Question #1 & #2*] Differentiation in Results**
>
> Although the ranking list of phemotypes exhibit a global trend similar to accuracy, this does not imply redundancy. A global correlation is both reasonable and necessary, as overall capability naturally influences multiple behavioral dimensions; if phemotypes were entirely uncorrelated with accuracy, they would fail to reflect meaningful model behavior.
>
> The core contribution of the Probing Memes paradigm lies in decomposing this global trend into interpretable behavioral dimensions, thereby revealing structural differences that accuracy alone cannot capture.
>
> Once this decomposition is applied, substantial local divergences emerge. As shown in `Figure 7` and `Table 4`, many models exhibit phemotype trends that move opposite to accuracy. For example, although *glm-4.5* and *doubao-seed-1-6-flash-250715* have similar accuracies (rank 44 vs. 46), *doubao-seed-1-6-flash-250715* significantly outperforms *glm-4.5* on Vigilance, Mastery, and Transfer. Likewise, *MiniMax-Text-01 (CoT)* and *spark-X1 (IR)* have nearly identical accuracy (60.6 vs. 60.5; ranks 32 vs. 33) but differ by up to 10 positions across all phemotype dimensions; *qwen2.5-72b-a22b (IR)* shows a similar 10-rank shift in Ingenuity.
>
> These pronounced local differences are precisely the phenomena obscured by overall accuracy, and phemotypes provide a systematic way to characterize such behavioral structure.
>
> **Reference**
> [1] Birchall et al. Parrots Are All You Need: A Memetic Framework for Apparent Reasoning in LLMs (PhilArchive 2025)

---

> ### Author Response · Authors · 2025-11-25
> **Response to Reviewer WKs2 (Part 2/2)**
>
> **3. [*For Weakness #4, Question #1 & #4*] Practical Utility and Application**
>
> We have added substantial experiments as well as the missing explanations to address the practical utility. In `Section 5` and `Appendix E` (Applications and Case Studies), we outline several potential uses and application directions, and illustrate them with experiments:
>
> - **Phemotype-guided model selection (Detailed in `Appendix E.1`)**
> Our Probing Memes paradigm can naturally be applied to realistic multi-agent routing tasks, where items are dispatched to different agents based on their phemotype profiles. We use phemotype scores computed on MATH-500 to identify, among models with similar overall accuracy, those that are better suited for hard items and those that are better suited for easy items. On MATH-FULL, for one representative pair of similarly accurate models, difficulty-based routing attains a union accuracy of 77.02\%, compared to 72.24\% and 76.02\% for the two individual models and 73.89\%±0.29 for a balanced partition baseline. These experiments show how phemotype scores can be used to choose models that better match downstream requirements than considering accuracy alone.
>
> - **Property-guided dataset construction (Detailed in `Appendix E.2`)**
> Our paradigm can be used for property-guided subset construction. We study a concrete dataset reduction task inspired by IRT-based work. In this setting, we use *typicality* to select a compact subset of items. The resulting subset achieves performance close to strong domain-specific IRT baselines (on MATH and IFEval the Spearman rank correlations differ from the MetaBench baseline by only 0.011 and 0.003, respectively). It even surpasses these baselines (e.g., it achieves lower MAE than TinyBench on MATH, 0.026 vs 0.030, and on GPQA-Diamond, 0.023 vs 0.035). This concrete case illustrates how probe properties can be used to select compact yet informative subsets of items and preserve evaluation fidelity. However, the typicality-based subsets still show a substantial gap relative to the IRT baselines (e.g., on MMLU-Pro and BBH), reflecting a different design goal: the IRT methods are explicitly trained to optimize dataset reduction on each benchmark, whereas our method directly uses probe properties for selection. Depending on the specific research goal, different properties can be used together to guide subset construction (one can construct reduced datasets that focus on high-risk and high-surprise items).
>
> - **Methods for helping researchers conduct in-depth analysis of relationships among models (Detailed in `Appendix E.3`)**
> We use phemotypes to characterize over 4,000 models in the OpenLLM Leaderboard and visualize their distribution in the memetic space. Models that share the same base model tend to lie close to each other in this space. A similar conclusion holds for training strategies: nearby models also tend to adopt similar training strategies. This demonstrates that the design of phemotypes provides a practical way to uncover similarities and differences among LLMs.
>
> **4. [*For Weakness #3, Question #3*] Insight beyond Other Works**
>
> The Probing Memes paradigm is designed for fine-grained, item-level analysis and for characterizing model phemotypes. It quantifies phenomena that conventional evaluations miss, for instance, strong models that still make anomalous errors on questions most other models solve correctly, patterns that IRT- or factor-analysis–based approaches cannot capture. This yields a practical diagnostic tool: model designers can identify model-specific, high-surprise errors and target them for optimization. From the user side, models can be selected according to application needs; `Appendix E.1` provides a concrete case study on model selection.
>
> The paradigm itself is our main contribution. As shown in `Appendix B`, its components (probe properties and phemotypes), are extensible and can be adapted to different applications. It applies to arbitrary datasets and model populations, enabling diverse insights. For example, `Appendix F.3` reveals a counter-intuitive "degradation" phenomenon across reasoning modes: under more complex reasoning (usually assumed to improve capability), some items that were previously easy or low-risk become harder or more hazardous.
>
> The metrics discussed in Schilling-Wilhelmi et al. on IRT (reference [1] in this comment), such as precision and Hamming loss, are based solely on surface-level correctness. In contrast, the Probing Memes paradigm evaluates models via scores that reflect underlying capabilities (e.g., coping with risk and high-surprise phenomena). As shown in `Table 4` (`Appendix D.2`), phemotype-based ranks can differ substantially from accuracy ranks.
>
> **Reference**
>
> [1] Schilling-Wilhelmi et al. Lifting the benchmark iceberg with item-response theory (ICLR 2025)

---

### Author Response · Authors · 2025-11-25
**Summary of Revisions and Errata**

**For all reviewers:** We sincerely thank you for the time and efforts you devoted to reviewing our paper. Inspired by your comments and suggestions, **we have made the following revisions and additions**:

1. We have expanded Section 2 and Appendix B to provide more detailed formal definitions of memes, probe properties, and phemotypes.
2. We have added a new Section 5, *Applications and Case Studies*, together with Appendix E, to illustrate how our paradigm can be applied to downstream tasks.
3. We have added Appendix F, *Population-relative Analyses of Probe Properties and Phemotypes*, which studies the impact of different model population sizes and presents results under different reasoning populations.
4. We have added Appendix G, *Effects of parameter choices on the Probing Memes paradigm*, which discusses how algorithmic hyperparameters affect the results.
5. We have extended Appendix A to include additional discussion of related work.
6. We have corrected typographical errors and improved the clarity and fluency of the writing throughout the paper.

**Errata:** In the OpenLLM Leaderboard experiments, we inadvertently used an older implementation of the *uniqueness* metric. The revised version now uses the correct implementation. The previous code attempted to avoid `log(0)` by adding a tiny value (`1e-12`) to probabilities. Under `float32`, however, `1 - 1e-12` still evaluates to `1`, causing `log(0)` and producing `NaN` for cases where p=1. The corrected version sets the entropy to zero whenever p=0 or p=1, which is mathematically consistent since both contribute zero entropy.

This change **affects a small subset of *uniqueness* values** (those previously producing `NaN`), which in turn causes **minor differences** in the phemotype **Ingenuity** rankings for a few models. In Figure 8, a small number of missing points were due to `NaN` uniqueness values in the old code, but the Probe distributions remain unchanged. Figure 9 is virtually unaffected. **None of the main conclusions of the paper are impacted.**

**All experiments in Section 3 use the correct implementation and are therefore unaffected.**

We apologize for this oversight and have updated **Figures 8, 9, and Table 12 (Ing)** accordingly. The old implementation is included in the Supplementary Material as `probe_attributes_wrong_version(uniqueness).py` (line 43) **for transparency**.

---

### Author Response · Authors · 2025-12-03
**Overall Summary**

Dear Area Chair,

We sincerely appreciate your effort in evaluating this submission under these exceptional circumstances. We understand that the recent OpenReview incident has significantly increased the workload for the program committee and area chairs, and we are truly grateful for your time and service.

To support your evaluation, we prepared a concise summary of the main reviewer concerns and how our rebuttal and revisions address them. We sincerely appreciate the reviewers' efforts and constructive feedback. ***Reviewer iqfE also noted that their current score is 2 and may be raised to 4 or 6 if the paper's significance and novelty become clear during the discussion***.

Below, we outline the key concerns and strengths identified in the reviews.

## Important Concerns Raised by the Reviewers

**1. [*Mentioned by Reviewers WKs2, 2D5g, and iqfE*] Motivation for Introducing the Concept of Memes**

To clarify the motivation for the memetic perspective, we make explicit how memes connect to our paradigm, emphasizing that our paradigm is needed beyond existing evaluation approaches. We also refine the definitions of memes and phemotypes in `Section 2` and provide further details on their spaces in `Appendix B`. For a more detailed discussion, please refer to `Response to Reviewer WKs2 (Part 1/2), 1`, `Response to Reviewer 2D5g (Part 1/3), 1`, and `Response to Reviewer iqfE (Part 1/4), 1`.

**2. [*Mentioned by Reviewers WKs2, esKs, and iqfE*] Practical Utility and Applications**

We introduce several practical applications of our paradigm in the added `Section 5`. The experimental results demonstrate its practical value and potential. For a more detailed discussion, please refer to `Response to Reviewer WKs2 (Part 2/2), 3`, `Response to Reviewer esKs, 1`, and `Response to Reviewer iqfE (Part 3/4), 5`.

**3. [*Mentioned by Reviewers WKs2, 2D5g, and iqfE*] Discussion of Related Work and Insights Beyond**

Some reviewers asked for a clearer comparison with related work. We analyze how our paradigm differs from and improves upon these works, and add further discussion in `Appendix A`. For more details, please refer to `Response to Reviewer WKs2 (Part 2/2), 4`, `Response to Reviewer 2D5g (Part 1/3), 2`, and `Response to Reviewer iqfE (Part 2/4), 2`.

**4. [*Mentioned by Reviewers esKs, and iqfE*] Discussion of Experimental Parameters and Result Reliability**

Our paradigm introduces many novel metric computations and involves a large model population. In `Appendix F`, we show the stability of these metrics under different population scales, and in `Appendix G`, we report the effects of varying parameter choices. Please refer to `Response to Reviewer esKs, 2` and `Response to Reviewer iqfE (Part 2/4), 4` for further details.

**5. [*Mentioned by Reviewers 2D5g, and iqfE*] Underlying Reasons Behind the Phenomena We Uncover**

Our paradigm reveals several interesting phenomena (e.g., high-surprise cases). To further investigate the underlying reasons for these observations, we conduct an experimental study in `Appendix E.4`, focusing on high-surprise phenomena in MMLU-Redux. The ability to directly capture property-specific issues highlights the paradigm’s potential for fine-grained dataset analysis. For more details, please refer to `Response to Reviewer 2D5g (Part 2/3), 5` and `Response to Reviewer iqfE (Part 3/4), 5`.

## Key Strengths Highlighted by the Reviewers

**1. [*Mentioned by Reviewers WKs2, 2D5g, esKs, and iqfE*] Novel Evaluation Paradigm and Interpretable Metrics.**

Recognizes the importance of the problem and acknowledges that the proposed metrics are interesting (WKs2); Notes that probes and phemotypes are well-defined(2D5g); Comments that the proposed perception-matrix approach supports probe-level properties and interpretable phemotypes (esKs); Considers the paradigm's joint analysis along the sample and model dimensions to be an important methodological contribution (iqfE).

**2. [*Mentioned by Reviewers 2D5g, and iqfE*] Comprehensive Experiments.**

Comments that the experiments are extensive and substantiate the proposed paradigm.

**3. [*Mentioned by Reviewers 2D5g, esKs, and iqfE*] Significant Insights and Contributions**

Notes that the paradigm enables more granular analysis and extracts more information than existing approaches (2D5g); Believes that this paradigm exposes phenomena that are hidden by traditional benchmarks (esKs); Acknowledges that the paradigm is well-motivated and constitutes an important research direction (iqfE).

---

### Meta-Review · Area_Chair_HLzg · 2026-01-03

**Summary:**

Reviewers WKs2, 2D5g, and iqfE criticized the "meme" framing as unnecessary, arguing it added complexity without theoretical benefit over standard terms like "latent capabilities". Reviewers WKs2 and iqfE questioned the practical utility of the proposed metrics compared to standard accuracy or IRT. Reviewer esKs requested the inclusion of broader tasks (coding, RAG) and more robust evaluation methods (multi-judge adjudication), while Reviewer iqfE raised concerns regarding the reliability of probes with varying model population sizes and the mechanism behind specific findings like "surprise".

**Reviewer Concerns:**

**Addressed Concerns:**
* **Practical Utility:** Addressed for **WKs2** and **iqfE** by adding Section 5 and Appendix E, which demonstrated applications in phemotype-guided model routing and property-guided dataset reduction.
* **Reliability and Stability:** Addressed for **iqfE** and **esKs** by adding Appendix F, which analyzed the stability of properties under population subsampling and resampling.
* **Specific Metric Mechanisms:** Addressed for **iqfE** regarding the "surprise" metric on MMLU-Pro (investigating random guessing in Appendix E.4) and the choice of phi-coefficient over entropy.
* **Related Work:** Addressed for **2D5g** and **iqfE** by expanding the discussion and citations (e.g., OneBench, CAPA) in Appendix A.

**Outstanding Concerns:**
* **Meme Framing:** The fundamental disagreement regarding the necessity and clarity of the "meme" metaphor remains for **WKs2**, **2D5g**, and **iqfE**, as the authors defended the terminology rather than modifying it.
* **Broadening Tasks:** **esKs**'s request to extend the evaluation to coding, RAG, and agents was acknowledged but deferred to future work.
* **Evaluation Robustness (Methodology):** **esKs**'s suggestion to add multi-judge adjudication or multi-runs to the leaderboard evaluation was not followed by the authors due to cost constraints.
* **Correlation with Accuracy:** **WKs2**'s concern that phemotypes strongly correlate with accuracy, potentially limiting their distinct insight, persists in the results.

**Reviewer Scores:**

3. Reviewer Scores
* **Reviewer WKs2:** 2 -> 4. The reviewer's major objection to the "unclear practical utility" was addressed by the new experiments on model selection and dataset construction. However, their strong opposition to the "meme" framing and the lack of differentiation from accuracy suggests a hesitancy to fully endorse the paper.
* **Reviewer 2D5g:** 6 -> 6. This reviewer was already marginally positive. The rebuttal addressed their specific requests for citations and clarified metric definitions. While the reviewer's concern regarding the meme framework was unlikely to be resolved, the core empirical contribution was acknowledged, likely keeping the score stable.
* **Reviewer esKs:** 6 -> 6. The reviewer is unlikely to increase their score because the authors did not address the request to broaden tasks (coding, RAG) and did not follow the suggestion for multi-judge adjudication due to cost, leaving the experimental scope and robustness verification partially unfulfilled.
* **Reviewer iqfE:** 2 -> 6. The reviewer explicitly stated they might raise their score to 6 if the paper's significance and practical value became clear. The authors directly addressed this by adding substantial results on practical applications (routing, reduction) and addressing the reliability/population size concerns, meeting the reviewer's specific conditions for an upgrade.

---

### Decision · Program_Chairs · 2026-01-26

Reject